# Bounds for the smallest eigenvalue of the NTK for arbitrary spherical data of arbitrary dimension

**Kedar Karhadkar[1], Michael Murray[1], Guido Montúfar[1,2,3]**
[1]Department of Mathematics, UCLA
[2]Department of Statistics & Data Science, UCLA
[3]Max Planck Institute MiS

## Abstract

Bounds on the smallest eigenvalue of the neural tangent kernel (NTK) are a key ingredient in the analysis of neural network optimization and memorization. However, existing results require distributional assumptions on the data and are limited to a high-dimensional setting, where the input dimension $d_0$ scales at least logarithmically in the number of samples $n$. In this work we remove both of these requirements and instead provide bounds in terms of a measure of distance between data points: notably these bounds hold with high probability even when $d_0$ is held constant versus $n$. We prove our results through a novel application of the hemisphere transform.

## 1 Introduction

A popular approach for studying the optimization dynamics of neural networks is analyzing the neural tangent kernel (NTK), which corresponds to the Gram matrix obtained from the Jacobian of the network parametrization map (Jacot et al., 2018). When the network parameters are adjusted by gradient descent, the network function follows a kernel gradient descent in function space with respect to the NTK. By bounding the smallest eigenvalue of the NTK away from zero it is possible to obtain global convergence guarantees for gradient descent parameter optimization (Du et al., 2019b; Oymak & Soltanolkotabi, 2020) as well as results on generalization (Arora et al., 2019a; Montanari & Zhong, 2022) and data memorization capacity (Montanari & Zhong, 2022; Nguyen et al., 2021; Bombari et al., 2022). These key advances highlight the importance of deriving tight, quantitative bounds for the smallest eigenvalue of the NTK at initialization.

While initial breakthroughs on the convergence of gradient optimization in neural networks (Li & Liang, 2018; Du et al., 2019a; Allen-Zhu et al., 2019) required unrealistic conditions on the width of the layers, subsequent and substantive efforts have reduced the level of overparametrization required to ensure that the NTK is well conditioned at initialization (Zou & Gu, 2019; Oymak & Soltanolkotabi, 2020). In particular, Nguyen (2021); Nguyen et al. (2021); Banerjee et al. (2023) showed that layer width scaling linearly in the number of training samples $n$ suffices to bound the smallest eigenvalue and Montanari & Zhong (2022); Bombari et al. (2022) obtained results for networks with sub-linear layer width and the minimum possible number of parameters $\tilde{\Omega}(n)$ up to logarithmic factors. However, and as discussed in Section 2, the bounds provided in prior works require that the data is drawn from a distribution satisfying a Lipschitz concentration property, and only hold with high probability if the input dimension $d_0$ scales as $\sqrt{n}$ (Bombari et al., 2022) or polylog($n$) (Nguyen et al., 2021). These existing results therefore require that the dimension of the data grows unbounded as the number of training samples $n$ increases and as such there is a gap in our understanding of cases where the data is sampled from a fixed, or lower-dimensional space.

In this work we present new lower and upper bounds on the smallest eigenvalue of a randomly initialized, fully connected ReLU network: compared with prior work, our results hold for arbitrary

38th Conference on Neural Information Processing Systems (NeurIPS 2024).

data on a sphere of arbitrary dimension. Our techniques are novel and rely on the hemisphere transform as well as the addition formula for spherical harmonics.

We study neural networks denoted as functions $f : \mathbb{R}^{d_0} \times \mathcal{P} \to \mathbb{R}$, where $\mathcal{P}$ is an inner product space. To be clear, $f(\boldsymbol{x}; \boldsymbol{\theta})$ denotes the output of the network for a given input $\boldsymbol{x} \in \mathbb{R}^{d_0}$ and parameter choice $\boldsymbol{\theta} \in \mathcal{P}$. For brevity we occasionally write $f(\boldsymbol{x})$ in place of $f(\boldsymbol{x}; \boldsymbol{\theta})$ if the context is clear. We use $n$ to denote the size of the training sample, $d_0$ the dimension of the input features, $L$ the network depth, $d_l$ the width of the $l$th layer and $\sigma : \mathbb{R} \to \mathbb{R}$ the ReLU activation function. Given $n$ input data points $\boldsymbol{x}_1, \cdots, \boldsymbol{x}_n \in \mathbb{R}^{d_0}$ we write $\boldsymbol{X} = [\boldsymbol{x}_1, \cdots, \boldsymbol{x}_n] \in \mathbb{R}^{d_0 \times n}$ and define $F : \mathcal{P} \to \mathbb{R}^n$ to be the evaluation of the network on these $n$ data points as a function of the parameter $\boldsymbol{\theta}$,

$$F(\boldsymbol{\theta}) = [f(\boldsymbol{x}_1; \boldsymbol{\theta}), \cdots, f(\boldsymbol{x}_n; \boldsymbol{\theta})]^T.$$

We define the neural tangent kernel (NTK) of $F$ as

$$\boldsymbol{K}(\boldsymbol{\theta}) = (\nabla_{\boldsymbol{\theta}} F(\boldsymbol{\theta}))^* (\nabla_{\boldsymbol{\theta}} F(\boldsymbol{\theta})) \in \mathbb{R}^{n \times n}, \tag{1}$$

where the gradient $\nabla$ and adjoint $*$ are taken with respect to the inner product on $\mathcal{P}$ and the Euclidean inner product on $\mathbb{R}^n$. More explicitly $[\boldsymbol{K}(\boldsymbol{\theta})]_{ik} = \langle \nabla_{\boldsymbol{\theta}} f(\boldsymbol{x}_i; \boldsymbol{\theta}), \nabla_{\boldsymbol{\theta}} f(\boldsymbol{x}_k; \boldsymbol{\theta}) \rangle$. For convenience we write $\boldsymbol{K}$ in place of $\boldsymbol{K}(\boldsymbol{\theta})$. We are concerned with the minimum eigenvalue $\lambda_{\min}(\boldsymbol{K})$, which depends both on the input data $\boldsymbol{X}$ and the parameter $\boldsymbol{\theta}$. We say the dataset $\boldsymbol{x}_1, \cdots, \boldsymbol{x}_n$ is $\delta$-*separated* for $\delta \in (0, \sqrt{2}]$ if $\min_{i \neq k} \min(\|\boldsymbol{x}_i - \boldsymbol{x}_k\|, \|\boldsymbol{x}_i + \boldsymbol{x}_k\|) \geq \delta$, which is a measure of distance in direction.

**Main contributions.** Our results are for data that lies on a sphere and is $\delta$-separated for some $\delta \in (0, \sqrt{2}]$. Unlike prior work we do not make any assumptions on the distribution from which the data is sampled, e.g., uniform on the sphere or Lipschitz concentrated, and we do not require the input dimension $d_0$ to scale with the number of samples $n$.

- In Theorem 1 we consider shallow ReLU networks with input dimension $d_0$ and hidden width $d_1$ and prove that if $d_1 = \tilde{\Omega}(\|\boldsymbol{X}\|^2 d_0^3 \delta^{-2})$ then with high probability $\lambda_{\min}(\boldsymbol{K}) = \tilde{\Omega}(d_0^{-3} \delta^2)$. Furthermore, defining $\delta' = \min_{i \neq k} \|\boldsymbol{x}_i - \boldsymbol{x}_k\|$, we have $\lambda_{\min}(\boldsymbol{K}) = O(\delta')$.

- In Theorem 8 we illustrate how our results for shallow networks can be extended to cover depth-$L$ networks. In particular, if the layer widths satisfy a pyramidal condition, meaning $d_l \geq d_{l+1}$ for $l \in \{1, \cdots, L-1\}$, $d_{L-1} \gtrsim 2^L \log(nL/\epsilon)$ and $d_1 = \tilde{\Omega}(n d_0^3 \delta^{-4})$, then $\lambda_{\min}(\boldsymbol{K}) = \tilde{\Omega}(d_0^{-3} \delta^4)$ and $\lambda_{\min}(\boldsymbol{K}) = O(L)$ with high probability.

- Our results allow us to analyze the smallest eigenvalue of the NTK for data drawn from any distribution for which one can establish $\delta$-separation with high probability in terms of $d_0$ and $n$. For example, for shallow networks with data drawn uniformly from a sphere, in Corollary 2 we show that if $d_0 d_1 = \tilde{\Omega}(n^{1+4/(d_0-1)})$, then with high probability $\lambda_{\min}(\boldsymbol{K}) = \tilde{O}\left(n^{-2/(d_0-1)}\right)$ and $\lambda_{\min}(\boldsymbol{K}) = \tilde{\Omega}\left(n^{-4/(d_0-1)}\right)$. Moreover, this bound is tight up to logarithmic factors for $d_0 = \Omega(\log(n))$ matching prior findings for this regime.

The rest of this paper is structured as follows: in Section 2 we provide a summary of related works and compare and contrast our results with the existing state of the art; in Section 3 we present our results for shallow networks; finally in Section 4 we extend our shallow results to the deep case.

**Notations.** With regard to general points on notation we let $[n] = \{1, 2, \cdots, n\}$ denote the set of the first $n$ positive integers. If $\boldsymbol{x} \in \mathbb{R}^d$ then we let $[\boldsymbol{x}]_i$ denote the $i$th entry of $\boldsymbol{x}$. If $f$ and $g$ are real-valued functions, we write $f \lesssim g$ or $f = O(g)$ when there exists an absolute constant $C$ such that $f(x) \leq Cg(x)$ for all $x$. Similarly, we write $f \gtrsim g$ or $f = \Omega(g)$ when there exists a constant $c$ such that $f(x) \geq cg(x)$ for all $x$. We write $f \asymp g$ when $f \lesssim g$ and $f \gtrsim g$ both hold. The notation $\tilde{\Omega}$ hides logarithmic factors. Logarithms are generally considered to be in base $e$, though in most settings the particular choice of base can be absorbed by a constant.

## 2 Related work

**Prior work on the NTK.** Jacot et al. (2018) highlight that the optimization dynamics of neural networks are controlled by the Gram matrix of the Jacobian of the network function, an object referred to as the NTK Gram matrix, or, as we refer to it here, simply the NTK. That work also shows that

in the infinite-width limit the NTK converges in probability to a deterministic kernel. Of particular interest is the observation that in the infinite-width setting the network behaves like a linear model (Lee et al., 2019). Further, if a network is polynomially wide in the number of samples then the smallest eigenvalue of the NTK can be lower bounded in terms of the smallest eigenvalue of its infinite-width analog. As a result, assuming the latter is positive, global convergence guarantees for gradient descent can be obtained (Du et al., 2019a,b; Allen-Zhu et al., 2019; Zou & Gu, 2019; Lee et al., 2019; Oymak & Soltanolkotabi, 2020; Zou et al., 2020; Nguyen & Mondelli, 2020; Nguyen, 2021; Banerjee et al., 2023). The positive definiteness of the NTK is equivalent to the Jacobian having full rank, which can also be used to study the loss landscape (Liu et al., 2020, 2022; Karhadkar et al., 2023). Beyond the smallest eigenvalue, there is interest in characterizing the full spectrum of the NTK (Basri et al., 2019; Geifman et al., 2020; Fan & Wang, 2020; Bietti & Bach, 2021; Murray et al., 2023), which has implications on the dynamics of the empirical risk (Arora et al., 2019b; Velikanov & Yarotsky, 2021) as well as the generalization error (Cao et al., 2021; Basri et al., 2020; Cui et al., 2021; Jin et al., 2022; Bowman & Montúfar, 2022). Finally, although a powerful and successful tool for analyzing neural networks it must be noted that the NTK has limitations, most notably perhaps that it struggles to explain the rich feature learning commonly observed in practice (Lee et al., 2020a; Chizat et al., 2019; Liu et al., 2020).

**Prior work on the smallest eigenvalue of the NTK.**   Many of the prior works discussed so far assume or prove that $\lambda_{\min}(\boldsymbol{K})$ is positive, but do not provide a quantitative lower bound. Here we discuss works seeking to address this issue and to which we view our work as complementary. For shallow ReLU networks and data drawn uniformly from the sphere, Xie et al. (2017, Theorem 3) and Montanari & Zhong (2022, Theorem 3.2) provide lower bounds on the smallest singular and eigenvalue value of the Jacobian and NTK respectively. In addition to requiring the data to be drawn uniform from the sphere both of these results are high dimensional in the sense that for Xie et al. (2017, Theorem 3) to be non-vacuous it is necessary that $d_0 = \Omega(d_1 n^2)$, while Montanari & Zhong (2022, Theorem 3.2) requires, as per their Assumption 3.1, that $d_0 = \tilde{\Omega}(\sqrt{n})$.

Nguyen et al. (2021, Theorem 4.1) derives lower and upper bounds for the smallest eigenvalue of the NTK for deep ReLU networks under standard initialization conditions assuming the data is drawn from a distribution satisfying a Lipschitz concentration property. They show that the NTK is well conditioned if the network has a layer of width of order equal to the number of data points $n$ up to logarithmic factors. Concretely, if at least one layer has width linear in $n$ (ignoring logarithmic factors) and the others are at least poly-logarithmic in $n$, then $\lambda_{\min}(\boldsymbol{K}) = \Omega(\mu_r^2(\sigma)d_0)$ (or $\Omega(\mu_r^2(\sigma))$ with normalized data), where $\mu_r(\sigma)$ denotes the $r$th Hermite coefficient of $\sigma$ with any even integer $r \geq 2$. However, in their result the bound holds with high probability only if $d_0$ scales as $\log(n)$.

Bombari et al. (2022, Theorem 1) derive lower and upper bounds for the smallest eigenvalue of the NTK under similar conditions as Nguyen et al. (2021, Theorem 4.1) aside from the following: they consider smooth rather than ReLU activation functions, the widths follow a loose pyramidal topology, meaning $d_l = O(d_{l-1})$ for all $l \in [L-1]$, $d_{L-1}d_{L-2}$ scales linearly in $n$ (ignoring logarithmic factors), and there exists a $\gamma > 0$ such that $n^\gamma = O(d_{L-1})$. Under these conditions they show that $\lambda_{\min}(\boldsymbol{K}) = \Omega(d_{L-1}d_{L-2})$ with high probability as both $d_{L-1}$ and $n$ grow. This result illustrates that for the NTK to be well conditioned it suffices that the number of neurons grows as $\tilde{\Omega}(\sqrt{n})$. The loose pyramidal condition on the widths implies $d_{L-1}d_{L-2} = O(d_0^2)$ and as they also assume that $n = o(d_{L-1}d_{L-2})$ then $n = o(d_0^2)$ which in turn implies $d_0 = \Omega(\sqrt{n})$.

The rough strategy used by both Bombari et al. (2022) and Nguyen et al. (2021), as well as in our own results, can be described in terms of two main steps. In the first step, one bounds the smallest eigenvalue of a shallow network. The results for the shallow case can then be extended to the deep case, e.g., via a layerwise decomposition of the NTK matrix. This second step is architecture-dependent and its proof depends on the bounds derived in the first step. Our results focus on improving the first step which imply corresponding improvements for the second step.

## 3   Shallow networks

Here we study the smallest eigenvalue of the NTK of a shallow neural network. The parameter space $\mathcal{P}$ of this network is $\mathbb{R}^{d_1 \times d_0} \times \mathbb{R}^{d_1}$ and it is equipped with the inner product

$$\langle (\boldsymbol{W}, \boldsymbol{v}), (\boldsymbol{W}', \boldsymbol{v}') \rangle = \text{Trace}(\boldsymbol{W}^T \boldsymbol{W}') + \boldsymbol{v}^T \boldsymbol{v}'.$$

For convenience we sometimes write $d = d_0$. The neural network $f : \mathbb{R}^{d_0} \times \mathcal{P} \to \mathbb{R}$ is defined as

$$f(\boldsymbol{x}; \boldsymbol{W}, \boldsymbol{v}) = \frac{1}{\sqrt{d_1}} \sum_{j=1}^{d_1} v_j \sigma(\boldsymbol{w}_j^T \boldsymbol{x}), \tag{2}$$

where $\boldsymbol{W} = [\boldsymbol{w}_1, \cdots, \boldsymbol{w}_{d_1}]^T \in \mathbb{R}^{d_1 \times d_0}$ are the inner layer weights, $\boldsymbol{v} = [v_1, \cdots, v_{d_1}]^T \in \mathbb{R}^{d_1}$ the outer layer weights, and $\boldsymbol{\theta} = (\boldsymbol{W}, \boldsymbol{v})$. We consider the ReLU activation function applied entrywise with $\sigma(z) = \max\{0, z\}$. The derivative $\dot{\sigma}$ satisfies $\dot{\sigma}(z) = 1$ for $z > 0$ and $\dot{\sigma}(z) = 0$ for $z < 0$. Although $\sigma$ is not differentiable at 0, we take $\dot{\sigma}(0) = 0$ by convention. Unless otherwise stated we assume that the entries of $\boldsymbol{W}$ and $\boldsymbol{v}$ are drawn mutually iid from a standard Gaussian distribution $\mathcal{N}(0, 1)$. Our main result for shallow networks is the following theorem.

**Theorem 1.** *Let $d \geq 3$, $\epsilon \in (0, 1)$, and $\delta, \delta' \in (0, \sqrt{2})$. Suppose that $\boldsymbol{x}_1, \cdots, \boldsymbol{x}_n \in \mathbb{S}^{d-1}$ are $\delta$-separated and $\min_{i \neq k} \|\boldsymbol{x}_i - \boldsymbol{x}_k\| \leq \delta'$. Define*

$$\lambda = \left( 1 + \frac{d \log(1/\delta)}{\log(d)} \right)^{-3} \delta^2.$$

*If $d_1 \gtrsim \frac{\|\boldsymbol{X}\|^2}{\lambda} \log \frac{n}{\epsilon}$, then with probability at least $1 - \epsilon$,*

$$\lambda \lesssim \lambda_{\min}(\boldsymbol{K}) \lesssim \delta'.$$

A proof of Theorem 1 is provided in Appendix C.7. Suppressing logarithmic factors, Theorem 1 implies that $d_1 = \tilde{\Omega}\left( \|\boldsymbol{X}\|^2 d_0^3 \delta^{-2} \right)$ suffices to ensure that $\lambda_{\min}(\boldsymbol{K}) = \tilde{\Omega}(d_0^{-3} \delta^2)$ and $\lambda_{\min}(\boldsymbol{K}) = O(\delta')$ with high probability (note the trivial bound $\|\boldsymbol{X}\|^2 \leq \|\boldsymbol{X}\|_F^2 \leq n$). We emphasize that unlike existing results i) we make no distributional assumptions on the data, instead only assuming a milder $\delta$-separated condition, and ii) our bounds hold with high probability even if $d_0$ is held constant.

A few further remarks are in order. First, the condition $d_0 \geq 3$ is necessary because our technique relies on the addition formula for spherical harmonics (Efthimiou & Frye, 2014, Theorem 4.11); the bound we derive based on this formula (Lemma 15 in Appendix A.2) becomes vacuous for $d_0 < 3$. However, for $d_0 = 2$ analogous bounds could be derived using more elementary tools while the case $d_0 = 1$ is of little interest as only a trivial dataset is possible. Moreover, data in $\mathbb{S}^1$ could be embedded in $\mathbb{S}^2$ since we do not impose any distributional assumptions.

Second, one can use Theorem 1 to bound the smallest eigenvalue of the NTK for data drawn from the uniform distribution on the sphere by bounding $\delta$ with high probability in terms of $n$ and $d$. We use that $\delta = \Omega(n^{-2/d_0})$ and $\delta' = O(n^{-2/d_0})$ with high probability. We direct the interested reader to Appendix C.8 for further details.

**Corollary 2.** *Let $d \geq 3$, $n \geq 2$, $\epsilon \in (0, 1)$, $\boldsymbol{x}_1, \cdots, \boldsymbol{x}_n \sim U(\mathbb{S}^{d-1})$ be mutually iid. Define*

$$\lambda = \left( 1 + \frac{\log(n/\epsilon)}{\log(d)} \right)^{-3} \left( \frac{\epsilon^2}{n^4} \right)^{1/(d-1)}.$$

*If $d_1 \gtrsim \frac{1}{\lambda} \left( 1 + \frac{n + \log(1/\epsilon)}{d} \right) \log \frac{n}{\epsilon}$, then with probability at least $1 - \epsilon$ over the data and network parameters,*

$$\lambda \lesssim \lambda_{\min}(\boldsymbol{K}) \lesssim \left( \frac{\log(1/\epsilon)}{n^2} \right)^{1/(d-1)}.$$

The above corollary implies that if $d_0 d_1 = \tilde{\Omega}\left( n^{1+4/(d_0-1)} \right)$, then with high probability $\lambda_{\min}(\boldsymbol{K}) = \tilde{\Omega}(n^{-4/(d_0-1)})$ and $\lambda_{\min}(\boldsymbol{K}) = \tilde{O}(n^{-2/(d_0-1)})$. In particular, for data sampled uniformly from a sphere, the scaling $d_0 = \Omega(\log n)$ is both necessary and sufficient for $\lambda_{\min}(\boldsymbol{K})$ to be $\tilde{\Theta}(1)$. In particular the bounds are sharp in this case.

### 3.1 Proof outline for Theorem 1

Recall the definitions of $F(\boldsymbol{\theta})$ and $\boldsymbol{K}$ in (1). For the choice of $f$ given in (2), a straightforward decomposition of the NTK with respect to the inner and outer weights gives

$$\boldsymbol{K} = \boldsymbol{K}_1 + \boldsymbol{K}_2, \tag{3}$$

where $K_1 = \nabla_W F(\theta)^* \nabla_W F(\theta)$ and $K_2 = \nabla_v F(\theta)^* \nabla_v F(\theta) = \frac{1}{d_1}\sigma(WX)^T \sigma(WX)$. As both $K_1$ and $K_2$ are positive semi-definite,

$$\lambda_{\min}(K) \geq \lambda_{\min}(K_1) + \lambda_{\min}(K_2); \tag{4}$$

see, e.g., Horn & Johnson (2012, Theorem 4.3.1). Our proof now follows the highlighted steps below.

**1) Bound the smallest eigenvalue in terms of the infinite-width limit.**   We proceed to bound both $\lambda_{\min}(K_1)$ and $\lambda_{\min}(K_2)$ in terms of the smallest eigenvalues of their infinite-width counterparts, see Lemmas 3 and 4 below, which act as good approximations for sufficiently wide networks.

**Lemma 3.** *Suppose that $x_1, \cdots, x_n \in \mathbb{S}^{d-1}$. Let*

$$\lambda_1 = \lambda_{\min}\left(\mathbb{E}_{u \sim U(\mathbb{S}^{d-1})}\left[\dot{\sigma}\left(X^T u\right)\dot{\sigma}\left(u^T X\right)\right]\right).$$

*If $\lambda_1 > 0$ and $d_1 \gtrsim \lambda_1^{-1}\|X\|^2 \log\frac{n}{\epsilon}$, then with probability at least $1 - \epsilon$*

$$\lambda_{\min}(K_1) \gtrsim \lambda_1.$$

**Lemma 4.** *Suppose that $x_1, \cdots, x_n \in \mathbb{S}^{d-1}$. Let*

$$\lambda_2 = d\lambda_{\min}\left(\mathbb{E}_{u \sim U(\mathbb{S}^{d-1})}\left[\sigma(X^T u)\sigma(u^T X)\right]\right).$$

*If $\lambda_2 > 0$ and $d_1 \gtrsim \frac{n}{\lambda_2}\log\left(\frac{n}{\lambda_2}\right)\log\left(\frac{n}{\epsilon}\right)$, then with probability at least $1 - \epsilon$*

$$\lambda_{\min}(K_2) \gtrsim \lambda_2.$$

We prove Lemmas 3 and 4 in Appendices C.1 and C.2 respectively. Observe that while the parameters of the model are initialized as Gaussian, the expectations above are taken with respect to the uniform measure on the sphere. The motivation for using the uniform measure on the sphere is that it enables us to work with spherical harmonics, for which there is the highly useful *addition formula* (see, e.g., Efthimiou & Frye, 2014, Theorem 4.11). The exchange of measures is possible in the case of Lemma 3 due to the scale invariance of $\dot{\sigma}$, while for Lemma 4 it is possible because $\sigma$ is homogeneous.

**2) Interpret the infinite-width kernel in terms of a hemisphere transform.**   Next, for a given $X$ and $\psi \in \{\sqrt{d}\sigma, \dot{\sigma}\}$ we define the limiting NTK $K_\psi^\infty \in \mathbb{R}^{n \times n}$ as

$$K_\psi^\infty = \mathbb{E}_{u \sim U(\mathbb{S}^{d-1})}\left[\psi\left(X^T u\right)\psi\left(u^T X\right)\right]. \tag{5}$$

Consider a fixed vector $z \in \mathbb{S}^{n-1}$ and interpret the Euclidean inner product $\langle\psi(X^T u), z\rangle$ as a function of $u \in \mathbb{S}^{d-1}$. It will prove useful to think of this map as an integral transform. To this end let $\mathcal{M}(\mathbb{S}^{d-1})$ denote the vector space of signed Radon measures on $\mathbb{S}^{d-1}$ and fix $\psi \in \{\sqrt{d}\sigma, \dot{\sigma}\}$. For a signed Radon measure $\mu \in \mathcal{M}(\mathbb{S}^{d-1})$ we introduce the integral transform $T_\psi\mu : \mathbb{S}^{d-1} \to \mathbb{R}$, defined as

$$(T_\psi\mu)(u) = \int_{\mathbb{S}^{d-1}} \psi(\langle u, x\rangle)d\mu(x). \tag{6}$$

Note for $\psi \in \{\sqrt{d}\sigma, \dot{\sigma}\}$ this is a *hemisphere transform* (Rubin, 1999) as the integrand $\psi(\langle u, \cdot\rangle)$ is supported on a hemisphere normal to $u$. We provide background material on the hemisphere transform in Appendix B. Let $\mathcal{M}_X \subset \mathcal{M}$ denote the space of signed Radon measures supported on the data set $\{x_1, \cdots, x_n\}$. For each measure $\mu \in \mathcal{M}_X$ there exists a vector $z \in \mathbb{R}^n$ such that $\mu = \sum_{i=1}^n z_i\delta_{x_i}$, where $\delta_x$ is the Dirac measure supported on $x$. We write $\mu = \mu_z$ to indicate this correspondence. The following lemma relates the smallest eigenvalue of $K_\psi^\infty$ to the norm of the hemisphere transform of a measure supported on the data; a proof is provided in Appendix C.3.

**Lemma 5.** *Fix $X \in \mathbb{R}^{d \times n}$ and $\psi \in \{\sqrt{d}\sigma, \dot{\sigma}\}$. For all $z \in \mathbb{R}^n$, $\langle K_\psi^\infty z, z\rangle = \|T_\psi\mu_z\|^2$. Moreover,*

$$\lambda_{\min}(K_\psi^\infty) = \inf_{\|z\|=1} \|T_\psi\mu_z\|^2.$$

**3) Bound the hemisphere transform norm via spherical harmonics.** We proceed to lower bound $\|T_\psi \mu_z\|^2$ for all $z \in \mathbb{R}^d$. Let $L^2(\mathbb{S}^{d-1})$ denote the Hilbert space of real-valued, square-integrable functions with respect to the uniform probability measure on $\mathbb{S}^{d-1}$, and let $\mathcal{C}(\mathbb{S}^{d-1}) \subset L^2(\mathbb{S}^{d-1})$ denote the subspace of continuous functions. For $\mu \in \mathcal{M}(\mathbb{S}^{d-1})$ and $g \in \mathcal{C}(\mathbb{S}^{d-1})$ we define

$$\langle \mu, g \rangle := \int_{\mathbb{S}^{d-1}} g(\boldsymbol{x}) d\mu(\boldsymbol{x}).$$

If $g_1, \cdots, g_N \in L^2(\mathbb{S}^{d-1})$ are orthonormal, in particular consider $g_r$ as spherical harmonics, then via a Bessel inequality

$$\|T_\psi \mu_z\|^2 \geq \sum_{a=1}^N |\langle T_\psi \mu_z, g_a \rangle|^2 = \sum_{a=1}^N |\langle \mu_z, T_\psi g_a \rangle|^2 = \sum_{a=1}^N \left| \sum_{i=1}^n (T_\psi g_a)(\boldsymbol{x}_i) z_i \right|^2.$$

Importantly, $T_\psi$ is self-adjoint (see Lemma 17 in Appendix B for details) and the spherical harmonics are eigenfunctions of $T_\psi$, i.e., $T_\psi g_a = \kappa_a g_a$. A summary of the key properties of spherical harmonics needed for our results are provided in Appendix A.2. Therefore

$$\|T_\psi \mu_z\|^2 \geq \sum_{a=1}^N \left| \sum_{i=1}^n (T_\psi g_a)(\boldsymbol{x}_i) z_i \right|^2 = \sum_{a=1}^N \kappa_a^2 \left| \sum_{i=1}^n g_a(\boldsymbol{x}_i) z_i \right|^2 \geq \min_a \kappa_a^2 \|\boldsymbol{D}z\|_2^2,$$

where $\boldsymbol{D} \in \mathbb{R}^{N \times n}$ is a matrix with entries $[\boldsymbol{D}]_{ai} = g_a(\boldsymbol{x}_i)$. As a result

$$\lambda_{\min}(\boldsymbol{K}_\psi^\infty) \geq \min_a \kappa_a^2 \sigma_{\min}^2(\boldsymbol{D}).$$

**4) Bound the hemisphere transform and spherical harmonics on the data.** The following result shows that if we let the functions $(g_a)_{a \in [N]}$ be spherical harmonics and allow $N$ to be sufficiently large, then we can bound the minimum singular value of $\boldsymbol{D}$. In what follows let $\mathcal{H}_r^d$ denote the vector space of degree-$r$ harmonic homogeneous polynomials on $d$ variables.

**Lemma 6.** *Suppose $\boldsymbol{x}_1, \cdots, \boldsymbol{x}_n \in \mathbb{S}^{d-1}$ are $\delta$-separated. Suppose that $\beta \in \{0, 1\}$ and that $R \in \mathbb{Z}_{\geq 0}$ are such that $N := \sum_{r=0}^R \dim(\mathcal{H}_{2r+\beta}^d)$ satisfies $N \geq C \left( \frac{\delta^4}{2} \right)^{-(d-2)/2}$ where $C > 0$ is a universal constant. Let $g_1, \cdots, g_N$ be spherical harmonics which form an orthonormal basis of $\bigoplus_{r=0}^R \mathcal{H}_{2r+\beta}^d$. If $\boldsymbol{D} \in \mathbb{R}^{N \times n}$ is defined as $\boldsymbol{D}_{ai} = g_a(\boldsymbol{x}_i)$ then $\sigma_{\min}(\boldsymbol{D}) \geq \sqrt{\frac{N}{2}}$.*

A proof of Lemma 6 can be found in Appendix C.4. By carefully choosing values for $R$ and $N$ in Lemma 6 and performing some asymptotics on the resulting expressions, we arrive at the following bound on the hemisphere transform of a measure.

**Lemma 7.** *Let $d \geq 3$ and suppose that $\boldsymbol{x}_1, \cdots, \boldsymbol{x}_n \in \mathbb{S}^{d-1}$ are $\delta$-separated. For all $z \in \mathbb{R}^n$ with $\|z\| \leq 1$ then*

$$\|T_\psi \mu_z\|^2 \gtrsim \begin{cases} \left( 1 + \frac{d \log(1/\delta)}{\log d} \right)^{-3} \delta^2 & \text{if } \psi = \dot{\sigma} \\ \left( 1 + \frac{d \log(1/\delta)}{\log d} \right)^{-3} \delta^4 & \text{if } \psi = \sqrt{d}\sigma. \end{cases}$$

A proof of Lemma 7 is provided in Appendix C.5. The lower bound of Theorem 1 follows by bounding $\lambda_1$, as defined in Lemma 3, using Lemma 7.

Before proceeding to the upper bound, we pause to remark on the generality of this argument for handling other activation functions. First, we use the positive homogeneity of the activation function in order to write $\lambda_{\min}(\boldsymbol{K}_\psi^\infty)$ as the $L^2(\mathbb{S}^{d-1})$ norm of a function on the sphere. This is beneficial as it allows us to work with the spherical harmonics and use the associated addition formula. The ReLU activation and its derivative are also convenient with regard to computing the eigenvalues of the hemisphere transform (or more generally the eigenvalues of the integral operator). In particular, this requires evaluating integrals against Gegenbauer polynomials for which analytic expressions are available. For polynomial or piecewise polynomial activations similar results could be obtained. However, for other activations, e.g., tanh or sigmoid, such quantities appear challenging to compute.

**5) Upper bound.** The upper bound of Theorem 1 is simpler than the lower bound and hinges on the following calculation. Let $\boldsymbol{x}_i, \boldsymbol{x}_k$ be two data points. Then

$$\lambda_{\min}(\boldsymbol{K}) \le \frac{1}{2}(\boldsymbol{e}_i - \boldsymbol{e}_k)^T \boldsymbol{K}(\boldsymbol{e}_i - \boldsymbol{e}_k) = \frac{1}{2}\|\nabla_{\boldsymbol{\theta}} f(\boldsymbol{x}_i) - \nabla_{\boldsymbol{\theta}} f(\boldsymbol{x}_k)\|^2.$$

Therefore it suffices to upper bound the norm of $\nabla_{\boldsymbol{\theta}} f(\boldsymbol{x}_i) - \nabla_{\boldsymbol{\theta}} f(\boldsymbol{x}_k)$. We choose $i, k \in [n]$ such that $\boldsymbol{x}_i, \boldsymbol{x}_k$ are the two closest points in the dataset. We then translate this into a statement about the gradients. If $\|\boldsymbol{x}_i - \boldsymbol{x}_k\| \le \delta$, then with high probability over the network parameters, $\|\nabla_{\boldsymbol{\theta}} f(\boldsymbol{x}_i) - \nabla_{\boldsymbol{\theta}} f(\boldsymbol{x}_k)\|^2 \lesssim \delta$ (see Lemma 29), and we arrive at the desired upper bound in Theorem 1.

# 4   From shallow to deep neural networks

Our goal here is to detail just one approach as how the results of Section 3 can be extended to deep networks. To be clear, here we consider a fully connected network with input dimension $d_0$ and $L$ layers, where each layer has width $d_1, \cdots, d_L$ respectively and $d_L = 1$. The parameter space $\mathcal{P}$ is a product space of matrices $\prod_{l=1}^{L} \mathbb{R}^{d_l \times d_{l-1}}$, equipped with the inner product

$$\langle (\boldsymbol{W}_1, \cdots, \boldsymbol{W}_L), (\boldsymbol{W}_1', \cdots, \boldsymbol{W}_L') \rangle = \sum_{l=1}^{L} \text{Trace}(\boldsymbol{W}_l^T \boldsymbol{W}_l').$$

The feature maps $f_l : \mathbb{R}^{d_0} \times \mathcal{P} \to \mathbb{R}^{d_l}$ of the neural network are given by

$$f_l(\boldsymbol{x}; \boldsymbol{\theta}) = \begin{cases} \boldsymbol{x} & l = 0 \\ \sigma(\boldsymbol{W}_l f_{l-1}(\boldsymbol{x}; \boldsymbol{\theta})) & l \in [L-1] \\ \boldsymbol{W}_l f_{l-1}(\boldsymbol{x}; \boldsymbol{\theta}) & l = L, \end{cases}$$

where $\boldsymbol{W}_l \in \mathbb{R}^{d_l \times d_{l-1}}$ for all $l \in [L]$, $\boldsymbol{\theta} = (\boldsymbol{W}_1, \cdots, \boldsymbol{W}_L)$ and $\sigma$ is the ReLU function $x \mapsto \max(0, x)$ applied elementwise. We define the network map $f$ to be the final feature map multiplied by a normalizing constant:

$$f = \left( \prod_{l=1}^{L-1} \sqrt{\frac{2}{d_l}} \right) f_L. \tag{7}$$

Given $n$ data points $\boldsymbol{x}_1, \cdots, \boldsymbol{x}_n$, we bound the smallest eigenvalue of the NTK (1) associated with this particular choice of $f$.

**Theorem 8.** *Suppose $\epsilon \in (0, 1/3)$, $\delta \in (0, \sqrt{2}]$, $d_0 \ge 3$, the data $\boldsymbol{x}_1, \boldsymbol{x}_2, \cdots, \boldsymbol{x}_n \in \mathbb{S}^{d_0-1}$ is $\delta$-separated and define*

$$\lambda = \left( 1 + \frac{d_0 \log(1/\delta)}{\log d_0} \right)^{-3} \delta^4.$$

*With regard to the network architecture, let $L \ge 3$, $d_l \ge d_{l+1}$ for all $l \in [L-1]$, $d_{L-1} \gtrsim 2^L \log\left(\frac{nL}{\epsilon}\right)$ and $d_1 \gtrsim \frac{n}{\lambda} \log\left(\frac{n}{\lambda}\right) \log\left(\frac{n}{\epsilon}\right)$. Then with probability at least $1 - \epsilon$ over the network parameters*

$$\lambda \lesssim \lambda_{\min}(\boldsymbol{K}) \lesssim L.$$

We emphasize that these bounds make no distributional assumptions on the data other than lying on the sphere and being $\delta$-separated; in particular, they hold even for constant $d_0$. Indeed, if we consider $d_0$ as some constant then Theorem 8 implies that if the first layer is sufficiently wide, $d_1 = \tilde{\Omega}(n\delta^{-4})$, then with high probability over the parameters $\lambda_{\min}(\boldsymbol{K}) = \tilde{\Omega}(\delta^4)$ and $\lambda_{\min}(\boldsymbol{K}) = O(1)$.

A few remarks are in order. First, the pyramidal condition on the network widths could be relaxed by more directly borrowing techniques from Nguyen et al. (2021). We adopt this condition as it has the advantage of making the dependence of our bounds on the network depth $L$ clearer. Second, compared with Theorem 1 and ignoring log factors, we observe the lower bound differs by a factor of $\delta^2$. This arises as a result of the smallest eigenvalue of the feature Gram matrix $\boldsymbol{F}_1^T \boldsymbol{F}_1$ being equivalent to the Jacobian of a shallow network with respect to the second layer weights, not the inner layer weights, which has a different lower bound as per Lemma 7. For reasons apparent in the proof outline below the lower bound on $\lambda_{\min}(\boldsymbol{K})$ lacks a dependency on $L$, however we hypothesize it should also grow linearly with $L$ thereby matching the dependency of the upper bound. Finally, the upper bound itself follows a similar approach as used by Nguyen et al. (2021) and is weak in the sense that we cannot take advantage of the dataset separation for gradients deeper into the network. We remark that this is also a common problem in the prior work of Nguyen et al. (2021) and Bombari et al. (2022), we refer the reader to the proof outline below for further details.

### 4.1 Proof outline for Theorem 8

The proof of the deep case is structured around the decomposition of the NTK provided in Lemma 9 below. To state this decomposition we introduce the following quantities. For $l \in [L-1]$ we define the feature matrices $\boldsymbol{F}_l \in \mathbb{R}^{d_l \times n}$ by

$$\boldsymbol{F}_l = [f_l(\boldsymbol{x}_1), \cdots, f_l(\boldsymbol{x}_n)].$$

For $l \in [L-1]$ and $\boldsymbol{x} \in \mathbb{R}^d$ we define the activation patterns $\boldsymbol{\Sigma}_l(\boldsymbol{x}) \in \{0,1\}^{d_l \times d_l}$ to be the diagonal matrices

$$\boldsymbol{\Sigma}_l(\boldsymbol{x}) = \mathrm{diag}(\dot{\sigma}(\boldsymbol{W}_l f_{l-1}(\boldsymbol{x}))).$$

Finally, we let $\boldsymbol{1}_n$ denote the vector of all ones in $\mathbb{R}^n$.

**Lemma 9.** *Let $\boldsymbol{x}_1, \cdots, \boldsymbol{x}_n \in \mathbb{R}^d$ be nonzero. There exists an open set $\mathcal{U} \subset \mathcal{P}$ of full Lebesgue measure such that $f(\boldsymbol{x}_i; \cdot)$ is continuously differentiable on $\mathcal{U}$ for all $i \in [n]$. Moreover, for all $\boldsymbol{\theta} \in \mathcal{U}$ the NTK Gram matrix $\boldsymbol{K}$ defined in (1) with network function (7) satisfies*

$$\left( \prod_{l=1}^{L-1} \frac{d_l}{2} \right) \boldsymbol{K} = \sum_{l=0}^{L-1} (\boldsymbol{F}_l^T \boldsymbol{F}_l) \odot (\boldsymbol{B}_{l+1} \boldsymbol{B}_{l+1}^T),$$

*where the $i$th row of $\boldsymbol{B}_l \in \mathbb{R}^{n \times n_l}$ is defined as*

$$[\boldsymbol{B}_l]_{i,:} = \begin{cases} \boldsymbol{\Sigma}_l(\boldsymbol{x}_i) \left( \prod_{k=l+1}^{L-1} \boldsymbol{W}_k^T \boldsymbol{\Sigma}_k(\boldsymbol{x}_i) \right) \boldsymbol{W}_L^T, & l \in [L-1], \\ \boldsymbol{1}_n, & l = L. \end{cases}$$

For completeness we prove Lemma 9 in Appendix D.1. Observe each matrix summand in Lemma 9 is positive semi-definite (PSD) and recall for any two PSD matrices $\boldsymbol{A}$ and $\boldsymbol{B}$ one has $\lambda_{\min}(\boldsymbol{A} + \boldsymbol{B}) \geq \lambda_{\min}(\boldsymbol{A}) + \lambda_{\min}(\boldsymbol{B})$ (see e.g. Horn & Johnson, 2012, Theorem 4.3.1) and $\lambda_{\min}(\boldsymbol{A} \odot \boldsymbol{B}) \geq \lambda_{\min}(\boldsymbol{A}) \min_{i \in [n]} [\boldsymbol{B}]_{ii}$ (Schur, 1911). Therefore

$$\left( \prod_{l=1}^{L-1} \frac{d_l}{2} \right) \lambda_{\min}(\boldsymbol{K}) \geq \sum_{l=0}^{L-1} \lambda_{\min} \left( (\boldsymbol{F}_l^T \boldsymbol{F}_l) \odot (\boldsymbol{B}_{l+1} \boldsymbol{B}_{l+1}^T) \right) \geq \lambda_{\min} \left( \boldsymbol{F}_1^T \boldsymbol{F}_1 \right) \min_{i \in [n]} \| [\boldsymbol{B}_2]_{i,:} \|^2 .$$

In order to upper bound the smallest eigenvalue we follow Nguyen et al. (2021) and analyze the Raleigh quotient $R(\boldsymbol{u}) = \frac{\boldsymbol{u}^T \boldsymbol{K} \boldsymbol{u}}{\|\boldsymbol{u}\|^2}$. In particular, for any nonzero $\boldsymbol{u} \in \mathbb{R}^n$ we have $\lambda_{\min}(\boldsymbol{K}) \leq R(\boldsymbol{u})$ and therefore $\lambda_{\min}(\boldsymbol{K}) \leq R(\boldsymbol{e}_i) = [\boldsymbol{K}]_{ii}$ for all $i \in [n]$. As a result

$$\left( \prod_{l=1}^{L-1} \frac{d_l}{2} \right) \lambda_{\min}(\boldsymbol{K}) \leq \left[ \sum_{l=0}^{L-1} (\boldsymbol{F}_l^T \boldsymbol{F}_l) \odot (\boldsymbol{B}_{l+1} \boldsymbol{B}_{l+1}^T) \right]_{ii} = \sum_{l=0}^{L-1} \| f_l(\boldsymbol{x}_i) \|^2 \| [\boldsymbol{B}_{l+1}]_{i,:} \|^2.$$

Combining the upper and lower bounds we have

$$\lambda_{\min} \left( \boldsymbol{F}_1^T \boldsymbol{F}_1 \right) \min_{i \in [n]} \| [\boldsymbol{B}_2]_{i,:} \|^2 \leq \lambda_{\min}(\boldsymbol{K}) \left( \prod_{l=1}^{L-1} \frac{d_l}{2} \right) \leq \sum_{l=0}^{L-1} \| f_l(\boldsymbol{x}_i) \|^2 \| [\boldsymbol{B}_{l+1}]_{i,:} \|^2, \quad (8)$$

where the right hand side holds for any $i \in [n]$. Based on (8), we proceed first by bounding the norm of the network features. We achieve this via an inductive argument, bounding the norm of the features at one layer with high probability, and then conditioning on this event to bound the norm of the features at the next layer with high probability.

**Lemma 10.** *Let $\boldsymbol{x} \in \mathbb{S}^{d_0 - 1}$, $L \geq 2$ and $l \in [L-1]$. If $d_k \gtrsim l^2 \log(l/\epsilon)$ for all $k \in [l]$, then*

$$e^{-1} \left( \prod_{h=1}^{l} \frac{d_h}{2} \right) \leq \| f_l(\boldsymbol{x}) \|^2 \leq e \left( \prod_{h=1}^{l} \frac{d_h}{2} \right)$$

*holds with probability at least $1 - \epsilon$ over the network parameters.*

A proof of Lemma 10 is provided in Appendix D.2. Next we derive upper and lower bounds on the backpropagation terms $[\boldsymbol{B}_l]_{i,:}$. Our strategy for this is as follows: for $l \in [L-2]$, let $\boldsymbol{S}_l(\boldsymbol{x}) = \boldsymbol{\Sigma}_l(\boldsymbol{x}) \left( \prod_{k=l+1}^{L-1} \boldsymbol{W}_k^T \boldsymbol{\Sigma}_k(\boldsymbol{x}) \right)$ and observe

$$[\boldsymbol{B}_l]_{i,:} = \boldsymbol{S}_l(\boldsymbol{x}_i) \boldsymbol{W}_L^T.$$

Since $\boldsymbol{x}_i \in \mathbb{S}^{d_0-1}$, it is sufficient to lower bound $\|\boldsymbol{S}_l(\boldsymbol{x})\boldsymbol{W}_L^T\|_2^2$ for an arbitrary $\boldsymbol{x} \in \mathbb{S}^{d_0-1}$. As the vector $\boldsymbol{W}_L^T \in \mathbb{R}^{d_{L-1}}$ is distributed as $\boldsymbol{W}_L^T \sim \mathcal{N}(\boldsymbol{0}_{d_{L-1}}, \boldsymbol{I}_{d_{L-1}})$, following Vershynin (2018, Theorem 6.3.2) we have that for any $\boldsymbol{A} \in \mathbb{R}^{d_l \times d_{L-1}}$ and $t \geq 0$

$$\mathbb{P}(|\|\boldsymbol{A}\boldsymbol{W}_L^T\| - \|\boldsymbol{A}\|_F| \geq t) \leq 2\exp\left( -\frac{Ct^2}{\|\boldsymbol{A}\|^2} \right)$$

for some constant $C > 0$. As a result, with $t = \frac{1}{2}\|\boldsymbol{A}\|_F^2$ then

$$\mathbb{P}\left( \frac{1}{4}\|\boldsymbol{A}\|_F^2 \leq \|\boldsymbol{A}\boldsymbol{W}_L^T\|^2 \leq \frac{3}{4}\|\boldsymbol{A}\|_F^2 \right) \geq 1 - \exp\left( -C\frac{\|\boldsymbol{A}\|_F^2}{\|\boldsymbol{A}\|^2} \right).$$

In order to lower bound $\|\boldsymbol{S}_l(\boldsymbol{x})\boldsymbol{W}_L^T\|^2$ with high probability over the parameters it therefore suffices to condition on appropriate bounds for $\|\boldsymbol{S}_l(\boldsymbol{x})\|_F^2$ and $\|\boldsymbol{S}_l(\boldsymbol{x})\|_2^2$. These bounds are provided in Lemmas 34 and 35 in Appendices D.3 and D.4 respectively. With these two lemmas in place we can bound $\|\boldsymbol{S}_l(\boldsymbol{x}_i)\boldsymbol{W}_L^T\|^2$.

**Lemma 11.** *Let $\boldsymbol{x} \in \mathbb{S}^{d_0-1}$, suppose $L \geq 3$, $d_k \geq d_{k+1}$ for all $k \in [L-1]$ and $d_{L-1} \gtrsim 2^L \log\left(\frac{L}{\epsilon}\right)$. Then, for any $l \in [L-1]$, with probability at least $1 - \epsilon$ over the network parameters*

$$\|\boldsymbol{S}_l(\boldsymbol{x})\boldsymbol{W}_L^T\|^2 \asymp 2^{-L+l+1} \prod_{k=l}^{L-1} d_k.$$

By combining Lemma 11 with a union bound we arrive at the following corollary, relevant for the lower bound of (8).

**Corollary 12.** *Let $\boldsymbol{x}_i \in \mathbb{S}^{d_0-1}$ for all $i \in [n]$, $L \geq 3$, $d_l \geq d_{l+1}$ for all $l \in [L-1]$ and $d_{L-1} \gtrsim 2^L \log\left(\frac{nL}{\epsilon}\right)$. Then, for any $l \in [L-1]$, with probability at least $1 - \epsilon$ over the network parameters*

$$\min_{i \in [n]} \|[\boldsymbol{B}_2]_{i,:}\|^2 \gtrsim 2^{-L} \prod_{k=2}^{L-1} d_k.$$

The first-layer feature Gram matrix $\boldsymbol{F}_1^T\boldsymbol{F}_1$ in the deep case is identically distributed to $\boldsymbol{K}_2$ in the two-layer case; see (3) and the related definitions. Therefore we can apply Lemma 4 to lower bound the smallest eigenvalue of $\boldsymbol{F}_1^T\boldsymbol{F}_1$. This, in combination with Corollary 12, yields the lower bound of Theorem 8. The upper bound follows by combining the bound on the feature norms provided by Lemma 10 with the bound on the backpropagation terms given in Lemma 11. A detailed proof of Theorem 8 is provided in Appendix D.6.

## 5 Conclusion

**Summary and implications.** Quantitative bounds on the smallest eigenvalue of the NTK are a critical ingredient for many current analyses of network optimization. Prior works provide bounds which are only applicable for data drawn from particular distributions and for which the input dimension $d_0$ scales appropriately with the number of data samples $n$. This work plugs an important gap in the existing literature by providing bounds for arbitrary datasets on the sphere (including those drawn from any distribution on the sphere) in terms of a measure of distance between data points. Furthermore, these bounds are applicable for any $d_0$, in particular even $d_0$ held constant with respect to $n$.

**Limitations.** Our bounds currently only hold for the ReLU activation function. Another limitation, also present in prior work, is that our upper bound on the smallest eigenvalue of the NTK for deep networks in Theorem 8 does not capture the data separation. Finally, a mild limitation of this work is that we require the data to be normalized so as to lie on the sphere.

**Future work.** The proof techniques developed here could be applied to analyze the NTK in the context of other homogeneous activation functions. One could potentially relax the homogeneity condition on the activation function, or the condition of unit norm data, by considering an integral transform on the space $L^2(\mathbb{R}^d, \mu)$ rather than $L^2(\mathbb{S}^{d-1})$, where $\mu$ denotes the standard Gaussian measure (since the weights are drawn from a Gaussian distribution). Beyond fully connected networks, conducting comparable analyses in the context of other architectures, e.g., CNNs, GNNs, or transformers, would be valuable future work.

### Acknowledgment

This project has been supported by NSF CAREER 2145630, NSF 2212520, DFG 464109215 within SPP 2298 Theoretical Foundations of Deep Learning, and BMBF in DAAD project 57616814.

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

# A  Background material

## A.1  Concentration bounds

In order to bound the smallest eigenvalue of the finite-width NTK in terms of the expected, or infinite width NTK, we use the following matrix Chernoff bound variant.

**Lemma 13.** *Let $R > 0$, and let $\boldsymbol{Z}_1, \cdots, \boldsymbol{Z}_m \in \mathbb{R}^{n \times n}$ be iid symmetric random matrices such that $0 \preceq \boldsymbol{Z}_1 \preceq R\boldsymbol{I}$ almost surely. Then*

$$\mathbb{P}\left(\lambda_{\min}\left(\frac{1}{m}\sum_{j=1}^{m}\boldsymbol{Z}_j\right) \leq \frac{1}{2}\lambda_{\min}\left(\mathbb{E}[\boldsymbol{Z}_1]\right)\right) \leq n\exp\left(-\frac{Cm\lambda_{\min}(\mathbb{E}[\boldsymbol{Z}_1])}{R}\right).$$

*Here $C > 0$ is a universal constant.*

*Proof.* By Theorem 1.1 of Tropp (2012), for all $\delta > 0$

$$\mathbb{P}\left(\lambda_{\min}\left(\frac{1}{m}\sum_{j=1}^{m}\boldsymbol{Z}_j\right) \leq (1-\delta)\lambda_{\min}(\mathbb{E}[\boldsymbol{Z}_1])\right)$$

$$= \mathbb{P}\left(\lambda_{\min}\left(\sum_{j=1}^{m}\boldsymbol{Z}_j\right) \leq (1-\delta)\lambda_{\min}\left(\sum_{j=1}^{m}\mathbb{E}[\boldsymbol{Z}_j]\right)\right)$$

$$\leq n\left(\frac{e^{-\delta}}{(1-\delta)^{1-\delta}}\right)^{\frac{1}{R}\lambda_{\min}\left(\sum_{j=1}^{m}\mathbb{E}[\boldsymbol{Z}_j]\right)}$$

$$= n\left(\frac{e^{-\delta}}{(1-\delta)^{1-\delta}}\right)^{\frac{m}{R}\lambda_{\min}(\mathbb{E}[\boldsymbol{Z}_1])}.$$

Let $\delta = \frac{1}{2}$ and let $C = \frac{1}{2}\log\left(\frac{e}{2}\right) > 0$. Substituting into the above bound, we obtain

$$\mathbb{P}\left(\lambda_{\min}\left(\frac{1}{m}\sum_{j=1}^{m}\boldsymbol{Z}_j\right) \leq \frac{1}{2}\lambda_{\min}(\mathbb{E}[\boldsymbol{Z}_1])\right) \leq n\left(\frac{2}{e}\right)^{\frac{m}{2R}\lambda_{\min}(\mathbb{E}[\boldsymbol{Z}_1])}$$

$$= n\exp\left(-\frac{Cm\lambda_{\min}(\mathbb{E}[\boldsymbol{Z}_1])}{R}\right).$$

$\square$

Some of our NTK bounds will depend on the operator norm of the input data matrix $\boldsymbol{X}$, so it will be helpful to upper bound $\|\boldsymbol{X}\|$ with high probability.

**Lemma 14.** *Let $\epsilon > 0$. Let $\boldsymbol{X} = [\boldsymbol{x}_1, \cdots, \boldsymbol{x}_n] \in \mathbb{R}^{d \times n}$ be a random matrix whose columns are independent and uniformly distributed on $\mathbb{S}^{d-1}$. Then with probability at least $1 - \epsilon$,*

$$\|\boldsymbol{X}\|^2 \lesssim 1 + \frac{n + \log\frac{1}{\epsilon}}{d}.$$

*Proof.* We use a covering argument. Fix $\boldsymbol{u} \in \mathbb{S}^{d-1}$ and $\boldsymbol{v} \in \mathbb{S}^{n-1}$. By Lemma 2.2 of Ball (1997), for each $i \in [n]$ and $t \geq 0$,

$$\mathbb{P}(|\langle \boldsymbol{u}, \boldsymbol{x}_i\rangle| \geq t) \leq 2\exp\left(-\frac{dt^2}{2}\right).$$

In other words $\|\langle \boldsymbol{u}, \boldsymbol{x}_i\rangle\|_{\psi_2} \lesssim \frac{1}{\sqrt{d}}$. Then by Hoeffding's inequality, for all $t \geq 0$

$$\mathbb{P}(|\boldsymbol{u}^T\boldsymbol{X}\boldsymbol{v}| \geq t) = \mathbb{P}\left(\left|\sum_{i=1}^{n}[\boldsymbol{v}]_i\langle \boldsymbol{u}, \boldsymbol{x}_i\rangle\right| \geq t\right)$$

$$\leq 2\exp\left(-C_1 dt^2\right), \tag{9}$$

where $C_1 > 0$ is a constant.

Let $\boldsymbol{u}_1, \cdots, \boldsymbol{u}_M$ be a $\left(\frac{1}{4}\right)$-covering of $\mathbb{S}^{d-1}$. That is, $\boldsymbol{u}_1, \cdots, \boldsymbol{u}_M$ are a set of points in $\mathbb{S}^{d-1}$ such that for all $\boldsymbol{u} \in \mathbb{S}^{d-1}$, there exists $j \in [M]$ such that $\|\boldsymbol{u} - \boldsymbol{u}_j\| \leq \frac{1}{4}$. Since the $\left(\frac{1}{4}\right)$-covering number of $\mathbb{S}^{d-1}$ is at most $12^d$ (see Vershynin, 2018, Corollary 4.2.13), we can take $M \leq 12^d$. Similarly, let $\boldsymbol{u}_1, \cdots, \boldsymbol{u}_N$ be a $\left(\frac{1}{4}\right)$-covering of $\mathbb{S}^{n-1}$ with $N \leq 12^n$. By applying a union bound to (9), we obtain

$$\mathbb{P}(|\boldsymbol{u}_j^T \boldsymbol{X} \boldsymbol{v}_k| \geq t \text{ for some } j \in [M], k \in [N]) \leq 2(12^{d+n}) \exp\left(-C_1 d t^2\right).$$

Hence if

$$t = \sqrt{\frac{(d+n)\log 12 + \log \frac{2}{\epsilon}}{d}},$$

then

$$\mathbb{P}(|\boldsymbol{u}_j^T \boldsymbol{X} \boldsymbol{v}_k| \leq t \text{ for all } j \in [M], k \in [N]) \geq 1 - \epsilon.$$

Let us condition on this event for the rest of the proof. Now suppose that $\boldsymbol{u} \in \mathbb{S}^{d-1}$ and $\boldsymbol{v} \in \mathbb{S}^{n-1}$. By construction there exist $j \in [M]$ and $k \in [N]$ such that $\|\boldsymbol{u} - \boldsymbol{u}_j\| \leq \frac{1}{4}$ and $\|\boldsymbol{v} - \boldsymbol{v}_k\| \leq \frac{1}{4}$. Then

$$\begin{aligned}
|\boldsymbol{u}^T \boldsymbol{X} \boldsymbol{v}| &\leq |\boldsymbol{u}_j^T \boldsymbol{X} \boldsymbol{v}_k| + |(\boldsymbol{u} - \boldsymbol{u}_j)^T \boldsymbol{X} \boldsymbol{v}_k| + |\boldsymbol{u}^T \boldsymbol{X} (\boldsymbol{v} - \boldsymbol{v}_k)| \\
&\leq t + \|\boldsymbol{u} - \boldsymbol{u}_j\| \cdot \|\boldsymbol{v}_k\| \cdot \|\boldsymbol{X}\| + \|\boldsymbol{u}\| \cdot \|\boldsymbol{X}\| \cdot \|\boldsymbol{v} - \boldsymbol{v}_k\| \\
&\leq t + \frac{1}{4}\|\boldsymbol{X}\| + \frac{1}{4}\|\boldsymbol{X}\| \\
&= t + \frac{1}{2}\|\boldsymbol{X}\|.
\end{aligned}$$

Since this holds for all $\boldsymbol{u} \in \mathbb{S}^{d-1}$ and $\boldsymbol{v} \in \mathbb{S}^{n-1}$, we obtain

$$\|\boldsymbol{X}\| \leq t + \frac{1}{2}\|\boldsymbol{X}\|.$$

Rearranging yields

$$\begin{aligned}
\|\boldsymbol{X}\|^2 &\leq 4t^2 \\
&\lesssim 1 + \frac{n + \log \frac{1}{\epsilon}}{d}.
\end{aligned}$$

$\square$

## A.2   Spherical harmonics

Here we review some preliminaries on spherical harmonics necessary for our main results. For further details we refer the reader to Efthimiou & Frye (2014) and Axler et al. (2013, Chapter 5). Let $L^2(\mathbb{S}^{d-1})$ denote the Hilbert space of real-valued, square-integrable functions on the sphere $\mathbb{S}^{d-1}$, equipped with the inner product

$$\langle g, h \rangle = \int_{\mathbb{S}^{d-1}} g(\boldsymbol{x}) h(\boldsymbol{x}) \, dS(\boldsymbol{x}),$$

where $dS$ is the uniform probability measure on $\mathbb{S}^{d-1}$. We let $\mathcal{C}(\mathbb{S}^{d-1}) \subset L^2(\mathbb{S}^{d-1})$ denote the subset of functions which are continuous. We say that a function $g : \mathbb{R}^d \to \mathbb{R}$ is *harmonic* if it is twice continuously differentiable and

$$\sum_{r=1}^d \frac{\partial^2 g}{\partial^2 x_r}(\boldsymbol{x}) = 0$$

for all $\boldsymbol{x} \in \mathbb{S}^{d-1}$. We say that a polynomial $g : \mathbb{R}^d \to \mathbb{R}$ is *homogeneous* if there exists $r \in \mathbb{Z}_{\geq 0}$ such that

$$g(\lambda \boldsymbol{x}) = \lambda^r g(\boldsymbol{x})$$

for all $\lambda \in \mathbb{R}$ and $\boldsymbol{x} \in \mathbb{R}^d$. Let $\mathcal{H}_r^d$ denote the vector space of degree $r$ harmonic homogeneous polynomials on $d$ variables, viewed as functions $\mathbb{S}^{d-1} \to \mathbb{R}$. Each space $\mathcal{H}_r^d$ is a finite-dimensional vector space, with

$$
\begin{aligned}
\dim(\mathcal{H}_r^d) &= \binom{r+d-1}{d-1} - \binom{r+d-3}{d-1} \\
&= \frac{2r+d-2}{r}\binom{r+d-3}{d-2}.
\end{aligned}
$$

For $\nu \geq 0$ and $r \in \mathbb{N}$, we define the *Gegenbauer polynomials* $C_r^\nu$ by

$$
C_r^\nu(t) = \sum_{k=0}^{\lfloor r/2 \rfloor} (-1)^k \frac{\Gamma(r-k+\nu)}{\Gamma(\nu)\Gamma(k+1)\Gamma(r-2k+1)}(2t)^{r-2k}.
$$

There exists an orthonormal basis of $\mathcal{H}_r^d$ consisting of functions $Y_{r,s}^d$, $1 \leq s \leq \dim(\mathcal{H}_r^d)$, known as *spherical harmonics*. The spherical harmonics in $\mathcal{H}_r^d$ satisfy the addition formula

$$
\begin{aligned}
\sum_{s=1}^{\dim(\mathcal{H}_r^d)} Y_{r,s}^d(\boldsymbol{x})Y_{r,s}^d(\boldsymbol{x}') &= \frac{\dim(\mathcal{H}_r^d)C_r^{(d-2)/2}(\langle \boldsymbol{x},\boldsymbol{x}'\rangle)\Gamma(r+1)\Gamma(d-2)}{\Gamma(r+d-2)} \\
&= \frac{(2r+d-2)C_r^{(d-2)/2}(\langle \boldsymbol{x},\boldsymbol{x}'\rangle)}{d-2}
\end{aligned}
\tag{10}
$$

for all $\boldsymbol{x}, \boldsymbol{x}' \in \mathbb{S}^{d-1}$. In particular, from the identity $C_r^\nu(1) = \frac{\Gamma(2\nu+r)}{\Gamma(2\nu)\Gamma(r+1)}$ it follows that

$$
\sum_{s=1}^{\dim(\mathcal{H}_r^d)} |Y_{r,s}^d(\boldsymbol{x})|^2 = \dim(\mathcal{H}_r^d).
$$

We can orthogonally decompose $L^2(\mathbb{S}^{d-1})$ into a direct sum of the spaces of spherical harmonics:

$$
L^2(\mathbb{S}^{d-1}) = \bigoplus_{r=1}^\infty \mathcal{H}_r^d.
$$

That is, the spaces $\mathcal{H}_r^d$ are orthogonal and their linear span is dense in $L^2(\mathbb{S}^{d-1})$.

**Lemma 15.** *Let $\delta > 0$ and suppose that $\boldsymbol{x}, \boldsymbol{x}' \in \mathbb{S}^{d-1}$ satisfy $\|\boldsymbol{x}-\boldsymbol{x}'\|, \|\boldsymbol{x}+\boldsymbol{x}'\| \geq \delta$. If $R \in \mathbb{Z}_{\geq 0}$, and $\beta \in \{0,1\}$, then*

$$
\left| \sum_{r=0}^R \sum_{s=1}^{\dim(\mathcal{H}_{2r+\beta}^d)} Y_{2r+\beta,s}^d(\boldsymbol{x})Y_{2r+\beta,s}^d(\boldsymbol{x}') \right| \lesssim \left( \frac{\|\boldsymbol{x}-\boldsymbol{x}'\|^2}{2} \right)^{-(d-2)/4} \binom{2R+\beta+d-1}{d-1}^{1/2}.
$$

*Proof.* Let us define

$$
P(\boldsymbol{x}, \boldsymbol{x}') := \sum_{r=0}^R \sum_{s=1}^{\dim(\mathcal{H}_{2r+\beta}^d)} Y_{2r+\beta,s}^d(\boldsymbol{x})Y_{2r+\beta,s}^d(\boldsymbol{x}').
$$

By the addition formula (10),

$$
\begin{aligned}
|P(\boldsymbol{x},\boldsymbol{x}')| &= \left| \sum_{r=0}^R \frac{(4r+2\beta+d-2)C_{2r+\beta}^{(d-2)/2}(\langle \boldsymbol{x},\boldsymbol{x}'\rangle)}{d-2} \right| \\
&\lesssim \sum_{r=0}^R \frac{(r+d)|C_{2r+\beta}^{(d-2)/2}(\langle \boldsymbol{x},\boldsymbol{x}'\rangle)|}{d}.
\end{aligned}
\tag{11}
$$

In order to bound the right hand side of the above equation, we will need a bound for the Gegenbauer polynomials $C_{2r+\beta}^{(d-2)/2}$. By Theorem 1 of Nevai et al. (1994) (see also equation 2.8 of Xie et al. 2013), for all $\nu \geq \frac{1}{2}, r \geq 0$, and $t \in [0,1)$,

$$(1-t^2)^\nu C_r^\nu(t)^2 \leq \frac{2e(2+\sqrt{2}\nu)}{\pi} \frac{2^{1-2\nu}\pi}{\Gamma(\nu)^2} \frac{\Gamma(r+2\nu)}{\Gamma(r+1)(r+\nu)}$$

$$\lesssim \frac{\nu\Gamma(r+2\nu)}{2^{2\nu}(r+\nu)\Gamma(\nu)^2\Gamma(r+1)}.$$

Rearranging the above expression yields

$$|C_r^\nu(t)| \lesssim \frac{\nu^{1/2}\Gamma(r+2\nu)^{1/2}}{2^\nu(r+\nu)^{1/2}\Gamma(\nu)\Gamma(r+1)^{1/2}(1-t^2)^{\nu/2}}.$$

We now substitute the above bound into (11):

$$|P(\boldsymbol{x}, \boldsymbol{x}')| \lesssim \sum_{r=0}^R \frac{(r+d)\left(\frac{d-2}{2}\right)^{1/2}\Gamma(2r+\beta+d-2)^{1/2}}{d2^{(d-2)/2}\left(2r+\beta+\frac{d-2}{2}\right)^{1/2}\Gamma\left(\frac{d-2}{2}\right)\Gamma(2r+\beta+1)^{1/2}(1-\langle\boldsymbol{x}, \boldsymbol{x}'\rangle^2)^{(d-2)/4}}$$

$$\lesssim \frac{1}{(1-\langle\boldsymbol{x}, \boldsymbol{x}'\rangle^2)^{(d-2)/4}}\sum_{r=0}^R\left(\frac{r+d}{d}\right)^{1/2}\frac{\Gamma(2r+\beta+d-2)^{1/2}}{2^{(d-2)/2}\Gamma\left(\frac{d-2}{2}\right)\Gamma(2r+\beta+1)^{1/2}}.$$

The expression inside the sum is increasing as a function of $r$, so the above expression is bounded above by

$$\frac{1}{(1-\langle\boldsymbol{x}, \boldsymbol{x}'\rangle^2)^{(d-2)/4}}\left(\frac{R+d}{d}\right)^{1/2}\frac{\Gamma(2R+\beta+d-2)^{1/2}}{2^{(d-2)/2}\Gamma\left(\frac{d-2}{2}\right)\Gamma(2R+\beta+1)^{1/2}}$$

$$\lesssim \frac{1}{d^{1/2}(1-\langle\boldsymbol{x}, \boldsymbol{x}'\rangle^2)^{(d-2)/4}}\frac{\Gamma(2R+\beta+d-1)^{1/2}}{2^{(d-2)/2}\Gamma\left(\frac{d-2}{2}\right)\Gamma(2R+\beta+1)^{1/2}}. \tag{12}$$

By Stirling's approximation,

$$2^{(d-2)/2}\Gamma\left(\frac{d-2}{2}\right) \asymp 2^{(d-2)/2}\left(\frac{d-2}{2}\right)^{(d-3)/2}e^{-(d-2)/2}$$

$$= (d-2)^{(d-3)/2}e^{-(d-2)/2}$$

$$\asymp d^{-1/4}(d-2)^{(d-1.5)/2}e^{-(d-2)/2}$$

$$\asymp d^{-1/4}\Gamma(d-1)^{1/2}.$$

Substituting this into (12) yields

$$|P(\boldsymbol{x}, \boldsymbol{x}')| \leq \frac{1}{d^{1/4}(1-\langle\boldsymbol{x}, \boldsymbol{x}'\rangle^2)^{(d-2)/4}}\frac{\Gamma(2R+\beta+d-1)^{1/2}}{\Gamma(d-1)^{1/2}\Gamma(2R+\beta+1)^{1/2}}$$

$$\asymp \frac{d^{1/4}}{(R+d)^{1/2}(1-\langle\boldsymbol{x}, \boldsymbol{x}'\rangle^2)^{(d-2)/4}}\frac{\Gamma(2R+\beta+d)^{1/2}}{\Gamma(d)\Gamma(2R+\beta+1)^{1/2}}$$

$$= \frac{d^{1/4}}{(R+d)^{1/2}(1-\langle\boldsymbol{x}, \boldsymbol{x}'\rangle^2)^{(d-2)/4}}\left(\frac{2R+\beta+d-1}{d-1}\right)^{1/2}$$

$$\lesssim \frac{1}{(1-\langle\boldsymbol{x}, \boldsymbol{x}'\rangle^2)^{(d-2)/4}}\left(\frac{2R+\beta+d-1}{d-1}\right)^{1/2}$$

Since $\boldsymbol{x}, \boldsymbol{x}' \in \mathbb{S}^{d-1}$,

$$1 - \langle\boldsymbol{x}, \boldsymbol{x}'\rangle^2 = (1 + \langle\boldsymbol{x}, \boldsymbol{x}'\rangle)(1 - \langle\boldsymbol{x}, \boldsymbol{x}'\rangle)$$

$$= \frac{1}{4}\|\boldsymbol{x} + \boldsymbol{x}'\|^2\|\boldsymbol{x} - \boldsymbol{x}'\|^2$$

$$\gtrsim \frac{1}{4}\delta^4.$$

To conclude, we rewrite

$$|P(\boldsymbol{x}, \boldsymbol{x}')| \lesssim \left(\frac{\delta^4}{2}\right)^{-(d-2)/4} \binom{2R + \beta + d - 1}{d - 1}^{1/2}.$$

$\square$

## B  Preliminaries on hemisphere transforms

Let $\mathcal{M}(\mathbb{S}^{d-1})$ denote the vector space of signed Radon measures on $\mathbb{S}^{d-1}$. We denote the total variation of $\mu$ by $|\mu|$. We have a natural inclusion $L^2(\mathbb{S}^{d-1}) \subset \mathcal{M}(\mathbb{S}^{d-1})$ by associating a function $g$ to a signed measure $\mu$ defined by

$$\mu(E) = \int_E g(\boldsymbol{x})dS(\boldsymbol{x}).$$

If $\mu \in \mathcal{M}(\mathbb{S}^{d-1})$ and $g \in \mathcal{C}(\mathbb{S}^{d-1})$, we define the pairing $\langle \mu, g \rangle$ by

$$\langle \mu, g \rangle = \int_{\mathbb{S}^{d-1}} g(\boldsymbol{x})d\mu(\boldsymbol{x}).$$

This agrees with the usual definition of the inner product on $L^2(\mathbb{S}^{d-1})$ when $\mu \in L^2(\mathbb{S}^{d-1})$.

Fix $\psi \in \{\sqrt{d}\sigma, \dot{\sigma}\}$. If $\mu \in \mathcal{M}(\mathbb{S}^{d-1})$, we define its *hemisphere transform* (Rubin, 1999) $T_\psi \mu :$ $\mathbb{S}^{d-1} \to \mathbb{R}$ by

$$(T_\psi \mu)(\boldsymbol{\xi}) = \int_{\mathbb{S}^{d-1}} \psi(\langle \boldsymbol{\xi}, \boldsymbol{x} \rangle)d\mu(\boldsymbol{x}).$$

As is the case with many integral transforms, a hemisphere transform increases the regularity of the functions it is applied to.

**Lemma 16.** *If $\mu \in \mathcal{M}(\mathbb{S}^{d-1})$, then $T_\psi \mu \in L^2(\mathbb{S}^{d-1})$. If $g \in L^2(\mathbb{S}^{d-1})$, then $T_\psi g \in \mathcal{C}(\mathbb{S}^{d-1})$.*

*Proof.* Suppose that $\mu \in \mathcal{M}(\mathbb{S}^{d-1})$. Then

$$
\begin{aligned}
\int_{\mathbb{S}^{d-1}} (T_\psi \mu)(\boldsymbol{\xi})^2 dS(\boldsymbol{\xi}) &= \int_{\mathbb{S}^{d-1}} \left| \int_{\mathbb{S}^{d-1}} \psi(\langle \boldsymbol{\xi}, \boldsymbol{x} \rangle)d\mu(\boldsymbol{x}) \right|^2 dS(\boldsymbol{\xi}) \\
&\leq \int_{\mathbb{S}^{d-1}} \left| \int_{\mathbb{S}^{d-1}} \psi(\langle \boldsymbol{\xi}, \boldsymbol{x} \rangle)d|\mu|(\boldsymbol{x}) \right|^2 dS(\boldsymbol{\xi}) \\
&= \int_{\mathbb{S}^{d-1}} \int_{\mathbb{S}^{d-1}} \int_{\mathbb{S}^{d-1}} \psi(\langle \boldsymbol{\xi}, \boldsymbol{x} \rangle)\psi(\langle \boldsymbol{\xi}, \boldsymbol{x}' \rangle)d|\mu|(\boldsymbol{x})d|\mu|(\boldsymbol{x}')dS(\boldsymbol{\xi}) \\
&\leq \int_{\mathbb{S}^{d-1}} \int_{\mathbb{S}^{d-1}} \int_{\mathbb{S}^{d-1}} d^2 d|\mu|(\boldsymbol{x})d|\mu|(\boldsymbol{x}')dS(\boldsymbol{\xi}) \\
&= |\mu|(\mathbb{S}^{d-1})^2 d^2 \\
&< \infty,
\end{aligned}
$$

so $T\mu \in L^2(\mathbb{S}^{d-1})$.

Now suppose that $g \in L^2(\mathbb{S}^{d-1})$ and $\psi = \dot{\sigma}$. Suppose $\boldsymbol{\xi}, \boldsymbol{\xi}' \in \mathbb{S}^{d-1}$, and observe that

$$dS(\{\boldsymbol{x} \in \mathbb{S}^{d-1} : \langle \boldsymbol{x}, \boldsymbol{\xi} \rangle > 0, \langle \boldsymbol{x}, \boldsymbol{\xi}' \rangle \leq 0\}) = \frac{1}{2\pi} \arccos(\langle \boldsymbol{\xi}, \boldsymbol{\xi}' \rangle).$$

Similarly,

$$dS(\{\boldsymbol{x} \in \mathbb{S}^{d-1} : \langle \boldsymbol{x}, \boldsymbol{\xi} \rangle \leq 0, \langle \boldsymbol{x}, \boldsymbol{\xi}' \rangle > 0\}) = \frac{1}{2\pi} \arccos(\langle \boldsymbol{\xi}, \boldsymbol{\xi}' \rangle),$$

so

$$dS(\{\boldsymbol{x} \in \mathbb{S}^{d-1} : \dot{\sigma}(\langle \boldsymbol{x}, \boldsymbol{\xi} \rangle) \neq \dot{\sigma}(\langle \boldsymbol{x}, \boldsymbol{\xi}' \rangle)\}) = \frac{1}{\pi} \arccos(\langle \boldsymbol{\xi}, \boldsymbol{\xi}' \rangle).$$

We apply this calculation to bound the distance between $T_\psi g(\boldsymbol{\xi})$ and $T_\psi g(\boldsymbol{\xi}')$:

$$|T_\psi g(\boldsymbol{\xi}) - T_\psi g(\boldsymbol{\xi}')| = \left| \int_{\mathbb{S}^{d-1}} \dot{\sigma}(\langle \boldsymbol{x}, \boldsymbol{\xi} \rangle) g(\boldsymbol{x}) dS(\boldsymbol{x}) - \int_{\mathbb{S}^{d-1}} \dot{\sigma}(\langle \boldsymbol{x}, \boldsymbol{\xi}' \rangle) g(\boldsymbol{x}) dS(\boldsymbol{x}) \right|$$

$$\le \int_{\mathbb{S}^{d-1}} |\dot{\sigma}(\langle \boldsymbol{x}, \boldsymbol{\xi} \rangle) - \dot{\sigma}(\langle \boldsymbol{x}, \boldsymbol{\xi}' \rangle)| g(\boldsymbol{x}) dS(\boldsymbol{x})$$

$$\le \|g\|_{L^2} \left( \int_{\mathbb{S}^{d-1}} |\dot{\sigma}(\langle \boldsymbol{x}, \boldsymbol{\xi} \rangle) - \dot{\sigma}(\langle \boldsymbol{x}, \boldsymbol{\xi}' \rangle)|^2 dS(\boldsymbol{x}) \right)^{1/2}$$

$$= \|g\|_{L^2} \left( dS(\{\boldsymbol{x} \in \mathbb{S}^{d-1} : \dot{\sigma}(\langle \boldsymbol{x}, \boldsymbol{\xi} \rangle) \ne \dot{\sigma}(\langle \boldsymbol{x}, \boldsymbol{\xi}' \rangle)\}) \right)^{1/2}$$

$$= \frac{1}{\pi} \|g\|_{L^2} \sqrt{\arccos(\langle \boldsymbol{\xi}, \boldsymbol{\xi}' \rangle)}.$$

Here the third line follows from Cauchy-Schwarz. As $\boldsymbol{\xi} \to \boldsymbol{\xi}'$, $\arccos(\langle \boldsymbol{\xi}, \boldsymbol{\xi}' \rangle) \to 0$ and so $|T_\psi g(\boldsymbol{\xi}) - T_\psi g(\boldsymbol{\xi}')| \to 0$. Therefore, $T_\psi g \in \mathcal{C}(\mathbb{S}^{d-1})$.

Finally suppose that $g \in L^2(\mathbb{S}^{d-1})$ and $\psi = \sqrt{d}\sigma$. For all $\boldsymbol{\xi} \in \mathbb{S}^{d-1}$,

$$|d\sigma(\langle \boldsymbol{x}, \boldsymbol{\xi} \rangle) g(\boldsymbol{x})| \le \sqrt{d}|g(\boldsymbol{x})| \in L^1(\mathbb{S}^{d-1}).$$

So by the dominated convergence theorem, for all $\boldsymbol{\xi}' \in \mathbb{S}^{d-1}$,

$$\lim_{\boldsymbol{\xi} \to \boldsymbol{\xi}'} T_\psi g(\boldsymbol{\xi}) = \lim_{\boldsymbol{\xi} \to \boldsymbol{\xi}'} \int_{\mathbb{S}^{d-1}} \sqrt{d}\sigma(\langle \boldsymbol{x}, \boldsymbol{\xi} \rangle) g(\boldsymbol{x}) dS(\boldsymbol{x})$$

$$= \int_{\mathbb{S}^{d-1}} \lim_{\boldsymbol{\xi} \to \boldsymbol{\xi}'} \sqrt{d}\sigma(\langle \boldsymbol{x}, \boldsymbol{\xi} \rangle) g(\boldsymbol{x}) dS(\boldsymbol{x})$$

$$= \int_{\mathbb{S}^{d-1}} \sqrt{d}\sigma(\langle \boldsymbol{x}, \boldsymbol{\xi}' \rangle) g(\boldsymbol{x}) dS(\boldsymbol{x})$$

$$= T_\psi g(\boldsymbol{\xi}').$$

Therefore $T_\psi g \in \mathcal{C}(\mathbb{S}^{d-1})$. $\qquad\square$

By the above lemma, for any $\mu \in \mathcal{M}(\mathbb{S}^{d-1})$ and $g \in L^2(\mathbb{S}^{d-1})$, the expressions $\langle T_\psi \mu, g \rangle$ and $\langle \mu, T_\psi g \rangle$ are well-defined and finite. In fact, they are equal to each other.

**Lemma 17.** *Suppose that $\mu \in \mathcal{M}(\mathbb{S}^{d-1})$ and $g \in L^2(\mathbb{S}^{d-1})$. Then*
$$\langle T_\psi \mu, g \rangle = \langle \mu, T_\psi g \rangle.$$

*Proof.* We compute

$$\langle T_\psi \mu, g \rangle = \int_{\mathbb{S}^{d-1}} (T_\psi \mu)(\boldsymbol{\xi}) g(\boldsymbol{\xi}) dS(\xi)$$

$$= \int_{\mathbb{S}^{d-1}} \int_{\mathbb{S}^{d-1}} \psi(\langle \boldsymbol{x}, \boldsymbol{\xi} \rangle) g(\boldsymbol{\xi}) d\mu(\boldsymbol{x}) dS(\boldsymbol{\xi})$$

$$= \int_{\mathbb{S}^{d-1}} \int_{\mathbb{S}^{d-1}} \psi(\langle \boldsymbol{x}, \boldsymbol{\xi} \rangle) g(\boldsymbol{\xi}) dS(\boldsymbol{\xi}) d\mu(\boldsymbol{x})$$

$$= \int_{\mathbb{S}^{d-1}} T_\psi g(\boldsymbol{x}) d\mu(\boldsymbol{x})$$

$$= \langle \mu, T_\psi g \rangle.$$

It remains to justify the change in order of integration in the third line. This follows from Fubini's theorem and the calculation

$$\int_{\mathbb{S}^{d-1}} \int_{\mathbb{S}^{d-1}} |\psi(\langle \boldsymbol{x}, \boldsymbol{\xi} \rangle) g(\boldsymbol{\xi})| dS(\boldsymbol{\xi}) d|\mu|(\boldsymbol{x}) \le \int_{\mathbb{S}^{d-1}} \int_{\mathbb{S}^{d-1}} \sqrt{d}|g(\boldsymbol{\xi})| dS(\boldsymbol{\xi}) d|\mu|(\boldsymbol{x})$$

$$= \int_{\mathbb{S}^{d-1}} \sqrt{d}\|g\|_{L^1} d|\mu|(\boldsymbol{x})$$

$$= \sqrt{d}\|g\|_{L^1} |\mu|(\mathbb{S}^{d-1})$$

$$< \infty,$$

where the last line follows since $g \in L^2(\mathbb{S}^{d-1}) \subset L^1(\mathbb{S}^{d-1})$. $\qquad\square$

In order to characterize how a hemisphere transform acts on $L^2(\mathbb{S}^{d-1})$ and in particular on the spherical harmonics, we will use the *Funk-Hecke formula* (see Seeley, 1966) which states that a certain class of integral operators on $\mathbb{S}^{d-1}$ has an eigendecomposition of spherical harmonics.

**Lemma 18** (Funk-Hecke formula). *Let $\psi : [-1, 1] \to \mathbb{R}$ be a measurable function such that*

$$\int_{-1}^{1} |\psi(t)|(1 - t^2)^{(d-3)/2}dt < \infty.$$

*Then for all $g \in \mathcal{H}_r^d$*

$$\int_{\mathbb{S}^{d-1}} \psi(\langle \boldsymbol{x}, \boldsymbol{\xi} \rangle)g(\boldsymbol{x})dS(\boldsymbol{x}) = c_{r,d}g(\boldsymbol{\xi}),$$

*where*

$$c_{r,d} = \frac{\Gamma(r + 1)\Gamma(d - 2)\Gamma\left(\frac{d}{2}\right)}{\sqrt{\pi}\Gamma(d - 2 + r)\Gamma\left(\frac{d-1}{2}\right)} \int_{-1}^{1} \psi(t)C_r^{(d-2)/2}(t)(1 - t^2)^{(d-3)/2}dt.$$

We will now use the Funk-Hecke formula to compute the coefficients $c_{r,d}$ in the cases where $\psi = \sqrt{d}\sigma$ and $\psi = \dot{\sigma}$. In the following calculations we will use the *Legendre duplication formula*

$$\Gamma(z)\Gamma\left(z + \frac{1}{2}\right) = 2^{1-2z}\sqrt{\pi}\Gamma(2z)$$

and *Euler's reflection formula*

$$\Gamma(1 - z)\Gamma(z) = \frac{\pi}{\sin \pi z}.$$

**Lemma 19.** *For all $d \geq 3$ and $r \geq 0$,*

$$\int_0^1 C_r^{(d-2)/2}(t)(1 - t^2)^{(d-3)/2}dt = \frac{\sqrt{\pi}\Gamma(d + r - 2)\Gamma\left(\frac{d-1}{2}\right)}{2\Gamma(d - 2)\Gamma(r + 1)\Gamma\left(1 - \frac{r}{2}\right)\Gamma\left(\frac{d+r}{2}\right)}.$$

*and*

$$\int_0^1 tC_r^{(d-2)/2}(t)(1 - t^2)^{(d-3)/2}dt = \frac{\sqrt{\pi}\Gamma(d + r - 2)\Gamma\left(\frac{d-1}{2}\right)}{4\Gamma(d - 2)\Gamma(r + 1)\Gamma\left(\frac{3-r}{2}\right)\Gamma\left(\frac{d+r+1}{2}\right)}.$$

*Proof.* We apply the following identity (see Gradshteyn & Ryzhik, 2014, Equation 7.311.2):

$$\int_0^1 t^{r+2\rho}C_r^\nu(t)(1 - t^2)^{\nu-1/2}dt = \frac{\Gamma(2\nu + r)\Gamma(2\rho + r + 1)\Gamma\left(\nu + \frac{1}{2}\right)\Gamma\left(\rho + \frac{1}{2}\right)}{2^{r+1}\Gamma(2\nu)\Gamma(2\rho + 1)r!\Gamma(r + \nu + \rho + 1)}.$$

By the Legendre duplication formula, we have

$$\Gamma\left(\rho + \frac{1}{2}\right)\Gamma(\rho + 1) = 2^{-2\rho}\sqrt{\pi}\Gamma(2\rho + 1)$$

so we can rewrite the above equation as

$$\int_0^1 t^{r+2\rho}C_r^\nu(t)(1 - t^2)^{\nu-1/2}dt = \frac{\sqrt{\pi}\Gamma(2\nu + r)\Gamma(2\rho + r + 1)\Gamma\left(\nu + \frac{1}{2}\right)}{2^{2\rho+r+1}\Gamma(2\nu)\Gamma(\rho + 1)\Gamma(r + 1)\Gamma(r + \nu + \rho + 1)}. \tag{13}$$

Substituting $\rho = -r/2$ and $\nu = (d - 2)/2$ into (13) yields

$$\int_0^1 C_r^{(d-2)/2}(t)(1 - t^2)^{(d-3)/2}dt = \frac{\sqrt{\pi}\Gamma(d + r - 2)\Gamma\left(\frac{d-1}{2}\right)}{2\Gamma(d - 2)\Gamma\left(1 - \frac{r}{2}\right)\Gamma(r + 1)\Gamma\left(\frac{d+r}{2}\right)},$$

which establishes the first identity of the claim.

Substituting $\rho = (1 - r)/2$ and $\nu = (d - 2)/2$ into (13) yields

$$\int_0^1 C_r^{(d-2)/2}(t)(1 - t^2)^{(d-3)/2}dt = \frac{\sqrt{\pi}\Gamma(d + r - 2)\Gamma\left(\frac{d-1}{2}\right)}{4\Gamma(d - 2)\Gamma\left(\frac{3-r}{2}\right)\Gamma(r + 1)\Gamma\left(\frac{d+r+1}{2}\right)},$$

which establishes the second identity of the claim. $\qquad\square$

**Lemma 20.** *Suppose that $g \in \mathcal{H}_r^d$ and $d \geq 3$. Then for all $r \geq 0$, $T_{\dot{\sigma}}g = c_{r,d}g$, where*

$$c_{r,d} = \frac{\Gamma\left(\frac{d}{2}\right)}{2\Gamma\left(1 - \frac{r}{2}\right)\Gamma\left(\frac{r}{2} + \frac{d}{2}\right)}.$$

*Moreover, if $0 \leq r \leq R$, then*

$$|c_{2r+1,d}| \geq \frac{\Gamma\left(\frac{d}{2}\right)\Gamma\left(\frac{2R+1}{2}\right)}{2\pi\Gamma\left(\frac{d+2R+1}{2}\right)}.$$

*Proof.* Let $g \in \mathcal{H}_r^d$. By Lemma 18,

$$T_{\dot{\sigma}}g = c_{r,d}g,$$

where

$$
\begin{aligned}
c_{r,d} &= \frac{\Gamma(r+1)\Gamma(d-2)\Gamma\left(\frac{d}{2}\right)}{\sqrt{\pi}\Gamma(d-2+r)\Gamma\left(\frac{d-1}{2}\right)} \int_{-1}^{1} \dot{\sigma}(t)C_r^{(d-2)/2}(t)(1-t^2)^{(d-3)/2}dt \\
&= \frac{\Gamma(r+1)\Gamma(d-2)\Gamma\left(\frac{d}{2}\right)}{\sqrt{\pi}\Gamma(d-2+r)\Gamma\left(\frac{d-1}{2}\right)} \int_{0}^{1} C_r^{(d-2)/2}(t)(1-t^2)^{(d-3)/2}dt.
\end{aligned}
$$

By Lemma 19, this is equal to

$$\frac{\Gamma(r+1)\Gamma(d-2)\Gamma\left(\frac{d}{2}\right)}{\sqrt{\pi}\Gamma(d-2+r)\Gamma\left(\frac{d-1}{2}\right)} \cdot \frac{\sqrt{\pi}\Gamma(d+r-2)\Gamma\left(\frac{d-1}{2}\right)}{2\Gamma(d-2)\Gamma(r+1)\Gamma\left(1-\frac{r}{2}\right)\Gamma\left(\frac{d+r}{2}\right)} = \frac{\Gamma\left(\frac{d}{2}\right)}{2\Gamma\left(1-\frac{r}{2}\right)\Gamma\left(\frac{d+r}{2}\right)}$$

as claimed.

Now we proceed with the second statement. We claim that whenever $0 \leq r \leq R$,

$$|c_{2R+1,d}| \leq |c_{2r+1,d}|.$$

We prove this by induction on $R$. For the base case $R = r$, the claim trivially holds. Now suppose that the claim holds for some $R \geq r$. Then

$$
\begin{aligned}
|c_{2(R+1)+1,d}| &= \left| \frac{\Gamma\left(\frac{d}{2}\right)}{2\Gamma\left(1 - \frac{2R+3}{2}\right)\Gamma\left(\frac{2R+3}{2} + \frac{d}{2}\right)} \right| \\
&= \left| \frac{\left(-\frac{2R+1}{2}\right)\Gamma\left(\frac{d}{2}\right)}{2\Gamma\left(1 - \frac{2R+1}{2}\right)\left(\frac{2R+1}{2} + \frac{d}{2}\right)\Gamma\left(\frac{2R+1}{2} + \frac{d}{2}\right)} \right| \\
&= |c_{2R+1,d}|\frac{2R+1}{2R+1+d} \\
&\leq |c_{2R+1,d}| \\
&\leq |c_{2r+1,d}|.
\end{aligned}
$$

Hence by induction $|c_{2R+1,d}| \leq |c_{2r+1,d}|$ for all $0 \leq r \leq R$. Now suppose that $0 \leq r \leq R$. By Euler's reflection formula,

$$
\begin{aligned}
c_{2R+1,d} &= \frac{\Gamma\left(\frac{d}{2}\right)}{2\Gamma\left(1 - \frac{2R+1}{2}\right)\Gamma\left(\frac{2R+1}{2} + \frac{d}{2}\right)} \\
&= \frac{\Gamma\left(\frac{d}{2}\right)\sin\left(\pi\frac{2R+1}{2}\right)\Gamma\left(\frac{2R+1}{2}\right)}{2\pi\Gamma\left(\frac{2R+1}{2} + \frac{d}{2}\right)} \\
&= \frac{\Gamma\left(\frac{d}{2}\right)(-1)^R\Gamma\left(\frac{2R+1}{2}\right)}{2\pi\Gamma\left(\frac{2R+1}{2} + \frac{d}{2}\right)}
\end{aligned}
$$

so

$$
\begin{aligned}
|c_{2r+1,d}| &\geq |c_{2R+1,d}| \\
&= \frac{\Gamma\left(\frac{d}{2}\right)\Gamma\left(\frac{2R+1}{2}\right)}{2\pi\Gamma\left(\frac{d+2R+1}{2}\right)}.
\end{aligned}
$$

$\square$

**Lemma 21.** *Suppose that $g \in \mathcal{H}_r^d$ and $d \geq 3$. Then $T_{\sqrt{d}\sigma} g = c_{r,d} g$, where*

$$c_{r,d} = \frac{\sqrt{d}\Gamma\left(\frac{d}{2}\right)}{4\Gamma\left(\frac{3-r}{2}\right)\Gamma\left(\frac{d+r+1}{2}\right)}.$$

*Moreover, if $0 \leq r \leq R$, then*

$$|c_{2r,d}| \geq \frac{\sqrt{d}\Gamma\left(\frac{d}{2}\right)\Gamma\left(\frac{2R-1}{2}\right)}{4\pi\Gamma\left(\frac{d+2R+1}{2}\right)}.$$

*Proof.* The proof is analogous to that of Lemma 20. Let $g \in \mathcal{H}_r^d$. By Lemma 18,

$$T_{\sqrt{d}\sigma} g = c_{r,d} g,$$

where

$$
\begin{aligned}
c_{r,d} &= \frac{\Gamma(r+1)\Gamma(d-2)\Gamma\left(\frac{d}{2}\right)}{\sqrt{\pi}\Gamma(d-2+r)\Gamma\left(\frac{d-1}{2}\right)} \int_{-1}^{1} \sqrt{d}\sigma(t) C_r^{(d-2)/2}(t)(1-t^2)^{(d-3)/2} dt \\
&= \frac{\sqrt{d}\Gamma(r+1)\Gamma(d-2)\Gamma\left(\frac{d}{2}\right)}{\sqrt{\pi}\Gamma(d-2+r)\Gamma\left(\frac{d-1}{2}\right)} \int_{0}^{1} t C_r^{(d-2)/2}(t)(1-t^2)^{(d-3)/2} dt.
\end{aligned}
$$

By Lemma 19, this is equal to

$$\frac{\sqrt{d}\Gamma(r+1)\Gamma(d-2)\Gamma\left(\frac{d}{2}\right)}{\sqrt{\pi}\Gamma(d-2+r)\Gamma\left(\frac{d-1}{2}\right)} \cdot \frac{\sqrt{\pi}\Gamma(d+r-2)\Gamma\left(\frac{d-1}{2}\right)}{4\Gamma(d-2)\Gamma(r+1)\Gamma\left(\frac{3-r}{2}\right)\Gamma\left(\frac{d+r+1}{2}\right)} = \frac{\sqrt{d}\Gamma\left(\frac{d}{2}\right)}{4\Gamma\left(\frac{3-r}{2}\right)\Gamma\left(\frac{d+r+1}{2}\right)}$$

as claimed.

We claim that whenever $0 \leq r \leq R$,

$$|c_{2R,d}| \leq |c_{2r,d}|.$$

We prove this by induction on $R$. For the base case $R = r$, the claim trivially holds. Now suppose that the claim holds for some $R \geq r$. Then

$$
\begin{aligned}
|c_{2(R+1)}| &= \left| \frac{\sqrt{d}\Gamma\left(\frac{d}{2}\right)}{4\Gamma\left(\frac{1-2R}{2}\right)\Gamma\left(\frac{d+2R+3}{2}\right)} \right| \\
&= \left| \frac{\left(\frac{1-2R}{2}\right)\sqrt{d}\Gamma\left(\frac{d}{2}\right)}{4\Gamma\left(\frac{1-2R}{2}\right)\left(\frac{d+2R+1}{2}\right)\Gamma\left(\frac{d+2R+1}{2}\right)} \right| \\
&= c_{2R}\frac{|2R-1|}{d+2R+1} \\
&\leq c_{2R}.
\end{aligned}
$$

Hence by induction $|c_{2R}| \leq |c_{2r}|$ for all $0 \leq r \leq R$. Now suppose that $0 \leq r \leq R$. By Euler's reflection formula,

$$
\begin{aligned}
c_{2R,d} &= \frac{\sqrt{d}\Gamma\left(\frac{d}{2}\right)}{4\Gamma\left(\frac{3-2R}{2}\right)\Gamma\left(\frac{d+2R+1}{2}\right)} \\
&= \frac{\sqrt{d}\Gamma\left(\frac{d}{2}\right)\sin\left(\pi\frac{2R-1}{2}\right)\Gamma\left(\frac{2R-1}{2}\right)}{4\pi\Gamma\left(\frac{d+2R+1}{2}\right)} \\
&= \frac{(-1)^{R+1}\sqrt{d}\Gamma\left(\frac{d}{2}\right)\Gamma\left(\frac{2R-1}{2}\right)}{4\pi\Gamma\left(\frac{d+2R+1}{2}\right)}
\end{aligned}
$$

so

$$
\begin{aligned}
|c_{2r,d}| &\geq |c_{2R,d}| \\
&= \frac{\sqrt{d}\Gamma\left(\frac{d}{2}\right)\Gamma\left(\frac{2R-1}{2}\right)}{4\pi\Gamma\left(\frac{d+2R+1}{2}\right)}.
\end{aligned}
$$

$\square$

# C   Proofs for Section 3

First we observe the connection between the smallest eigenvalue of the expected NTK when the weights are drawn uniformly over the sphere versus as Gaussian.

**Lemma 22.** *If $X \in \mathbb{R}^{d_0 \times n}$, then*

$$\lambda_{\min}\left(\mathbb{E}_{\boldsymbol{w} \sim \mathcal{N}(\mathbf{0}_d, \boldsymbol{I}_d)}\left[\sigma\left(\boldsymbol{X}^T \boldsymbol{w}\right)\sigma\left(\boldsymbol{w}^T \boldsymbol{X}\right)\right]\right) = d_0 \lambda_{\min}\left(\mathbb{E}_{\boldsymbol{u} \sim U(\mathbb{S}^{d_0-1})}\left[\sigma\left(\boldsymbol{X}^T \boldsymbol{u}\right)\sigma\left(\boldsymbol{u}^T \boldsymbol{X}\right)\right]\right).$$

*Proof.* Since the distribution of $\boldsymbol{w}$ is rotationally invariant, we can decompose $\boldsymbol{w} = \alpha \boldsymbol{u}$, where $\alpha = \|\boldsymbol{w}\|$, $\boldsymbol{u}$ is uniformly distributed on $\mathbb{S}^{d_0-1}$, and $\alpha$ and $\boldsymbol{u}$ are independent. Then

$$\begin{aligned}
\lambda_{\min}\left(\mathbb{E}_{\boldsymbol{w} \sim \mathcal{N}(\mathbf{0}_d, \boldsymbol{I}_d)}\left[\sigma\left(\boldsymbol{X}^T \boldsymbol{w}\right)\sigma\left(\boldsymbol{w}^T \boldsymbol{X}\right)\right]\right) &= \lambda_{\min}\left(\mathbb{E}\left[\sigma\left(\boldsymbol{X}^T \boldsymbol{w}\right)\sigma\left(\boldsymbol{w}^T \boldsymbol{X}\right)\right]\right)\\
&= \lambda_{\min}\left(\mathbb{E}\left[\alpha^2 \sigma\left(\boldsymbol{X}^T \boldsymbol{u}\right)\sigma\left(\boldsymbol{u}^T \boldsymbol{X}\right)\right]\right)\\
&= \lambda_{\min}\left(\mathbb{E}\left[\alpha^2\right]\mathbb{E}\left[\sigma\left(\boldsymbol{X}^T \boldsymbol{u}\right)\sigma\left(\boldsymbol{u}^T \boldsymbol{X}\right)\right]\right)\\
&= d_0 \lambda_{\min}\left(\mathbb{E}\left[\sigma\left(\boldsymbol{X}^T \boldsymbol{u}\right)\sigma\left(\boldsymbol{u}^T \boldsymbol{X}\right)\right]\right).
\end{aligned}$$

$\square$

Lemma 22 is useful in that studying the expected NTK in the shallow setting for uniform weights here will prove more convenient than working directly with Gaussian weights.

## C.1   Proof of Lemma 3

**Lemma 3.** *Suppose that $\boldsymbol{x}_1, \cdots, \boldsymbol{x}_n \in \mathbb{S}^{d-1}$. Let*

$$\lambda_1 = \lambda_{\min}\left(\mathbb{E}_{\boldsymbol{u} \sim U(\mathbb{S}^{d-1})}\left[\dot{\sigma}\left(\boldsymbol{X}^T \boldsymbol{u}\right)\dot{\sigma}\left(\boldsymbol{u}^T \boldsymbol{X}\right)\right]\right).$$

*If $\lambda_1 > 0$ and $d_1 \gtrsim \lambda_1^{-1}\|\boldsymbol{X}\|^2 \log \frac{n}{\epsilon}$, then with probability at least $1 - \epsilon$*

$$\lambda_{\min}(\boldsymbol{K}_1) \gtrsim \lambda_1.$$

*Proof.* By the scale-invariance of $\dot{\sigma}$,

$$\lambda_1 = \lambda_{\min}\left(\mathbb{E}_{\boldsymbol{u} \sim \mathcal{N}(\mathbf{0}_d, \boldsymbol{I}_d)}\left[\dot{\sigma}\left(\boldsymbol{X}^T \boldsymbol{u}\right)\dot{\sigma}\left(\boldsymbol{u}^T \boldsymbol{X}\right)\right]\right).$$

For each $i \in [n]$ and $j \in [d_1]$,

$$\nabla_{\boldsymbol{w}_j} f(\boldsymbol{x}_i) = \frac{1}{\sqrt{d_1}} v_j \dot{\sigma}\left(\langle \boldsymbol{w}_j^T, \boldsymbol{x}_i \rangle\right) \boldsymbol{x}_i$$

and therefore

$$\boldsymbol{K}_1 = \frac{1}{d_1} \sum_{j=1}^{d_1} \boldsymbol{Z}_j,$$

where

$$\boldsymbol{Z}_j = v_j^2 \left(\dot{\sigma}\left(\boldsymbol{X}^T \boldsymbol{w}_j\right)\dot{\sigma}\left(\boldsymbol{w}_j^T \boldsymbol{X}\right)\right) \odot \left(\boldsymbol{X}^T \boldsymbol{X}\right).$$

For each $j \in [d_1]$, let $\xi_j \in \{0, 1\}$ be a random variable taking value 1 if $|v_j| \leq 1$ and taking value 0 otherwise. Since $v_j$ is a standard Gaussian there exists a universal constant $C_1 > 0$ with $\mathbb{E}[\xi_j v_j] = C_1$ for all $j$. We also define $\boldsymbol{Z}_j' = \xi_j \boldsymbol{Z}_j$. Note that $\boldsymbol{Z}_j' \succeq \mathbf{0}$, and by the inequality $\lambda_{\max}(\boldsymbol{A} \odot \boldsymbol{B}) \leq \max_i [\boldsymbol{A}]_{ii} \lambda_{\max}(\boldsymbol{B})$,

$$\begin{aligned}
\|\boldsymbol{Z}_j'\| &= \left\|\xi_j v_j^2 \left(\dot{\sigma}\left(\boldsymbol{X}^T \boldsymbol{w}_j\right)\dot{\sigma}\left(\boldsymbol{w}_j \boldsymbol{X}\right)\right) \odot \left(\boldsymbol{X}^T \boldsymbol{X}\right)\right\|\\
&\leq \max_{i \in [n]}\left|\left(\xi_j v_j^2 \left[\dot{\sigma}\left(\boldsymbol{X}^T \boldsymbol{w}_j\right)\dot{\sigma}\left(\boldsymbol{w}_j^T \boldsymbol{X}\right)\right)\right]_{ii}\right| \cdot \|\boldsymbol{X}^T \boldsymbol{X}\|\\
&= \max_{i \in [n]}\left|\xi_j v_j^2 \dot{\sigma}\left(\boldsymbol{w}_j^T \boldsymbol{x}_i\right)^2\right| \cdot \|\boldsymbol{X}\|^2\\
&\leq \|\boldsymbol{X}\|^2.
\end{aligned}$$

Furthermore by the inequality $\lambda_{\min}(\boldsymbol{A} \odot \boldsymbol{B}) \geq \min_i [\boldsymbol{A}]_{ii} \lambda_{\min}(\boldsymbol{B})$,

$$
\begin{aligned}
\lambda_{\min}\left(\mathbb{E}[\boldsymbol{Z}_j']\right) &= \lambda_{\min}\left(\mathbb{E}\left[\xi_j v_j^2 \left(\dot{\sigma}\left(\boldsymbol{X}^T \boldsymbol{w}_j\right) \dot{\sigma}\left(\boldsymbol{w}_j^T \boldsymbol{X}\right)\right)\right] \odot \left(\boldsymbol{X}^T \boldsymbol{X}\right)\right) \\
&\geq \lambda_{\min}\left(\mathbb{E}\left[\xi_j v_j^2 \left(\dot{\sigma}\left(\boldsymbol{X}^T \boldsymbol{w}_j\right) \dot{\sigma}\left(\boldsymbol{w}_j^T \boldsymbol{X}\right)\right)\right]\right) \min_{i \in [n]}\left|\left(\boldsymbol{X}^T \boldsymbol{X}\right)_{ii}\right| \\
&= \lambda_{\min}\left(\mathbb{E}\left[\xi_j v_j^2\right] \mathbb{E}\left[\left(\dot{\sigma}\left(\boldsymbol{X}^T \boldsymbol{w}_j\right) \dot{\sigma}\left(\boldsymbol{w}_j^T \boldsymbol{X}\right)\right)\right]\right) \min_{i \in [n]}\|\boldsymbol{x}_i\|^2 \\
&= C_1 \lambda_{\min}\left(\mathbb{E}\left[\left(\dot{\sigma}\left(\boldsymbol{X}^T \boldsymbol{w}_j\right) \dot{\sigma}\left(\boldsymbol{w}_j^T \boldsymbol{X}\right)\right)\right]\right) \\
&= C_1 \lambda_1.
\end{aligned}
$$

So by Lemma 13, for all $t \geq 0$

$$
\mathbb{P}\left(\lambda_{\min}\left(\frac{1}{d_1}\sum_{j=1}^{d_1} \boldsymbol{Z}_j'\right) \leq C_1 \lambda_1\right) \leq \mathbb{P}\left(\lambda_{\min}\left(\frac{1}{d_1}\sum_{j=1}^{d_1} \boldsymbol{Z}_j'\right) \leq \mathbb{E}[\boldsymbol{Z}_1']\right)
$$

$$
\leq n \exp\left(-\frac{C_2 d_1 \lambda_1}{\|\boldsymbol{X}\|^2}\right)
$$

where $C_2 > 0$ is a constant. Since $\boldsymbol{Z}_j \succeq \boldsymbol{Z}_j'$ for all $j \in [d_1]$, if $d_1 \geq \frac{1}{C_2 \lambda_1}\|\boldsymbol{X}\|^2 \log\left(\frac{n}{\epsilon}\right)$, then

$$
\mathbb{P}\left(\lambda_{\min}\left(\frac{1}{d_1}\sum_{j=1}^{d_1} \boldsymbol{Z}_j\right) \leq C_1 \lambda_1\right) \leq n \exp\left(-\frac{C_2 d_1 \lambda_1}{\|\boldsymbol{X}\|^2}\right)
$$

$$
\leq \epsilon.
$$

$\square$

## C.2 Proof of Lemma 4

**Lemma 23.** *Suppose that $\boldsymbol{x}_1, \cdots, \boldsymbol{x}_n \in \mathbb{S}^{d_0-1}$. Let*

$$
\lambda_2 = d_0 \lambda_{\min}\left(\mathbb{E}_{\boldsymbol{u} \sim U(\mathbb{S}^{d_0-1})}\left[\sigma(\boldsymbol{X}^T \boldsymbol{u})\sigma(\boldsymbol{u}^T \boldsymbol{X})\right]\right).
$$

*If $\lambda_2 > 0$ and $d_1 \gtrsim \frac{n}{\lambda_2} \log\left(\frac{n}{\lambda_2}\right) \log\left(\frac{n}{\epsilon}\right)$, then with probability at least $1 - \epsilon$, $\lambda_{\min}(\boldsymbol{K}_2) \geq \frac{\lambda_2}{4}$.*

*Proof.* Note that by Lemma 22,

$$
\lambda_2 = \lambda_{\min}\left(\mathbb{E}_{\boldsymbol{w} \sim \mathcal{N}(\boldsymbol{0}_d, \boldsymbol{I}_d)}\left[\sigma\left(\boldsymbol{X}^T \boldsymbol{w}\right) \sigma\left(\boldsymbol{w}^T \boldsymbol{X}\right)\right]\right).
$$

For each $i \in [n]$ and $j \in [d_1]$,

$$
\nabla_{v_j} f(\boldsymbol{x}_i) = \frac{1}{\sqrt{d_1}} \sigma(\boldsymbol{w}_j^T \boldsymbol{x}_i)
$$

and therefore

$$
\boldsymbol{K}_2 = \frac{1}{d_1}\sum_{j=1}^{d_1} \boldsymbol{Z}_j,
$$

where

$$
\boldsymbol{Z}_j = \sigma\left(\boldsymbol{X}^T \boldsymbol{w}_j\right) \sigma\left(\boldsymbol{w}_j^T \boldsymbol{X}\right).
$$

By Vershynin (2018, Theorem 6.3.2), for each $j \in [d_1]$

$$
\begin{aligned}
\left\|\left\|\boldsymbol{X}^T \boldsymbol{w}_j\right\|\right\|_{\psi_2} &\lesssim \left\|\left\|\boldsymbol{X}^T \boldsymbol{w}_j\right\| - \left\|\boldsymbol{X}^T\right\|_F\right\|_{\psi_2} + \left\|\boldsymbol{X}^T\right\|_F \\
&\lesssim \left\|\boldsymbol{X}^T\right\| + \left\|\boldsymbol{X}^T\right\|_F \\
&\lesssim \left\|\boldsymbol{X}^T\right\|_F \\
&= \|\boldsymbol{X}\|_F \\
&= \sqrt{n}.
\end{aligned}
$$

So by Hoeffding's inequality, for all $t \geq 0$

$$\mathbb{P}\left(\left\|\boldsymbol{X}^T \boldsymbol{w}_j\right\|^2 \geq t\right) = \mathbb{P}\left(\left\|\boldsymbol{X}^T \boldsymbol{w}_j\right\| \geq \sqrt{t}\right) \leq 2 \exp\left(-\frac{C_1 t}{n}\right) \tag{14}$$

for some constant $C_1 > 0$. Let $s = \frac{n}{C_1} \log \frac{4n}{\lambda_2 C_1}$. For each $j \in [d_1]$ let $\xi_j \in \{0, 1\}$ be a random variable taking value 1 if $\|\boldsymbol{X}^T \boldsymbol{w}_j\|^2 \leq s$ and taking value 0 otherwise. Let $\boldsymbol{Z}'_j = \xi_j \boldsymbol{Z}_j$. For each $j \in [m]$, $\boldsymbol{Z}'_j \succeq 0$, and

$$\begin{aligned}
\left\|\boldsymbol{Z}'_j\right\| &= \left\|\xi_j \sigma\left(\boldsymbol{X}^T \boldsymbol{w}_j\right) \sigma\left(\boldsymbol{w}_j^T \boldsymbol{X}\right)\right\| \\
&= \left\|\xi_j \sigma\left(\boldsymbol{X}^T \boldsymbol{w}_j\right)\right\|^2 \\
&\leq s.
\end{aligned}$$

Moreover,

$$\begin{aligned}
\left\|\mathbb{E}[\boldsymbol{Z}_j] - \mathbb{E}[\boldsymbol{Z}'_j]\right\| &= \left\|\mathbb{E}\left[(1 - \xi_j)\sigma\left(\boldsymbol{X}^T \boldsymbol{w}_j\right) \sigma\left(\boldsymbol{w}_j^T \boldsymbol{X}\right)\right]\right\| \\
&\leq \mathbb{E}\left[(1 - \xi_j)\left\|\sigma\left(\boldsymbol{X}^T \boldsymbol{w}_j\right) \sigma\left(\boldsymbol{w}_j^T \boldsymbol{X}\right)\right\|\right] \\
&= \mathbb{E}\left[(1 - \xi_j)\left\|\sigma\left(\boldsymbol{X}^T \boldsymbol{w}_j\right)\right\|^2\right] \\
&= \frac{1}{2}\mathbb{E}\left[(1 - \xi_j)\left\|\boldsymbol{X}^T \boldsymbol{w}_j\right\|^2\right] \\
&= \frac{1}{2}\int_s^\infty \mathbb{P}\left(\left\|\boldsymbol{X}^T \boldsymbol{w}_j\right\|^2 \geq t\right) dt \\
&\leq 2\int_s^\infty \exp\left(-\frac{C_1 t}{n}\right) dt \\
&= \frac{2n}{C_1}\exp\left(-\frac{C_1 s}{n}\right) \\
&= \frac{\lambda_2}{2}.
\end{aligned}$$

Here we used (14) in line 6. By Weyl's inequality,

$$\lambda_{\min}(\mathbb{E}[\boldsymbol{Z}'_j]) \geq \lambda_{\min}(\mathbb{E}[\boldsymbol{Z}_j]) - \left\|\mathbb{E}[\boldsymbol{Z}_j] - \mathbb{E}[\boldsymbol{Z}'_j]\right\| = \lambda_2 - \frac{\lambda_2}{2} = \frac{\lambda_2}{2}.$$

By Lemma 13,

$$\begin{aligned}
\mathbb{P}\left(\lambda_{\min}\left(\frac{1}{d_1}\sum_{j=1}^{d_1} \boldsymbol{Z}'_j\right) \leq \frac{\lambda_2}{4}\right) &\leq \mathbb{P}\left(\lambda_{\min}\left(\frac{1}{m}\sum_{j=1}^{m} \boldsymbol{Z}'_j\right) \leq \frac{1}{2}\lambda_{\min}(\mathbb{E}[\boldsymbol{Z}'_1])\right) \\
&\leq n \exp\left(-\frac{C_2 d_1 \lambda_{\min}(\mathbb{E}[\boldsymbol{Z}'_1])}{s}\right) \\
&\leq n \exp\left(\frac{-C_2 d_1 \lambda_2}{2s}\right).
\end{aligned}$$

Since $\boldsymbol{Z}'_j \preceq \boldsymbol{Z}_j$ for all $j$, for $d_1 \geq \frac{2s}{C_2 \lambda_2} \log \frac{n}{\epsilon}$ this implies

$$\mathbb{P}\left(\lambda_{\min}\left(\frac{1}{d_1}\sum_{j=1}^{d_1} \boldsymbol{Z}_j\right) \leq \frac{\lambda_2}{4}\right) \leq n \exp\left(-\frac{C_2 d_1 \lambda_2}{2s}\right)$$

$$\leq \epsilon.$$

In other words,

$$\mathbb{P}\left(\lambda_{\min}(\boldsymbol{K}_2) \geq \frac{\lambda_2}{4}\right) \geq 1 - \epsilon.$$

$\square$

## C.3 Proof of Lemma 5

**Lemma 5.** *Fix $\boldsymbol{X} \in \mathbb{R}^{d \times n}$ and $\psi \in \{\sqrt{d}\sigma, \dot\sigma\}$. For all $\boldsymbol{z} \in \mathbb{R}^n$, $\langle \boldsymbol{K}_\psi^\infty \boldsymbol{z}, \boldsymbol{z} \rangle = \|T_\psi \mu_{\boldsymbol{z}}\|^2$. Moreover,*

$$\lambda_{\min}(\boldsymbol{K}_\psi^\infty) = \inf_{\|\boldsymbol{z}\|=1} \|T_\psi \mu_{\boldsymbol{z}}\|^2.$$

*Proof.* We compute

$$
\begin{aligned}
\langle \boldsymbol{K}_\psi^\infty \boldsymbol{z}, \boldsymbol{z} \rangle &= \mathbb{E}_{\boldsymbol{w} \sim U(\mathbb{S}^{d-1})} \left[ \left| \psi\left(\boldsymbol{w}^T \boldsymbol{X}\right) \boldsymbol{z} \right|^2 \right] \\
&= \int_{\mathbb{S}^{d-1}} \left| \psi\left(\boldsymbol{w}^T \boldsymbol{X}\right) \boldsymbol{z} \right|^2 dS(\boldsymbol{w}) \\
&= \int_{\mathbb{S}^{d-1}} \left| \sum_{i=1}^n \psi(\langle \boldsymbol{w}, \boldsymbol{x}_i \rangle) z_i \right|^2 dS(\boldsymbol{w}) \\
&= \int_{\mathbb{S}^{d-1}} \left| \int_{\mathbb{S}^{d-1}} \psi(\langle \boldsymbol{w}, \boldsymbol{x} \rangle) d\mu_{\boldsymbol{z}}(\boldsymbol{x}) \right|^2 dS(\boldsymbol{w}) \\
&= \int_{\mathbb{S}^{d-1}} \left| T_\psi \mu_{\boldsymbol{z}}(\boldsymbol{w}) \right|^2 dS(\boldsymbol{w}) \\
&= \|T_\psi \mu_{\boldsymbol{z}}\|^2
\end{aligned}
$$

which establishes the first part of the result. The second part of the result follows immediately by writing

$$\lambda_{\min}(\boldsymbol{K}_\psi^\infty) = \inf_{\|\boldsymbol{z}\|=1} \langle \boldsymbol{K}_\psi^\infty \boldsymbol{z}, \boldsymbol{z} \rangle = \inf_{\|\boldsymbol{z}\|=1} \|T_\psi \mu_{\boldsymbol{z}}\|^2.$$

$\square$

## C.4 Proof of Lemma 6

**Lemma 6.** *Suppose $\boldsymbol{x}_1, \cdots, \boldsymbol{x}_n \in \mathbb{S}^{d-1}$ are $\delta$-separated. Suppose that $\beta \in \{0,1\}$ and that $R \in \mathbb{Z}_{\geq 0}$ are such that $N := \sum_{r=0}^R \dim(\mathcal{H}_{2r+\beta}^d)$ satisfies $N \geq C \left(\frac{\delta^4}{2}\right)^{-(d-2)/2}$ where $C > 0$ is a universal constant. Let $g_1, \cdots, g_N$ be spherical harmonics which form an orthonormal basis of $\bigoplus_{r=0}^R \mathcal{H}_{2r+\beta}^d$. If $\boldsymbol{D} \in \mathbb{R}^{N \times n}$ is defined as $\boldsymbol{D}_{ai} = g_a(\boldsymbol{x}_i)$ then $\sigma_{\min}(\boldsymbol{D}) \geq \sqrt{\frac{N}{2}}$.*

*Proof.* Note that

$$N = \sum_{r=0}^R \left( \binom{2r+\beta+d-1}{d-1} - \binom{2r+\beta+d-3}{d-1} \right) = \binom{2R+\beta+d-1}{d-1}.$$

Let us write $\boldsymbol{D} = [\boldsymbol{d}_1, \cdots, \boldsymbol{d}_n]$. Fix $i, k \in [n]$ with $i \neq k$. By the addition formula (10),

$$
\begin{aligned}
\|\boldsymbol{d}_i\|^2 &= \sum_{a=1}^N g_a(\boldsymbol{x}_i)^2 \\
&= \sum_{r=0}^R \sum_{s=1}^{\dim(\mathcal{H}_{2r+\beta}^d)} Y_{r,s}^d(\boldsymbol{x}_i)^2 \\
&= \sum_{r=0}^R \dim(\mathcal{H}_{2r+\beta}^d) \\
&= N.
\end{aligned}
$$

By Lemma 15 and $\delta$-separation, there exists a constant $C > 0$ such that

$$
\begin{aligned}
|\langle \boldsymbol{d}_i, \boldsymbol{d}_k \rangle| &= \left| \sum_{a=1}^{N} g_a(\boldsymbol{x}_i) g_a(\boldsymbol{x}_k) \right| \\
&\leq C \left( \frac{\delta^4}{2} \right)^{-(d-2)/4} \left( \frac{2R + \beta + d - 1}{d - 1} \right)^{1/2} \\
&= C N^{1/2} \left( \frac{\delta^4}{2} \right)^{-(d-2)/4}.
\end{aligned}
$$

Suppose that

$$
N \geq 2C^2 \left( \frac{\delta^4}{2} \right)^{-(d-2)/2}.
$$

Observe that $\sigma_{\min}(\boldsymbol{D})$ is the square root of the minimum eigenvalue of $\boldsymbol{D}^T \boldsymbol{D}$. By the Gershgorin circle theorem, the minimum eigenvalue of $\boldsymbol{D}^T \boldsymbol{D}$ is at least

$$
\min_{i \in [n]} \left( |(\boldsymbol{D}^T \boldsymbol{D})_{ii}| - \sum_{k \neq i} |\boldsymbol{D}^T \boldsymbol{D}|_{ik} \right) = \min_{i \in [n]} \left( \|\boldsymbol{d}_i\|^2 - \sum_{k \neq i} |\langle \boldsymbol{d}_i, \boldsymbol{d}_k \rangle| \right)
$$

$$
\geq \frac{N}{2}.
$$

The result follows. $\qquad\square$

### C.5 Proof of Lemma 7

**Lemma 24.** *Let $\epsilon \in (0, 1)$ and let $\delta > 0$. Suppose that $\boldsymbol{x}_1, \cdots, \boldsymbol{x}_n \in \mathbb{S}^{d-1}$ form a $\delta$-separated dataset. Let $R \in \mathbb{N}$ be such that*

$$
\binom{2R + d - 1}{d - 1} \geq C \left( \frac{\delta^4}{2} \right)^{-(d-2)/2}
$$

*where $C > 0$ is a universal constant. Then*

$$
\|T_\psi \mu_{\boldsymbol{z}}\|^2 \gtrsim \begin{cases} (d + R)^{1/2} d^{-1/2} R^{-3/2} & \text{if } \psi = \dot{\sigma} \\ (d + R)^{-1/2} d^{1/2} R^{-3/2} & \text{if } \psi = \sqrt{d}\sigma \end{cases}
$$

*for all $\boldsymbol{z} \in \mathbb{R}^n$ with $\|\boldsymbol{z}\| \leq 1$.*

*Proof.* Let $C$ be the same constant as in Lemma 6 and suppose that

$$
\binom{2R + d - 1}{d - 1} \geq C \left( \frac{\delta^4}{2} \right)^{-(d-2)/2}.
$$

Let $\beta \in \{0, 1\}$ satisfy $\beta = 1$ when $\psi = \dot{\sigma}$ and $\beta = 0$ when $\psi = d\sigma$. Let $N = \sum_{r=0}^{R} \dim(\mathcal{H}_{2r+\beta}^d)$. Note that

$$
\begin{aligned}
N &= \sum_{r=0}^{R} \left( \binom{2r + d + \beta - 1}{d - 1} - \binom{2r + d + \beta - 3}{d - 1} \right) \\
&= \binom{2R + d + \beta - 1}{d - 1} \\
&\geq \binom{2R + d - 1}{d - 1} \\
&\geq C \left( \frac{\delta^4}{2} \right)^{-(d-2)/2}.
\end{aligned}
$$

Let $g_1, \cdots, g_N$ be spherical harmonics forming an orthonormal basis of $\bigoplus_{r=1}^{R} \mathcal{H}_{2r-1}^d$, and let $\boldsymbol{B} \in \mathbb{R}^{N \times n}$ be the matrix defined by $\boldsymbol{B}_{ai} = g_a(\boldsymbol{x}_i)$. By Lemma 6, $\sigma_{\min}(\boldsymbol{B}) \geq \sqrt{\frac{N}{2}}$ with probability at least $1 - \epsilon$. Since the functions $g_a$ are orthonormal,

$$\|T_\psi \mu_{\boldsymbol{z}}\|^2 \geq \sum_{a=1}^{N} |\langle T_\psi \mu_{\boldsymbol{z}}, g_a \rangle|^2.$$

By Lemma 17 the above expression is equal to

$$\sum_{a=1}^{N} |\langle \mu_{\boldsymbol{z}}, T_\psi g_a \rangle|^2 = \sum_{r=0}^{R} \sum_{s=1}^{\dim(\mathcal{H}_{2r+\beta}^d)} |\langle \mu_{\boldsymbol{z}}, T_\psi Y_{2r+\beta,s} \rangle|^2.$$

By Lemmas 20 and 21, $T_\psi Y_{2r+\beta,s} = c_{2r+\beta,d} Y_{2r+\beta,s}$, where $c_{2r+\beta} \in \mathbb{R}$ and

$$|c_{2r+\beta,d}| \gtrsim \begin{cases} \frac{\Gamma\left(\frac{d}{2}\right)\Gamma\left(\frac{2R+1}{2}\right)}{\Gamma\left(\frac{d+2R+1}{2}\right)} & \text{if } \psi = \dot{\sigma} \\ \frac{\sqrt{d}\Gamma\left(\frac{d}{2}\right)\Gamma\left(\frac{2R-1}{2}\right)}{\Gamma\left(\frac{d+2R+1}{2}\right)} & \text{if } \psi = \sqrt{d}\sigma. \end{cases} \tag{15}$$

Hence

$$\begin{aligned}
\|T_\psi \mu_{\boldsymbol{z}}\|^2 &\geq \sum_{r=0}^{R} \sum_{s=1}^{\dim(\mathcal{H}_{2r+\beta}^d)} |c_{2r+\beta,d}|^2 |\langle \mu_{\boldsymbol{z}}, Y_{2r+\beta,s} \rangle|^2 \\
&\geq \min_{0 \leq r \leq R} \left(|c_{2r+\beta,d}|^2\right) \sum_{r=0}^{R} \sum_{s=1}^{\dim(\mathcal{H}_{2r+\beta}^d)} |\langle \mu_{\boldsymbol{z}}, Y_{2r+\beta,s} \rangle|^2 \\
&= \min_{0 \leq r \leq R} \left(|c_{2r+\beta,d}|^2\right) \sum_{a=1}^{N} |\langle \mu_{\boldsymbol{z}}, g_a \rangle|^2 \\
&= \min_{0 \leq r \leq R} \left(|c_{2r+\beta,d}|^2\right) \sum_{a=1}^{N} \left|\sum_{i=1}^{n} z_i g_a(\boldsymbol{x}_i)\right|^2 \\
&= \min_{0 \leq r \leq R} \left(|c_{2r+\beta,d}|^2\right) \|\boldsymbol{B}\boldsymbol{z}\|^2 \\
&\geq \min_{0 \leq r \leq R} \left(|c_{2r+\beta,d}|^2\right) \sigma_{\min}(\boldsymbol{B})^2 \\
&\geq \frac{N}{2} \min_{0 \leq r \leq R} \left(|c_{2r+\beta,d}|^2\right).
\end{aligned}$$

So by (15),

$$\|T_\psi \mu_{\boldsymbol{z}}\|^2 \gtrsim \begin{cases} \frac{N\Gamma\left(\frac{d}{2}\right)^2 \Gamma\left(\frac{2R+1}{2}\right)^2}{\Gamma\left(\frac{d+2R+1}{2}\right)^2} & \text{if } \psi = \dot{\sigma} \\ \frac{Nd^2\Gamma\left(\frac{d}{2}\right)^2 \Gamma\left(\frac{2R-1}{2}\right)^2}{\Gamma\left(\frac{d+2R+1}{2}\right)^2} & \text{if } \psi = d\sigma. \end{cases} \tag{16}$$

We now separately analyze the cases where $\psi = \dot{\sigma}$ and $\psi = d\sigma$.

**Case 1:** $\psi = \dot{\sigma}$. In this case

$$\begin{aligned}
\|T_\psi \mu_{\boldsymbol{z}}\|^2 &\gtrsim N \frac{\Gamma\left(\frac{d}{2}\right)^2 \Gamma\left(\frac{2R+1}{2}\right)^2}{\Gamma\left(\frac{d+2R+1}{2}\right)^2} \\
&= \binom{2R+d}{d-1} \cdot \frac{\Gamma\left(\frac{d}{2}\right)^2 \Gamma\left(\frac{2R+1}{2}\right)^2}{\Gamma\left(\frac{d+2R+1}{2}\right)^2} \\
&= \frac{\Gamma(2R+d+1)}{\Gamma(d)\Gamma(2R+2)} \cdot \frac{\Gamma\left(\frac{d}{2}\right)^2 \Gamma\left(\frac{2R+1}{2}\right)^2}{\Gamma\left(\frac{d+2R+1}{2}\right)^2}.
\end{aligned}$$

Then by Stirling's approximation,

$$\|T_\psi\mu_{\mathbf{z}}\|^2 \gtrsim \frac{(2R+d+1)^{2R+d+1/2}e^{-2R-d-1}}{d^{d-1/2}e^{-d}(2R+2)^{2R+3/2}e^{-2R-2}} \cdot \frac{\left(\frac{d}{2}\right)^{d-1}e^{-d}\left(\frac{2R+1}{2}\right)^{2R}e^{-2R-1}}{\left(\frac{d+2R+1}{2}\right)^{d+2R}e^{-d-2R-1}}$$

$$\gtrsim (d+2R+1)^{1/2}d^{-1/2}\left(\frac{2R+1}{2R+2}\right)^{2R}(2R+2)^{-3/2}$$

$$\gtrsim (d+2R+1)^{1/2}d^{-1/2}(2R+2)^{-3/2}$$

$$\gtrsim (d+R)^{1/2}d^{-1/2}R^{-3/2}.$$

Here the third inequality follows from the observations

$$\left(\frac{2R+1}{2R+2}\right)^{2R} > 0$$

and

$$\lim_{R\to\infty}\left(\frac{2R+1}{2R+2}\right)^{2R} = \lim_{R\to\infty}\left(1-\frac{1}{2R+2}\right)^{2R} = e^{-1}.$$

**Case 2:** $\psi = \sqrt{d}\sigma$. In this case

$$\|T_\psi\mu_{\mathbf{z}}\|^2 \gtrsim N\frac{d\Gamma\left(\frac{d}{2}\right)^2\Gamma\left(\frac{2R-1}{2}\right)^2}{\Gamma\left(\frac{d+2R+1}{2}\right)^2}$$

$$= \binom{2R+d-1}{d-1}\cdot\frac{d\Gamma\left(\frac{d}{2}\right)^2\Gamma\left(\frac{2R-1}{2}\right)^2}{\Gamma\left(\frac{d+2R+1}{2}\right)^2}$$

$$= \frac{\Gamma(2R+d)}{\Gamma(d)\Gamma(2R+1)}\cdot\frac{d\Gamma\left(\frac{d}{2}\right)^2\Gamma\left(\frac{2R-1}{2}\right)^2}{\Gamma\left(\frac{d+2R+1}{2}\right)^2}.$$

Then by Stirling's approximation,

$$\|T_\psi\mu_{\mathbf{z}}\|^2 \gtrsim \frac{(2R+d)^{2R+d-1/2}e^{-2R-d}}{d^{d-1/2}e^{-d}(2R+1)^{2R+1/2}e^{-2R-1}} \cdot \frac{d\left(\frac{d}{2}\right)^{d-1}e^{-d}\left(\frac{2R-1}{2}\right)^{2R-2}e^{-2R+1}}{\left(\frac{d+2R+1}{2}\right)^{d+2R}e^{-d-2R-1}}$$

$$\gtrsim (d+2R)^{-1/2}\left(\frac{d+2R}{d+2R+1}\right)^{d+2R}d^{1/2}(2R-1)^{-2}(2R+1)^{1/2}\left(\frac{2R-1}{2R+1}\right)^{2R}$$

$$\gtrsim (d+2R)^{-1/2}d^{1/2}R^{-3/2}\left(\frac{d+2R}{d+2R+1}\right)^{d+2R}\left(\frac{2R-1}{2R+1}\right)^{2R}$$

$$= (d+2R)^{-1/2}d^{1/2}\left(1-\frac{1}{d+2R+1}\right)^{d+2R}\left(1-\frac{2}{2R+1}\right)^{2R}$$

$$\gtrsim (d+R)^{-1/2}d^{1/2}R^{-3/2}.$$

Hence we have established the desired bound on $\|T_\psi\mu_{\mathbf{z}}\|^2$ in all cases. $\qquad\square$

**Lemma 7.** *Let $d \geq 3$ and suppose that $\mathbf{x}_1,\cdots,\mathbf{x}_n \in \mathbb{S}^{d-1}$ are $\delta$-separated. For all $\mathbf{z} \in \mathbb{R}^n$ with $\|\mathbf{z}\| \leq 1$ then*

$$\|T_\psi\mu_{\mathbf{z}}\|^2 \gtrsim \begin{cases} \left(1+\frac{d\log(1/\delta)}{\log d}\right)^{-3}\delta^2 & \text{if } \psi = \dot{\sigma} \\ \left(1+\frac{d\log(1/\delta)}{\log d}\right)^{-3}\delta^4 & \text{if } \psi = \sqrt{d}\sigma. \end{cases}$$

*Proof.* We will consider multiple cases depending on the relative scaling of $d$ and $n$. Let $C > 0$ be the same constant as in Lemma 24. First suppose that $d \geq C\left(\frac{\delta^4}{2}\right)^{-(d-2)/2}$. Let $R = 1$. Then

$$\binom{2R+d-1}{d-1} = d \geq C\left(\frac{\delta^4}{2}\right)^{(d-2)/2}.$$

By Lemma 24, $\|T_\psi \mu_{\mathbf{z}}\|^2 \gtrsim 1$ in this case.

Next suppose that $d \leq C\left(\frac{\delta^4}{2}\right)^{-(d-2)/2}$ and $\sqrt{d}\log d \geq (8\log(1+C) + 16d)\log\frac{2}{\delta}$. Let

$$R = \left\lceil \frac{\log(1+C) + 2d\log(2/\delta)}{\log d} \right\rceil.$$

Note that since $d \leq \left(\frac{\delta^4}{2}\right)^{-(d-2)/2}$, we have

$$\frac{\log(1+C) + 2d\log(2/\delta)}{\log d} \geq \frac{2d\log(2/\delta)}{\frac{d-2}{2}\log(2/\delta^4)} \geq 1$$

and therefore

$$R \leq \frac{2\log(1+C) + 4d\log(2/\delta)}{\log d} \leq \frac{\sqrt{d}}{4}.$$

By definition,

$$R \geq \frac{\log(1+C) + 2d\log(2/\delta)}{\log(d)}$$

so that

$$\binom{2R+d-1}{d-1} \geq \left(\frac{2R+d-1}{2R}\right)^{2R}$$

$$\geq \left(\frac{d}{2R}\right)^{2R}$$

$$= \exp\left(2R(\log(d) - \log(2R))\right)$$

$$\geq \exp\left(2R\left(\log(d) - \log\left(\sqrt{d}\right)\right)\right)$$

$$= \exp\left(R\log d\right)$$

$$\geq \exp(\log(1+C) + 2d\log(2/\delta))$$

$$\geq C\left(\frac{2}{\delta}\right)^{2d}$$

$$\geq C\left(\frac{2}{\delta^4}\right)^{(d-2)/2}.$$

Then by Lemma 24, the following bounds hold. If $\psi = \dot\sigma$, then

$$\|T_\psi \mu_{\mathbf{z}}\|^2 \gtrsim (d+R)^{1/2}d^{-1/2}R^{-3/2}$$

$$\gtrsim R^{-3/2}$$

$$\gtrsim \left(1 + \frac{d\log(1/\delta)}{\log d}\right)^{-3/2}$$

$$\gtrsim \left(1 + \frac{d\log(1/\delta)}{\log d}\right)^{-3}\delta^2.$$

If $\psi = \sqrt{d}\sigma$, then

$$\|T_\psi \mu_{\mathbf{z}}\|^2 \gtrsim (d+R)^{-1/2}d^{1/2}R^{-3/2}$$

$$\gtrsim (d+\sqrt{d})^{-1/2}d^{1/2}R^{-3/2}$$

$$\gtrsim R^{-3/2}$$

$$\gtrsim \left(1 + \frac{d\log(2/\delta)}{\log d}\right)^{-3/2}$$

$$\gtrsim \left(1 + \frac{\log(n/\epsilon)}{\log d}\right)^{-3}\delta^4.$$

Finally suppose that $\sqrt{d}\log d \leq (8\log(1+C)+16d)\log\frac{2}{\delta}$ and let $R = \left\lceil (1+2C)d\left(\frac{2}{\delta}\right)^{2(d-2)/(d-1)}\right\rceil$. Then

$$R \lesssim 1+d\left(\frac{2}{\delta}\right)^{2(d-2)/(d-1)}$$

$$\leq (1+d)\left(\frac{2}{\delta}\right)^{2(d-2)/(d-1)}$$

$$\leq \left(1+\sqrt{d}\right)^2\left(\frac{2}{\delta}\right)^{2(d-2)/(d-1)}$$

$$\lesssim \left(1+\frac{d\log(1/\delta)}{\log(d)}\right)^2\left(\frac{2}{\delta}\right)^{2(d-2)/(d-1)}$$

$$\lesssim \left(1+\frac{d\log(1/\delta)}{\log(d)}\right)^2\delta^{-2}$$

and

$$\binom{2R+d-1}{d-1} \geq \left(\frac{2R+d-1}{d-1}\right)^{d-1}$$

$$\geq \left(\frac{R}{d}\right)^{d-1}$$

$$\geq \left(1+\frac{2C}{d}\right)^{d-1}\left(\frac{2}{\delta}\right)^{2/(d-2)}.$$

$$\geq \frac{2C(d-1)}{d}\left(\frac{2}{\delta}\right)^{2/(d-2)}$$

$$\geq C\left(\frac{2}{\delta}\right)^{2/(d-2)}.$$

So by Lemma 24 the following bounds hold. If $\psi=\dot{\sigma}$, then

$$\|T_\psi\mu_{\mathbf{z}}\|^2 \gtrsim (d+R)^{1/2}d^{-1/2}R^{-3/2}$$

$$\gtrsim (1+d)^{-1/2}R^{-1}$$

$$\gtrsim \left(1+\frac{d\log(1/\delta)}{\log d}\right)^{-1}\left(1+\frac{d\log(1/\delta)}{\log d}\right)^{-2}\delta^2$$

$$= \left(1+\frac{d\log(1/\delta)}{\log d}\right)^{-3}\delta^2.$$

If $\psi=\sqrt{d}\sigma$, then

$$\|T_\psi\mu_{\mathbf{z}}\|^2 \gtrsim (d+R)^{-1/2}d^{1/2}R^{-3/2}$$

$$\gtrsim \left(d+d\left(\frac{2}{\delta}\right)^{2(d-2)/(d-1)}\right)^{-1/2}d^{1/2}\left(d\left(\frac{2}{\delta}\right)^{2(d-2)/(d-1)}\right)^{-3/2}$$

$$\gtrsim d^{-3/2}\left(1+\left(\frac{2}{\delta}\right)^{2(d-2)/(d-1)}\right)^{-1/2}\left(\frac{2}{\delta}\right)^{-3(d-2)/(d-1)}$$

$$\gtrsim (1+d)^{-3/2}\left(\frac{2}{\delta}\right)^{-4(d-2)/(d-1)}$$

$$\gtrsim \left(1+\frac{d\log(1/\delta)}{\log d}\right)\delta^4.$$

Hence we have shown the desired bound on $\|T_\psi\mu_{\mathbf{z}}\|^2$ in all cases. $\qquad\square$

## C.6 Upper bound on the minimum eigenvalue of the NTK

Our strategy to upper bound $\lambda_{\min}(\boldsymbol{K})$ will be to prove that if two data points $\boldsymbol{x}, \boldsymbol{x}'$ are close, then the Jacobian of the network does not separate points too much. We will need to find upper bounds for both $\|\sigma(\boldsymbol{W}\boldsymbol{x}) - \sigma(\boldsymbol{W}\boldsymbol{x}')\|$ and $\|\dot{\sigma}(\boldsymbol{W}\boldsymbol{x}) - \dot{\sigma}(\boldsymbol{W}\boldsymbol{x}')\|$.

**Lemma 25.** *Let $\epsilon \in (0, 1)$. Suppose that $\boldsymbol{x}, \boldsymbol{x}' \in \mathbb{S}^{d-1}$ with $\|\boldsymbol{x} - \boldsymbol{x}'\| = \delta$. If $d_1 = \Omega\left(\log \frac{1}{\epsilon}\right)$, then with probability at least $1 - \epsilon$,*

$$\|\sigma(\boldsymbol{W}\boldsymbol{x}) - \sigma(\boldsymbol{W}\boldsymbol{x}')\| \lesssim \delta\sqrt{d_1}.$$

*Proof.* Note that $\|\sigma(\boldsymbol{W}\boldsymbol{x}) - \sigma(\boldsymbol{W}\boldsymbol{x}')\|^2$ can be written a sum of iid subexponential random variables:

$$\|\sigma(\boldsymbol{W}\boldsymbol{x}) - \sigma(\boldsymbol{W}\boldsymbol{x}')\|^2 = \sum_{j=1}^{d_1}(\sigma(\langle \boldsymbol{w}_j, \boldsymbol{x}\rangle) - \sigma(\langle \boldsymbol{w}_j, \boldsymbol{x}'\rangle))^2.$$

Since the entries of each $\boldsymbol{w}_j$ are iid standard Gaussian random variables and $\sigma$ is 1-Lipschitz,

$$\begin{aligned}
\|(\sigma(\langle \boldsymbol{w}_j, \boldsymbol{x}\rangle) - \sigma(\langle \boldsymbol{w}_j, \boldsymbol{x}'\rangle))^2\|_{\psi_1} &= \|\sigma(\langle \boldsymbol{w}_j, \boldsymbol{x}\rangle) - \sigma(\langle \boldsymbol{w}_j, \boldsymbol{x}'\rangle)\|_{\psi_2}^2 \\
&\leq \|\langle \boldsymbol{w}_j, \boldsymbol{x} - \boldsymbol{x}'\rangle\|_{\psi_2}^2 \\
&= \|\boldsymbol{x} - \boldsymbol{x}'\|^2 \\
&= \delta^2.
\end{aligned}$$

Moreover,

$$\begin{aligned}
\mathbb{E}[(\sigma(\langle \boldsymbol{w}_j, \boldsymbol{x}\rangle) - \sigma(\langle \boldsymbol{w}_j, \boldsymbol{x}'\rangle))^2] &\leq \mathbb{E}[|\langle \boldsymbol{w}_j, \boldsymbol{x} - \boldsymbol{x}'\rangle|^2] \\
&= \|\boldsymbol{x} - \boldsymbol{x}'\|^2 \\
&= \delta^2.
\end{aligned}$$

So by Bernstein's inequality, for all $t \geq 0$

$$\begin{aligned}
&\mathbb{P}\left(\|\sigma(\boldsymbol{W}\boldsymbol{x}) - \sigma(\boldsymbol{W}\boldsymbol{x}')\|^2 \geq \delta^2 d_1 + t\right) \\
&\leq \mathbb{P}\left(\|\sigma(\boldsymbol{W}\boldsymbol{x}) - \sigma(\boldsymbol{W}\boldsymbol{x}')\|^2 \geq \mathbb{E}[\|\sigma(\boldsymbol{W}\boldsymbol{x}) - \sigma(\boldsymbol{W}\boldsymbol{x}')\|^2] + t\right) \\
&\leq 2\exp\left(-C\min\left(\frac{t^2}{d_1\delta^4}, \frac{t}{\delta^2}\right)\right)
\end{aligned}$$

where $C > 0$ is a universal constant. Setting $t = \delta^2 d_1$ with $d_1 \geq \frac{1}{C}\log\frac{2}{\epsilon}$ yields

$$\mathbb{P}(\|\sigma(\boldsymbol{W}\boldsymbol{x}) - \sigma(\boldsymbol{W}\boldsymbol{x}')\|^2 \geq 2\delta^2 d_1) \leq 2\exp(-Cd_1) \leq \epsilon.$$

This establishes the result. $\qquad\square$

**Lemma 26.** *Suppose that $\boldsymbol{x}, \boldsymbol{x}' \in \mathbb{S}^{d-1}$. If $\boldsymbol{w} \sim \mathcal{N}(\boldsymbol{0}, \boldsymbol{I}_d)$, then*

$$\mathbb{P}(\dot{\sigma}(\langle \boldsymbol{w}, \boldsymbol{x}\rangle) \neq \dot{\sigma}(\langle \boldsymbol{w}, \boldsymbol{x}'\rangle)) \asymp \|\boldsymbol{x} - \boldsymbol{x}'\|.$$

*Proof.* Recall that for $\boldsymbol{x}, \boldsymbol{x}' \in \mathbb{S}^{d-1}$,

$$\mathbb{P}(\dot{\sigma}(\langle \boldsymbol{w}, \boldsymbol{x}\rangle) \neq \dot{\sigma}(\langle \boldsymbol{w}, \boldsymbol{x}'\rangle)) = \frac{\theta}{\pi},$$

where $\theta$ is the angle formed by $\boldsymbol{x}$ and $\boldsymbol{x}'$; that is, $\theta \in [0, \pi]$ with

$$\cos(\theta) = \langle \boldsymbol{x}, \boldsymbol{x}'\rangle = 1 - \frac{1}{2}\|\boldsymbol{x} - \boldsymbol{x}'\|^2.$$

By Taylor's theorem, $1 - \cos(\theta) = \frac{1}{2}\theta^2 + O(\theta^3)$, so $1 - \cos(\theta) \asymp \theta^2$ for $\theta \in [0, \pi]$. This implies that $\theta^2 \asymp \|\boldsymbol{x} - \boldsymbol{x}'\|^2$, so $\theta \asymp \|\boldsymbol{x} - \boldsymbol{x}'\|$ and therefore

$$\mathbb{P}(\dot{\sigma}(\langle \boldsymbol{w}, \boldsymbol{x}\rangle) \neq \dot{\sigma}(\langle \boldsymbol{w}, \boldsymbol{x}'\rangle)) \asymp \|\boldsymbol{x} - \boldsymbol{x}'\|.$$

$\qquad\square$

**Lemma 27.** *Let $\epsilon \in (0,1)$. Suppose that $\boldsymbol{x}, \boldsymbol{x}' \in \mathbb{S}^{d-1}$ with $\|\boldsymbol{x} - \boldsymbol{x}'\| \leq \delta$. If $d_1 = \Omega\left(\frac{1}{\delta} \log \frac{1}{\epsilon}\right)$, then with probability at least $1 - \epsilon$,*

$$\|\boldsymbol{v} \odot \dot{\sigma}(\boldsymbol{W}\boldsymbol{x}) - \boldsymbol{v} \odot \dot{\sigma}(\boldsymbol{W}\boldsymbol{x})\| \lesssim \sqrt{\delta d_1}.$$

*Proof.* Observe that

$$\|\dot{\sigma}(\boldsymbol{W}\boldsymbol{x}) - \dot{\sigma}(\boldsymbol{W}\boldsymbol{x}')\|^2 = 4 \sum_{j=1}^{d_1} Z_j = 4|\mathcal{S}|,$$

where $Z_j \in \{0,1\}$ is equal to 1 if

$$\dot{\sigma}(\langle \boldsymbol{w}_j, \boldsymbol{x}\rangle) \neq \dot{\sigma}(\langle \boldsymbol{w}_j, \boldsymbol{x}'\rangle)$$

and 0 otherwise, and $\mathcal{S}$ consists of the $j \in [d_1]$ such that $Z_j = 1$. The $Z_j$ are iid Bernoulli random variables with parameter $p$, where $p \asymp \delta$ by Lemma 26. By Chernoff's inequality (see Vershynin, 2018, Theorem 2.3.1), for all $t \geq d_1 p$

$$\mathbb{P}\left(|\mathcal{S}| \geq t\right) \leq e^{-d_1 p} \left(\frac{e d_1 p}{t}\right)^t$$

Then setting $t = e d_1 p$ with $d_1 \geq \frac{1}{p} \log \frac{4}{\epsilon}$ yields

$$\mathbb{P}\left(|\mathcal{S}| \geq e d_1 \delta\right) \leq \mathbb{P}\left(|\mathcal{S}| \geq e d_1 p\right)$$
$$\leq e^{-d_1 p}$$
$$\leq \frac{\epsilon}{4}.$$

By the lower bound of Chernoff's inequality, for all $t \leq d_1 p$

$$\mathbb{P}(|\mathcal{S}| \leq t) \leq e^{-d_1 p}\left(\frac{e d_1 p}{t}\right)^t.$$

Then setting $t = \frac{d_1 p}{e}$ with $d_1 \geq \frac{2}{e-2}\frac{1}{p}\log\frac{4}{\epsilon}$ yields

$$\mathbb{P}\left(|\mathcal{S}| \leq \frac{d_1 p}{2}\right) \leq \exp\left(-\frac{e-2}{e} d_1 p\right)$$
$$\leq \frac{\epsilon}{4}.$$

Therefore, with probability at least $1 - \frac{\epsilon}{2}$,

$$\frac{d_1 \delta}{e} \leq |\mathcal{S}| \leq e d_1 \delta.$$

Let us denote this event by $\omega$. Observe that

$$\|\boldsymbol{v} \odot \dot{\sigma}(\boldsymbol{W}\boldsymbol{x}) - \boldsymbol{v} \odot \dot{\sigma}(\boldsymbol{W}\boldsymbol{x}')\|^2 = 2 \sum_{j \in \mathcal{S}} v_j^2$$

and recall that $v_j^2 \sim \mathcal{N}(0,1)$ for all $j \in [d_1]$. By Bernstein's inequality, for all $t \geq 0$

$$\mathbb{P}\left(\frac{1}{2}\|\boldsymbol{v} \odot \dot{\sigma}(\boldsymbol{W}\boldsymbol{x}) - \boldsymbol{v} \odot \dot{\sigma}(\boldsymbol{W}\boldsymbol{x}')\|^2 \geq |\mathcal{S}| + t \ \bigg| \ \mathcal{S}\right) \leq 2\exp\left(-C_1 \min\left(\frac{t^2}{|\mathcal{S}|}, t\right)\right)$$

where $C_1 > 0$ is a universal constant. Setting $t = |\mathcal{S}|$ yields

$$\mathbb{P}\left(\|\boldsymbol{v} \odot \dot{\sigma}(\boldsymbol{W}\boldsymbol{x}) - \boldsymbol{v} \odot \dot{\sigma}(\boldsymbol{W}\boldsymbol{x}')\| \geq 2\sqrt{|\mathcal{S}|} \ \bigg| \ \mathcal{S}\right) \leq 2\exp\left(-C_1 |\mathcal{S}|\right).$$

Then

$$\mathbb{P}\left(\|\boldsymbol{v}\odot\dot{\sigma}(\boldsymbol{W}\boldsymbol{x})-\boldsymbol{v}\odot\dot{\sigma}(\boldsymbol{W}\boldsymbol{x}')\|\leq 2\sqrt{ed_1\delta}\right)$$

$$\geq\mathbb{P}\left(\|\boldsymbol{v}\odot\dot{\sigma}(\boldsymbol{W}\boldsymbol{x})-\boldsymbol{v}\odot\dot{\sigma}(\boldsymbol{W}\boldsymbol{x}')\|\leq 2\sqrt{ed_1\delta},\ \omega\right)$$

$$\geq\mathbb{P}\left(\|\boldsymbol{v}\odot\dot{\sigma}(\boldsymbol{W}\boldsymbol{x})-\boldsymbol{v}\odot\dot{\sigma}(\boldsymbol{W}\boldsymbol{x}')\|\leq 2\sqrt{|\mathcal{S}|},\ \omega\right)$$

$$\geq\mathbb{E}\left[\mathbb{P}\left(\|\boldsymbol{v}\odot\dot{\sigma}(\boldsymbol{W}\boldsymbol{x})-\boldsymbol{v}\odot\dot{\sigma}(\boldsymbol{W}\boldsymbol{x}')\|\leq 2\sqrt{|\mathcal{S}|}\ \Big|\ \mathcal{S}\right)1_\omega\right]$$

$$\geq\mathbb{E}\left[(1-2\exp\left(-C_1|\mathcal{S}|\right))1_\omega\right]$$

$$\geq\left(1-2\exp\left(-C_1\frac{d_1\delta}{e}\right)\right)\mathbb{P}(\omega)$$

$$\geq\left(1-2\exp\left(-C_1\frac{d_1\delta}{e}\right)\right)\left(1-\frac{\epsilon}{2}\right),$$

where we used that $\omega$ is measurable with respect to $\mathcal{S}$ in the fourth line. So if $d_1\geq\frac{e}{C_1\delta}\log\frac{4}{\epsilon}$, then

$$\mathbb{P}\left(\|\boldsymbol{v}\odot\dot{\sigma}(\boldsymbol{W}\boldsymbol{x})-\boldsymbol{v}\odot\dot{\sigma}(\boldsymbol{W}\boldsymbol{x}')\|\leq 2\sqrt{ed_1\delta}\right)\geq\left(1-\frac{\epsilon}{2}\right)\left(1-\frac{\epsilon}{2}\right)$$

$$\geq 1-\epsilon.$$

$\square$

**Lemma 28.** *Suppose that $\boldsymbol{x}\in\mathbb{S}^{d-1}$. If $d_1=\Omega\left(\log\frac{1}{\epsilon}\right)$, then with probability at least $1-\epsilon$,*

$$\|\boldsymbol{v}\odot\dot{\sigma}(\boldsymbol{W}\boldsymbol{x})\|\lesssim\sqrt{d_1}.$$

*Proof.* Since $\dot{\sigma}(\langle\boldsymbol{w}_j,\boldsymbol{x}\rangle)\in\{0,1\}$ for all $j\in[d_1]$,

$$\|\boldsymbol{v}\odot\dot{\sigma}(\boldsymbol{W}\boldsymbol{x})\|^2=\sum_{j=1}^{d_1}v_j^2\dot{\sigma}(\langle\boldsymbol{w}_j,\boldsymbol{x}\rangle)$$

$$\leq\sum_{j=1}^{d_1}v_j^2.$$

Since the entries $v_j$ are iid standard Gaussian random variables, Bernstein's inequality implies for all $t\geq 0$

$$\mathbb{P}(\|\boldsymbol{v}\odot\dot{\sigma}(\boldsymbol{W}\boldsymbol{x})\|^2\geq d_1+t)\leq\mathbb{P}\left(\sum_{j=1}^{d_1}v_j^2\geq d_1+t\right)$$

$$\leq 2\exp\left(-C\min\left(\frac{t^2}{d_1},t\right)\right).$$

Setting $t=d_1$ with $d_1\geq\frac{1}{C}\log\frac{2}{\epsilon}$ yields

$$\mathbb{P}(\|\boldsymbol{v}\odot\dot{\sigma}(\boldsymbol{W}\boldsymbol{x})\|^2\geq 2d_1)\leq 2\exp(-Cd_1)\leq\epsilon.$$

$\square$

Now we prove our main lemma which we will use to relate the separation between data points to the NTK.

**Lemma 29.** *Let $\boldsymbol{x},\boldsymbol{x}'\in\mathbb{S}^{d-1}$ with $\|\boldsymbol{x}-\boldsymbol{x}'\|\leq\delta\leq 2$. Let $\epsilon\in(0,1)$. If $d_1=\Omega\left(\frac{1}{\delta}\log\frac{1}{\epsilon}\right)$, then with probability at least $1-\epsilon$,*

$$\|\nabla_{\boldsymbol{\theta}}f(\boldsymbol{x})-\nabla_{\boldsymbol{\theta}}f(\boldsymbol{x}')\|\lesssim\sqrt{\delta}.$$

*Proof.* By Lemma 27, if $d_1 \gtrsim \frac{1}{\delta} \log \frac{1}{\epsilon}$, then with probability at least $1 - \frac{\epsilon}{4}$,

$$\|\boldsymbol{v} \odot \dot{\sigma}(\boldsymbol{W}\boldsymbol{x}) - \boldsymbol{v} \odot \dot{\sigma}(\boldsymbol{W}\boldsymbol{x}')\| \lesssim \sqrt{\delta d_1}.$$

Let us denote this event by $\omega_1$. By Lemma 28, if $d_1 \gtrsim \log \frac{1}{\epsilon}$, then with probability at least $1 - \frac{\epsilon}{4}$,

$$\|\boldsymbol{v} \odot \dot{\sigma}(\boldsymbol{W}\boldsymbol{x})\| \lesssim \sqrt{d_1}.$$

Let us denote this event by $\omega_2$. If both $\omega_1$ and $\omega_2$ occur, then

$$
\begin{aligned}
&\|\nabla_{\boldsymbol{W}_1} f(\boldsymbol{x}) - \nabla_{\boldsymbol{W}_1} f(\boldsymbol{x}')\|_F \\
&= \frac{1}{\sqrt{d_1}} \|(\boldsymbol{v} \odot \dot{\sigma}(\boldsymbol{W}\boldsymbol{x})) \otimes \boldsymbol{x} - (\boldsymbol{v} \odot \dot{\sigma}(\boldsymbol{W}\boldsymbol{x}')) \otimes \boldsymbol{x}'\|_F \\
&\leq \frac{1}{\sqrt{d_1}} \|(\boldsymbol{v} \odot \dot{\sigma}(\boldsymbol{W}\boldsymbol{x})) \otimes \boldsymbol{x} - (\boldsymbol{v} \odot \dot{\sigma}(\boldsymbol{W}\boldsymbol{x})) \otimes \boldsymbol{x}'\|_F \\
&\quad + \frac{1}{\sqrt{d_1}} \|(\boldsymbol{v} \odot \dot{\sigma}(\boldsymbol{W}\boldsymbol{x})) \otimes \boldsymbol{x}' - (\boldsymbol{v} \odot \dot{\sigma}(\boldsymbol{W}\boldsymbol{x}')) \otimes \boldsymbol{x}'\|_F \\
&\leq \frac{1}{\sqrt{d_1}} \|\boldsymbol{v} \odot \dot{\sigma}(\boldsymbol{W}\boldsymbol{x})\| \cdot \|\boldsymbol{x} - \boldsymbol{x}'\| + \frac{1}{\sqrt{d_1}} \|\boldsymbol{v} \odot \dot{\sigma}(\boldsymbol{W}\boldsymbol{x}) - \boldsymbol{v} \odot \dot{\sigma}(\boldsymbol{W}\boldsymbol{x}')\| \cdot \|\boldsymbol{x}'\| \\
&\lesssim \frac{1}{\sqrt{d_1}} \sqrt{d_1} \delta + \frac{1}{\sqrt{d_1}} \sqrt{\delta d_1} \\
&\lesssim \sqrt{\delta}.
\end{aligned}
$$

By Lemma 25, if $d_l \gtrsim \log \frac{1}{\epsilon}$, then with probability at least $1 - \frac{\epsilon}{2}$,

$$
\|\nabla_{\boldsymbol{W}_2} f(\boldsymbol{x}) - \nabla_{\boldsymbol{W}_2} f(\boldsymbol{x}')\| = \frac{1}{\sqrt{d_1}} \|f_1(\boldsymbol{x}) - f_1(\boldsymbol{x}')\|
$$
$$
\lesssim \delta.
$$

Let us denote this event by $\omega_3$. If $\omega_1, \omega_2$, and $\omega_3$ all occur (which happens with probability at least $1 - \epsilon$), then

$$
\begin{aligned}
\|\nabla_{\boldsymbol{\theta}} f(\boldsymbol{x}) - \nabla_{\boldsymbol{\theta}} f(\boldsymbol{x}')\| &\lesssim \|\nabla_{\boldsymbol{W}_1} f(\boldsymbol{x}) - \nabla_{\boldsymbol{W}_1} f(\boldsymbol{x}')\|_F + \|\nabla_{\boldsymbol{W}_2} f(\boldsymbol{x}) - \nabla_{\boldsymbol{W}_2} f(\boldsymbol{x}')\| \\
&\lesssim \sqrt{\delta} + \delta \\
&\lesssim \sqrt{\delta}.
\end{aligned}
$$

$\square$

### C.7 Proof of Theorem 1

**Theorem 1.** *Let $d \geq 3$, $\epsilon \in (0, 1)$, and $\delta, \delta' \in (0, \sqrt{2})$. Suppose that $\boldsymbol{x}_1, \cdots, \boldsymbol{x}_n \in \mathbb{S}^{d-1}$ are $\delta$-separated and $\min_{i \neq k} \|\boldsymbol{x}_i - \boldsymbol{x}_k\| \leq \delta'$. Define*

$$\lambda = \left(1 + \frac{d \log(1/\delta)}{\log(d)}\right)^{-3} \delta^2.$$

*If $d_1 \gtrsim \frac{\|\boldsymbol{X}\|^2}{\lambda} \log \frac{n}{\epsilon}$, then with probability at least $1 - \epsilon$,*

$$\lambda \lesssim \lambda_{\min}(\boldsymbol{K}) \lesssim \delta'.$$

*Proof.* First we prove the lower bound. Let $\lambda_1$ be as it is defined in Lemma 3. By Lemma 5,

$$\lambda_1 = \inf_{\|\boldsymbol{z}\|=1} \|T_{\dot{\sigma}} \mu_{\boldsymbol{z}}\|^2.$$

Let

$$\lambda = \left(1 + \frac{d \log(1/\delta)}{\log(d)}\right)^{-3} \delta^2.$$

By Lemma 7, $\lambda_1 \geq C_1 \lambda$ for some constant $C_1 > 0$. By Lemma 3, there exist constants $C_2, C_3 > 0$ such that if $d_1 \geq \frac{C_2}{\lambda_1} \|X\|^2 \log \frac{n}{\epsilon}$ then

$$\mathbb{P}(\lambda_{\min}(K_1) < C_3 \lambda_1) \leq \frac{\epsilon}{2}. \tag{17}$$

Then for such $d_1$,

$$\mathbb{P}(\lambda_{\min}(K_1) \geq C_3 C_1 \lambda) \geq 1 - \frac{\epsilon}{2}.$$

This establishes the lower bound.

Next we prove the upper bound. Let $i, k \in [n]$ be two indices with $i \neq k$ such that $\|x_i - x_k\| \leq \delta'$. If $d_1 \gtrsim \frac{1}{\lambda} \log \frac{1}{\epsilon} \gtrsim \frac{1}{\delta'} \log \frac{1}{\epsilon}$, then by Lemma 29 there exists $C_4 > 0$ such that

$$\mathbb{P}(\|\nabla_{\boldsymbol{\theta}} f(x_i) - \nabla_{\boldsymbol{\theta}} f(x_k)\|^2 \geq C_4 \delta') \geq 1 - \frac{\epsilon}{2}.$$

Let us denote this event by $\omega$. If $\omega$ occurs, then

$$\begin{aligned}
\lambda_{\min}(K) &\lesssim (e_i - e_k)^T K (e_i - e_k) \\
&= \|\nabla_{\boldsymbol{\theta}} f(x) - \nabla_{\boldsymbol{\theta}} f(x_k)\|^2 \\
&\lesssim \delta'.
\end{aligned}$$

Hence, with probability at least $1 - \frac{\epsilon}{2}$, $\lambda_{\min}(K) \lesssim \delta'$. This establishes the upper bound for the minimum eigenvalue. The two-sided bound then immediately follows from a union bound. $\qquad \square$

## C.8 Uniform data on a sphere

Our main bounds for the smallest eigenvalue of the NTK are stated in terms of the amount of separation between data points. To interpret our results in terms of probability distributions on the sphere, we will use a couple of lemmas which quantify the amount of separation for data which is uniformly distributed.

For $\delta \in (0, 1/2)$ and $x \in \mathbb{S}^{d-1}$, we define the spherical cap

$$\mathrm{Cap}(x, \delta) = \{y \in \mathbb{S}^{d-1} : \|y - x\| \leq \delta\}.$$

and the double spherical cap

$$\mathrm{DoubleCap}(x, \delta) = \mathrm{Cap}(x, \delta) \cup \mathrm{Cap}(-x, \delta).$$

By Lemma 2.3 of Ball (1997),

$$dS(\mathrm{Cap}(x, \delta)) \geq \frac{1}{2} \left(\frac{\delta}{2}\right)^{d-1}. \tag{18}$$

We can also obtain a corresponding upper bound on the volume of a spherical cap.

**Lemma 30.** *For $x \in \mathbb{S}^{d-1}$ and $\delta \in (0, 1/2)$,*

$$dS(\mathrm{Cap}(x, \delta)) \leq \frac{4\sqrt{\pi}(C\delta)^{d-1}}{d^2}.$$

*Here $C > 0$ is a universal constant.*

*Proof.* For $\phi \in [0, \pi]$, let $\mathcal{S}_\phi$ denote the set of all $x' \in \mathbb{S}^{d-1}$ such that the angle between $x$ and $x'$ is at most $\phi$ (that is, $\langle x, x' \rangle \geq \cos(\phi)$). The measure of $\mathcal{S}_\phi$ is given by

$$\frac{B(\sin^2(\phi); (d-1)/2, 1/2)}{B((d-1)/2, 1/2)}$$

(see, e.g. Li, 2010). Here the numerator refers to the incomplete beta function and the denominator refers to the beta function. We can bound

$$\begin{aligned}
B\left(\sin^2(\phi); \frac{d-1}{2}, \frac{1}{2}\right) &= \int_0^{\sin^2(\phi)} t^{(d-3)/2}(1-t)^{-1/2} dt \\
&\leq \int_0^{\sin^2(\phi)} t^{(d-3)/2} dt \\
&= \frac{2}{d-1} \sin(\phi)^{d-1}.
\end{aligned}$$

and

$$B\left(\frac{d-1}{2},\frac{1}{2}\right) = \frac{\Gamma\left(\frac{d-1}{2}\right)\Gamma\left(\frac{1}{2}\right)}{\Gamma\left(\frac{d}{2}\right)}$$

$$\geq \frac{\Gamma\left(\frac{d-2}{2}\right)\sqrt{\pi}}{\Gamma\left(\frac{d}{2}\right)}$$

$$= \frac{2\sqrt{\pi}}{d-2}.$$

The above two bounds imply

$$dS(\mathcal{S}_\phi) \leq \frac{4\sqrt{\pi}\sin(\phi)^{d-1}}{(d-1)(d-2)} \leq \frac{4\sqrt{\pi}\sin(\phi)^{d-1}}{d^2} \leq \frac{4\sqrt{\pi}\phi^{d-1}}{d^2}. \tag{19}$$

Now suppose that $\boldsymbol{x}' \in \mathrm{Cap}(\boldsymbol{x},\delta)$. Then $\|\boldsymbol{x}-\boldsymbol{x}'\| \leq \delta$, so $1-\langle\boldsymbol{x},\boldsymbol{x}'\rangle \leq 2\delta^2$. Let $\phi = \arccos(\langle\boldsymbol{x},\boldsymbol{x}'\rangle)$ be the angle between $\boldsymbol{x}$ and $\boldsymbol{x}'$. By Taylor's theorem, $\cos(\phi) = 1 - \frac{\phi^2}{2} + O(\phi^3)$, so $1 - \cos(\phi) \asymp \phi^2$ for $\phi \in [0,\pi]$. Thus

$$2\delta^2 \geq 1 - \langle\boldsymbol{x},\boldsymbol{x}'\rangle = 1 - \cos(\phi) \asymp \phi^2.$$

So the angle between $\boldsymbol{x}$ and $\boldsymbol{x}'$ is at most $C\delta$ for some universal constant $C > 0$. It follows that $\mathrm{Cap}(\boldsymbol{x},\delta) \subseteq \mathcal{S}_{C\delta}$. Finally by (19),

$$dS(\mathrm{Cap}(\boldsymbol{x},\delta)) \leq \frac{4\sqrt{\pi}(C\delta)^{d-1}}{d^2}.$$

$\square$

Since $\delta \leq \frac{1}{2}$, the sets $\mathrm{Cap}(\boldsymbol{x},\delta)$ and $\mathrm{Cap}(-\boldsymbol{x},\delta)$ are disjoint by the triangle inequality. Hence

$$dS(\mathrm{DoubleCap}(\boldsymbol{x},\delta)) = 2\mathrm{Cap}(\boldsymbol{x},\delta)$$

and in particular by Lemma 30

$$dS(\mathrm{DoubleCap}(\boldsymbol{x},\delta)) \leq \frac{4\sqrt{\pi}(C\delta)^{d-1}}{d^2}. \tag{20}$$

for a constant $C > 0$.

**Lemma 31.** *Suppose that $n \geq 2$ and $\epsilon \in (0,1)$. If $\boldsymbol{x}_1,\cdots,\boldsymbol{x}_n \in \mathbb{S}^{d-1}$ are independent and uniformly distributed on $\mathbb{S}^{d-1}$, then with probability at least $1-\epsilon$, the dataset is $\delta$-separated with*

$$\delta \gtrsim \left(\frac{\epsilon}{n^2}\right)^{1/(d-1)}.$$

*Proof.* Let $\boldsymbol{e} = [1,0,\cdots,0]^T \in \mathbb{S}^{d-1}$. For each $\boldsymbol{x} \in \mathbb{S}^{d-1}$, there exists an orthogonal matrix $\boldsymbol{O}_{\boldsymbol{x}}$ such that $\boldsymbol{O}_{\boldsymbol{x}}\boldsymbol{x} = \boldsymbol{e}$. Note that for all $\boldsymbol{x} \in \mathbb{S}^{d-1}$ and $i \in [n]$, $\boldsymbol{O}_{\boldsymbol{x}}\boldsymbol{x}_i \overset{d}{=} \boldsymbol{x}_i$. Let $i,k \in [n]$ with $i \neq k$. Then for all $\delta \in (0,1/2)$,

$$\begin{aligned}
\mathbb{P}(\|\boldsymbol{x}_i - \boldsymbol{x}_k\| \leq \delta \text{ or } \|\boldsymbol{x}_i + \boldsymbol{x}_k\| \leq \delta) &= \mathbb{E}[\mathbb{P}(\|\boldsymbol{x}_i - \boldsymbol{x}_k\| \leq \delta \text{ or } \|\boldsymbol{x}_i + \boldsymbol{x}_k\| \leq \delta \mid \boldsymbol{x}_k)]\\
&= \mathbb{E}[\mathbb{P}(\|\boldsymbol{O}_{\boldsymbol{x}_k}\boldsymbol{x}_i - \boldsymbol{O}_{\boldsymbol{x}_k}\boldsymbol{x}_k\| \leq \delta \text{ or } \|\boldsymbol{O}_{\boldsymbol{x}_k}\boldsymbol{x}_i + \boldsymbol{O}_{\boldsymbol{x}_k}\boldsymbol{x}_k\| \leq \delta \mid \boldsymbol{x}_k)]\\
&= \mathbb{E}[\mathbb{P}(\|\boldsymbol{O}_{\boldsymbol{x}_k}\boldsymbol{x}_i - \boldsymbol{e}\| \leq \delta \text{ or } \|\boldsymbol{O}_{\boldsymbol{x}_k}\boldsymbol{x}_i + \boldsymbol{e}\| \leq \delta \mid \boldsymbol{x}_k)]\\
&= \mathbb{E}[\mathbb{P}(\|\boldsymbol{x}_i - \boldsymbol{e}\| \leq \delta \text{ or } \|\boldsymbol{x}_i + \boldsymbol{e}\| \leq \delta \mid \boldsymbol{x}_k)]\\
&= \mathbb{P}(\|\boldsymbol{x}_i - \boldsymbol{e}\| \leq \delta \text{ or } \|\boldsymbol{x}_i + \boldsymbol{e}\| \leq \delta).
\end{aligned}$$

The expression on the final line is the measure of $\mathrm{DoubleCap}(\boldsymbol{e},\delta)$, and by (20) is bounded above by

$$\frac{4\sqrt{\pi}(C\delta)^{d-1}}{d^2},$$

where $C > 0$ is a constant. So

$$\begin{aligned}
\mathbb{P}(\|\boldsymbol{x}_i - \boldsymbol{x}_k\| \leq \delta \text{ or } \|\boldsymbol{x}_i + \boldsymbol{x}_k\| \leq \delta \text{ for some } i \neq k) &\leq \sum_{i \neq k}\mathbb{P}(\|\boldsymbol{x}_i - \boldsymbol{x}_k\| \leq \delta \text{ or } \|\boldsymbol{x}_i + \boldsymbol{x}_k\| \leq \delta)\\
&\leq \frac{4\sqrt{\pi}n^2(C\delta)^{d-1}}{d^2}.
\end{aligned}$$

Setting $\delta = \min\left(\frac{1}{4}, \frac{1}{C}\left(\frac{\epsilon d^2}{4\sqrt{\pi}n^2}\right)^{1/(d-1)}\right)$, we obtain

$$\mathbb{P}(\|\boldsymbol{x}_i - \boldsymbol{x}_k\| \leq \delta \text{ or } \|\boldsymbol{x}_i + \boldsymbol{x}_k\| \leq \delta \text{ for some } i \neq k) \leq \epsilon.$$

Therefore, for this value of $\delta$, the dataset is $\delta$-separated with probability at least $1 - \epsilon$. To conclude, note that

$$\frac{1}{C}\left(\frac{\epsilon d^2}{4\sqrt{\pi}n^2}\right)^{1/(d-1)} \gtrsim \left(\frac{\epsilon}{n^2}\right)^{1/(d-1)}$$

since

$$\lim_{d\to\infty}\left(\frac{d^2}{4\sqrt{\pi}}\right)^{1/(d-1)} = 1.$$

$\square$

**Lemma 32.** *Suppose that $n \geq 2$ and $\epsilon \in (0,1)$. If $\boldsymbol{x}_1, \cdots, \boldsymbol{x}_n \in \mathbb{S}^{d-1}$ are selected iid from $U(\mathbb{S}^{d-1})$, then with probability at least $1 - \epsilon$, there exist $i, k \in [n]$ with $i \neq k$ such that*

$$\|\boldsymbol{x}_i - \boldsymbol{x}_k\| \lesssim \left(\frac{\log(1/\epsilon)}{n^2}\right)^{1/(d-1)}.$$

*Proof.* Let $\boldsymbol{e} = [1, 0, \cdots, 0]^T \in \mathbb{S}^{d-1}$. For each $\boldsymbol{x} \in \mathbb{S}^{d-1}$, there exists an orthogonal matrix $\boldsymbol{O}_x$ such that $\boldsymbol{O}_x \boldsymbol{x} = \boldsymbol{e}$. Note that for all $\boldsymbol{x} \in \mathbb{S}^{d-1}$ and $i \in [n]$, $\boldsymbol{O}_x \boldsymbol{x}_i \overset{d}{=} \boldsymbol{x}_i$. Let $i, k \in [n]$ with $i \neq k$. Then for all $\delta \in (0, 1/2)$,

$$\begin{aligned}
\mathbb{P}(\|\boldsymbol{x}_i - \boldsymbol{x}_k\| \leq \delta) &= \mathbb{E}[\mathbb{P}(\|\boldsymbol{x}_i - \boldsymbol{x}_k\| \leq \delta \mid \boldsymbol{x}_k)] \\
&= \mathbb{E}[\mathbb{P}(\|\boldsymbol{O}_{x_k}\boldsymbol{x}_i - \boldsymbol{O}_{x_k}\boldsymbol{x}_k\| \leq \delta \mid \boldsymbol{x}_k)] \\
&= \mathbb{E}[\mathbb{P}(\|\boldsymbol{O}_{x_k}\boldsymbol{x}_i - \boldsymbol{e}\| \leq \delta \mid \boldsymbol{x}_k)] \\
&= \mathbb{E}[\mathbb{P}(\|\boldsymbol{x}_i - \boldsymbol{e}\| \leq \delta \mid \boldsymbol{x}_k)] \\
&= \mathbb{P}(\|\boldsymbol{x}_i - \boldsymbol{e}\| \leq \delta).
\end{aligned}$$

The expression on the final line is the measure of $\text{Cap}(\boldsymbol{e}, \delta)$, and by Lemma 2.3 of Ball (1997) it is bounded below by $\frac{1}{2}\left(\frac{\delta}{2}\right)^{d-1}$. For each $i \in [n]$, let $\omega_i$ denote the event that $\|\boldsymbol{x}_j - \boldsymbol{x}_k\| > \delta$ for all $j, k \in [1, i]$ with $j \neq k$. Trivially $\mathbb{P}(\omega_1) = 1$. If $\omega_i$ occurs for some $i \in [1, n-1]$, then the sets $\text{Cap}(\boldsymbol{x}_j, \delta/2)$ for $j \in [i]$ are disjoint. Indeed, if $\boldsymbol{x} \in \text{Cap}(\boldsymbol{x}_j, \delta/2) \cap \text{Cap}(\boldsymbol{x}_k, \delta/2)$, then by the triangle inequality

$$\|\boldsymbol{x}_j - \boldsymbol{x}_k\| \leq \|\boldsymbol{x} - \boldsymbol{x}_j\| + \|\boldsymbol{x} - \boldsymbol{x}_k\| \leq \frac{\delta}{2} + \frac{\delta}{2} = \delta$$

which contradicts $\omega_i$. Now since these smaller spherical caps are disjoint, we can bound

$$\begin{aligned}
dS\left(\cup_{j=1}^i \{\boldsymbol{x} \in \mathbb{S}^{d-1} : \|\boldsymbol{x} - \boldsymbol{x}_j\| \leq \delta\}\right) &\geq dS\left(\cup_{j=1}^i \{\boldsymbol{x} \in \mathbb{S}^{d-1} : \|\boldsymbol{x} - \boldsymbol{x}_j\| \leq \delta/2\}\right) \\
&= dS\left(\cup_{j=1}^i \text{Cap}(\boldsymbol{x}_j, \delta/2)\right) \\
&= \sum_{j=1}^i dS(\text{Cap}(\boldsymbol{x}_j, \delta/2)) \\
&\geq \sum_{j=1}^i \frac{1}{2}\left(\frac{\delta}{4}\right)^{d-1} \\
&= \frac{i}{2}\left(\frac{\delta}{4}\right)^{d-1}.
\end{aligned}$$

Since $\boldsymbol{x}_{i+1}$ is chosen independently from $\boldsymbol{x}_1, \cdots, \boldsymbol{x}_i$, this implies

$$\begin{aligned}
\mathbb{P}(\omega_{i+1} \mid \omega_i) &= \mathbb{P}(\|\boldsymbol{x}_{i+1} - \boldsymbol{x}_j\| > \delta \ \forall j \in [i] \mid \omega_i) \\
&\leq 1 - \frac{i}{2}\left(\frac{\delta}{4}\right)^{d-1}.
\end{aligned}$$

By repeatedly conditioning we obtain

$$\mathbb{P}(\|\boldsymbol{x}_j - \boldsymbol{x}_k\| > \delta \ \ \forall j, k \in [n]) = \mathbb{P}(\omega_n)$$

$$= \mathbb{P}(\omega_1) \prod_{i=2}^{n} \mathbb{P}(\omega_i \mid \omega_1, \cdots, \omega_{i-1})$$

$$= \prod_{i=2}^{n} \mathbb{P}(\omega_i \mid \omega_{i-1})$$

$$\leq \prod_{i=2}^{n} \left(1 - \frac{i}{2}\left(\frac{\delta}{4}\right)^{d-1}\right)$$

$$\leq \prod_{i=2}^{n} \exp\left(-\frac{i}{2}\left(\frac{\delta}{4}\right)^{d-1}\right)$$

$$\leq \exp\left(-\frac{n^2}{2}\left(\frac{\delta}{4}\right)^{d-1}\right).$$

Let us set $\delta = \min\left(\frac{1}{4}, 4\left(\frac{2}{n^2}\log\frac{1}{\epsilon}\right)^{\frac{1}{d-1}}\right)$. The above bounds imply that

$$\mathbb{P}(\|\boldsymbol{x}_j - \boldsymbol{x}_k\| > \delta \ \ \forall j, k \in [n]) \leq \epsilon$$

so with probability at least $1 - \epsilon$, there exist $i, k \in [n]$ such that $\|\boldsymbol{x}_i - \boldsymbol{x}_k\| \leq \delta$ with

$$\delta \lesssim \left(n^{-2}\log\frac{1}{\epsilon}\right)^{1/(d-1)}$$

which is what we needed to show. $\qquad\square$

**Corollary 2.** *Let $d \geq 3$, $n \geq 2$, $\epsilon \in (0,1)$, $\boldsymbol{x}_1, \cdots, \boldsymbol{x}_n \sim U(\mathbb{S}^{d-1})$ be mutually iid. Define*

$$\lambda = \left(1 + \frac{\log(n/\epsilon)}{\log(d)}\right)^{-3}\left(\frac{\epsilon^2}{n^4}\right)^{1/(d-1)}.$$

*If $d_1 \gtrsim \frac{1}{\lambda}\left(1 + \frac{n + \log(1/\epsilon)}{d}\right)\log\frac{n}{\epsilon}$, then with probability at least $1 - \epsilon$ over the data and network parameters,*

$$\lambda \lesssim \lambda_{\min}(\boldsymbol{K}) \lesssim \left(\frac{\log(1/\epsilon)}{n^2}\right)^{1/(d-1)}.$$

*Proof.* By Lemma 14, with probability at least $1 - \frac{\epsilon}{4}$,

$$\|\boldsymbol{X}\|^2 \lesssim \left(1 + \frac{n + \log\frac{1}{\epsilon}}{d}\right).$$

Let us denote this event by $\omega_1$. Let us define

$$\delta := \min_{i \neq k} \min(\|\boldsymbol{x}_i - \boldsymbol{x}_k\|, \|\boldsymbol{x}_i + \boldsymbol{x}_k\|)$$

and

$$\delta' := \min_{i \neq k} \|\boldsymbol{x}_i - \boldsymbol{x}_k\|.$$

In particular, the dataset $\boldsymbol{x}_1, \cdots, \boldsymbol{x}_n$ is $\delta$-separated. By Lemma 31, with probability at least $1 - \frac{\epsilon}{4}$,

$$\delta \gtrsim \left(\frac{\epsilon}{n^2}\right)^{1/(d-1)}.$$

Let us denote this event by $\omega_2$. By Lemma 32, with probability at least $1 - \frac{\epsilon}{4}$,

$$\delta' \lesssim \left(\frac{\log(1/\epsilon)}{n^2}\right)^{1/(d-1)}.$$

Let us denote this event by $\omega_3$. We condition on $\omega_1, \omega_2$, and $\omega_3$ for the remainder of the proof. Define

$$\lambda' = \left(1 + \frac{d \log(1/\delta)}{\log(d)}\right)^{-3} \delta^2$$

and

$$\lambda = \left(1 + \frac{\log(n/\epsilon)}{\log(d)}\right)^{-3} \left(\frac{\epsilon^2}{n^4}\right)^{1/(d-1)};$$

note that

$$\lambda' \gtrsim \left(1 + \frac{d \log\left((n^2/\epsilon)^{1/(d-1)}\right)}{\log(d)}\right)^{-3} \left(\frac{\epsilon}{n^2}\right)^{2/(d-1)}$$

$$\gtrsim \left(1 + \frac{\log(n/\epsilon)}{\log(d)}\right)^{-3} \left(\frac{\epsilon^2}{n^4}\right)^{1/(d-1)}$$

$$= \lambda.$$

By Theorem 1, if

$$d_1 \gtrsim \frac{1}{\lambda}\left(1 + \frac{n + \log(1/\epsilon)}{d}\right) \log\left(\frac{n}{\epsilon}\right) \gtrsim \frac{1}{\lambda'}\|\boldsymbol{X}\|^2 \log\left(\frac{n}{\epsilon}\right),$$

then with probability at least $1 - \frac{\epsilon}{4}$ over the network weights,

$$\lambda_{\min}(\boldsymbol{K}) \gtrsim \lambda' \gtrsim \lambda$$

and

$$\lambda_{\min}(\boldsymbol{K}) \lesssim \delta' \lesssim \left(\frac{\log(1/\epsilon)}{n^2}\right)^{1/(d-1)}.$$

This is exactly the bound that we needed to show. By taking a union bound over all of the favorable events, it follows that this event happens with probability at least $1 - \epsilon$. $\qquad\square$

## D    Proof of Theorem 8

### D.1    Recap of the deep setting

Recall for the deep case we consider fully connected networks with $L$ layers and denote the layer widths with positive integers, $d_0, \cdots, d_L$ where $d_0 = d$ and $d_L = 1$. For $l \in [L-1]$ we define the feature matrices $\boldsymbol{F}_l \in \mathbb{R}^{d_l \times n}$ as

$$\boldsymbol{F}_l = [f_l(\boldsymbol{x}_1), \cdots, f_l(\boldsymbol{x}_n)].$$

For $l \in [L-1]$ and $\boldsymbol{x} \in \mathbb{R}^d$ we define the activation patterns $\boldsymbol{\Sigma}_l(\boldsymbol{x}) \in \{0,1\}^{d_l \times d_l}$ to be the diagonal matrices

$$\boldsymbol{\Sigma}_l(\boldsymbol{x}) = \text{diag}(\dot{\sigma}(\boldsymbol{W}_l f_{l-1}(\boldsymbol{x}))).$$

Lemma 9 provides a useful decomposition of the NTK.

**Lemma 9.** *Let $\boldsymbol{x}_1, \cdots, \boldsymbol{x}_n \in \mathbb{R}^d$ be nonzero. There exists an open set $\mathcal{U} \subset \mathcal{P}$ of full Lebesgue measure such that $f(\boldsymbol{x}_i; \cdot)$ is continuously differentiable on $\mathcal{U}$ for all $i \in [n]$. Moreover, for all $\boldsymbol{\theta} \in \mathcal{U}$ the NTK Gram matrix $\boldsymbol{K}$ defined in (1) with network function (7) satisfies*

$$\left(\prod_{l=1}^{L-1} \frac{d_l}{2}\right) \boldsymbol{K} = \sum_{l=0}^{L-1} (\boldsymbol{F}_l^T \boldsymbol{F}_l) \odot (\boldsymbol{B}_{l+1} \boldsymbol{B}_{l+1}^T),$$

*where the ith row of $\boldsymbol{B}_l \in \mathbb{R}^{n \times n_l}$ is defined as*

$$[\boldsymbol{B}_l]_{i,:} = \begin{cases} \boldsymbol{\Sigma}_l(\boldsymbol{x}_i) \left(\prod_{k=l+1}^{L-1} \boldsymbol{W}_k^T \boldsymbol{\Sigma}_k(\boldsymbol{x}_i)\right) \boldsymbol{W}_L^T, & l \in [L-1], \\ \mathbf{1}_n, & l = L. \end{cases}$$

*Proof.* For any $i \in [n]$, observe that $f(\boldsymbol{x}_i, \cdot)$ is a PAP function (Lee et al., 2020b, Definition 5) and therefore $f(\boldsymbol{x}_i, \cdot)$ is differentiable almost everywhere (Lee et al., 2020b, Proposition 4). As the union of $n$ null sets is also a null set, we conclude that there exists an open set $U$ of full measure such that for all $i \in [n]$ then $f(\boldsymbol{x}_i, \boldsymbol{\theta})$ is differentiable for any $\boldsymbol{\theta} \in U$.

Let $\frac{\partial f}{\partial \boldsymbol{\theta}}$ denote the true derivative of $f$ with respect to $\boldsymbol{\theta}$ when it exists and be the minimum norm sub-gradient otherwise. Using (Lee et al., 2020b, Corollary 13) then

$$\left( \prod_{l=1}^{L-1} \frac{d_l}{2} \right) \boldsymbol{K} \stackrel{a.e.}{=} \frac{\partial F_L(\boldsymbol{\theta})}{\partial \boldsymbol{\theta}}^T \frac{\partial F_L(\boldsymbol{\theta})}{\partial \boldsymbol{\theta}} = \sum_{l=1}^{L} \frac{\partial F_L(\boldsymbol{\theta})}{\partial \boldsymbol{W}_l}^T \frac{\partial F_L(\boldsymbol{\theta})}{\partial \boldsymbol{W}_l},$$

where $\frac{\partial F_L(\boldsymbol{\theta})}{\partial \boldsymbol{W}_l} \in \boldsymbol{R}^{d_l d_{l-1} \times n}$. By inspection, to prove the result claimed it therefore suffices to show for any $l \in [L]$, $\boldsymbol{\theta} \in U$ and $i, j \in [n]$ that

$$\langle \frac{\partial f_L(\boldsymbol{x}_i; \boldsymbol{\theta})}{\partial \boldsymbol{W}_l}, \frac{\partial f_L(\boldsymbol{x}_j; \boldsymbol{\theta})}{\partial \boldsymbol{W}_l} \rangle = \left( f_{l-1}(\boldsymbol{x}_i)^T f_{l-1}(\boldsymbol{x}_j; \boldsymbol{\theta}) \right) \left( [\boldsymbol{B}_l]_{i,:}^T [\boldsymbol{B}_l]_{j,:} \right). \tag{21}$$

First observe

$$\langle \frac{\partial f_L(\boldsymbol{x}_i; \boldsymbol{\theta})}{\partial \boldsymbol{W}_L}, \frac{\partial f_L(\boldsymbol{x}_j; \boldsymbol{\theta})}{\partial \boldsymbol{W}_L} \rangle = f_{L-1}(\boldsymbol{x}; \boldsymbol{\theta})^T f_{L-1}(\boldsymbol{x}; \boldsymbol{\theta}) \times 1$$

therefore establishing (21) for $l = L$. To establish (21) for $l \in [L-1]$, recall for $k \in [L-1]$ that $\boldsymbol{\Sigma}_k(\boldsymbol{x}) = \text{diag}\left( \dot{\sigma}(\boldsymbol{W}_k f_{k-1}(\boldsymbol{x})) \right)$ and define $\boldsymbol{\Sigma}_L(\boldsymbol{x}) = 1$. Observe for $1 \le l < k$, $k \in [L]$ that

$$\frac{\partial f_k(\boldsymbol{x}; \boldsymbol{\theta})}{\partial \boldsymbol{W}_l} = \boldsymbol{\Sigma}_k(\boldsymbol{x}) \boldsymbol{W}_k \frac{\partial f_{k-1}(\boldsymbol{x}; \boldsymbol{\theta})}{\partial \boldsymbol{W}_l} \tag{22}$$

while for $k = l$

$$\frac{\partial f_k(\boldsymbol{x}; \boldsymbol{\theta})}{\partial \boldsymbol{W}_k} = \boldsymbol{\Sigma}_k(\boldsymbol{x}) \otimes f_{k-1}(\boldsymbol{x}; \boldsymbol{\theta})^T. \tag{23}$$

As a result,

$$\frac{\partial f_L(\boldsymbol{x}; \boldsymbol{\theta})}{\partial \theta_l} = \boldsymbol{W}_L \left( \prod_{k=1}^{L-l+1} \boldsymbol{\Sigma}_{L-k}(\boldsymbol{x}) \boldsymbol{W}_{L-k} \right) \frac{\partial f_l(\boldsymbol{x}; \boldsymbol{\theta})}{\partial \boldsymbol{W}_l}$$

$$= \boldsymbol{W}_L \left( \prod_{k=1}^{L-l+1} \boldsymbol{\Sigma}_{L-k}(\boldsymbol{x}) \boldsymbol{W}_{L-k} \right) \left( \boldsymbol{\Sigma}_l(\boldsymbol{x}) \otimes f_{l-1}(\boldsymbol{x}; \boldsymbol{\theta}) \right)$$

where the first equality arises from iterating (22) and the second by applying (23). Proceeding,

$$\left\langle \frac{\partial f_L(\boldsymbol{x}_i)}{\partial \theta_l}, \frac{\partial f_L(\boldsymbol{x}_j)}{\partial \theta_l} \right\rangle$$

$$= \left( f_{l-1}(\boldsymbol{x}_i)^T f_{l-1}(\boldsymbol{x}_j) \right) \left( \left( \boldsymbol{\Sigma}_l(\boldsymbol{x}_i) \prod_{k=l+1}^{L-1} \boldsymbol{W}_k^T \boldsymbol{\Sigma}_k(\boldsymbol{x}_i) \right) \boldsymbol{W}_L^T \right)^T \left( \left( \boldsymbol{\Sigma}_l(\boldsymbol{x}_j) \prod_{k=l+1}^{L-1} \boldsymbol{W}_k^T \boldsymbol{\Sigma}_k(\boldsymbol{x}_j) \right) \boldsymbol{W}_L^T \right)$$

$$= \left( f_{l-1}(\boldsymbol{x}_i)^T f_{l-1}(\boldsymbol{x}_j) \right) \left( [\boldsymbol{B}_l]_{i,:}^T [\boldsymbol{B}_l]_{j,:} \right)$$

as claimed. $\qquad \square$

### D.2 Proof of Lemma 10

**Lemma 33.** *Let $\boldsymbol{z} \in \mathbb{R}^d$ be a fixed vector and $\boldsymbol{W} \in \mathbb{R}^{m \times d}$ a random matrix with mutually iid elements $[\boldsymbol{W}]_{ij} \sim \mathcal{N}(0, 1)$ for all $i \in [m]$ and $j \in [d]$. Consider the random vector $\boldsymbol{y} \in \mathbb{R}^m$ defined as $\boldsymbol{y} = \sigma(\boldsymbol{W} \boldsymbol{z})$ where $\sigma$ denotes the ReLU function applied elementwise. For $\delta \in (0, 1)$ if $m \gtrsim \delta^{-2} \log(1/\epsilon)$ then*

$$\mathbb{P}\left( (1 - \delta) \frac{m}{2} \|\boldsymbol{z}\|^2 \le \|\boldsymbol{y}\|^2 \le (1 + \delta) \frac{m}{2} \|\boldsymbol{z}\|^2 \right) \ge 1 - \epsilon.$$

*Proof.* For $i \in [m]$ define $Z_i = \frac{\boldsymbol{w}_i^T \boldsymbol{z}}{\|\boldsymbol{z}\|}$, then $Z_i \sim \mathcal{N}(0, 1)$ are mutually iid. Let $B_i = \mathbb{1}(Z_i > 0)$, note by symmetry $B_i \sim \text{Ber}(1/2)$, furthermore these random variables for $i \in [n]$ are also mutually iid with respect to one another. As $y_i = \|\boldsymbol{z}\| B_i Z_i$ then

$$\|\boldsymbol{y}\|_2^2 = \|\boldsymbol{z}\|^2 \sum_{i=1}^{m} B_i Z_i^2.$$

For convenience let $y' = y/\|z\|$ and define $\mathcal{S} = \{i \in [n] \; : \; B_i = 1\}$, then
$$\|y'\|^2 = \sum_{i \in \mathcal{S}} Z_i^2 \sim \chi^2(|\mathcal{S}|).$$
From (Laurent & Massart, 2000, Lemma 1) we have for any $t > 0$
$$\mathbb{P}\left(| \left(\|y'\|^2 - |\mathcal{S}|\right) | \geq 2\sqrt{|\mathcal{S}|t}\right) \leq 2\exp(-t).$$
For $\delta_1 \in (0, 1)$ let $t = \frac{|\mathcal{S}|\delta_1^2}{4}$, then
$$\mathbb{P}\left((1 - \delta_1)|\mathcal{S}| \leq \|y'\|^2 \leq (1 + \delta_1)|\mathcal{S}|\right) \geq 1 - 2\exp\left(-\frac{|\mathcal{S}|\delta_1^2}{4}\right).$$
Observe $|\mathcal{S}| = \sum_{i=1}^m B_i \sim \text{Bin}(m, 1/2)$. With $\delta_2 \in (0, 1)$ then applying Hoeffding's inequality we have
$$\mathbb{P}\left((1 - \delta_2)\frac{m}{2} \leq \sum_{i=1}^m B_i \leq (1 + \delta_2)\frac{m}{2}\right) \geq 1 - 2\exp\left(-\frac{\delta_2^2 m}{2}\right).$$
Let $\omega$ denote the event that $(1 - \delta_2)\frac{m}{2} \leq |\mathcal{S}| \leq (1 + \delta_2)\frac{m}{2}$. If $m \geq \frac{16}{\delta_1^2 \delta_2^2 (1 - \delta_2)} \log(4/\epsilon)$ then
$$\mathbb{P}\left((1 - \delta_1)(1 - \delta_2)\frac{m}{2} \leq \|y'\|^2 \leq (1 + \delta_1)(1 + \delta_2)\frac{m}{2}\right)$$
$$\geq \mathbb{P}\left((1 - \delta_1)(1 - \delta_2)\frac{m}{2} \leq \|y'\|^2 \leq (1 + \delta_1)(1 + \delta_2)\frac{m}{2} \;\mid\; \omega\right)\mathbb{P}(\omega)$$
$$\geq \mathbb{P}\left((1 - \delta_1)|\mathcal{S}| \leq \|y'\|^2 \leq (1 + \delta_1)|\mathcal{S}| \;\mid\; \omega\right)\mathbb{P}(\omega)$$
$$\geq \left(1 - 2\exp\left(-\frac{(1 - \delta_2)\delta_1^2 m}{8}\right)\right)\left(1 - 2\exp\left(-\frac{\delta_2^2 m}{2}\right)\right)$$
$$\geq \left(1 - \frac{\epsilon}{2}\right)\left(1 - \frac{\epsilon}{2}\right)$$
$$\geq 1 - \epsilon.$$
For some $\delta \in (0, 1)$ let $\delta_2 = \delta_1 = \delta/3$, then if $m \geq 1944\delta^{-2}\log(4/\epsilon)$ we have
$$\mathbb{P}\left((1 - \delta)\frac{m}{2} \leq \|y'\|^2 \leq (1 + \delta)\frac{m}{2}\right) \geq 1 - \epsilon$$
from which the result claimed follows. $\qquad\square$

**Lemma 10.** *Let $x \in \mathbb{S}^{d_0 - 1}$, $L \geq 2$ and $l \in [L - 1]$. If $d_k \gtrsim l^2 \log(l/\epsilon)$ for all $k \in [l]$, then*
$$e^{-1}\left(\prod_{h=1}^l \frac{d_h}{2}\right) \leq \|f_l(x)\|^2 \leq e\left(\prod_{h=1}^l \frac{d_h}{2}\right)$$
*holds with probability at least $1 - \epsilon$ over the network parameters.*

*Proof.* For $k \in [l]$ let $\omega_k$ denote the event that the inequality
$$\left(1 - \frac{1}{l}\right)^k \left(\prod_{h=1}^k \frac{d_h}{2}\right) \leq \|f_k(x)\|^2 \leq \left(1 + \frac{1}{l}\right)^k \left(\prod_{h=1}^k \frac{d_h}{2}\right)$$
holds. We proceed by induction to establish that $\mathbb{P}(\omega_k) \geq (1 - \frac{\epsilon}{l})^k$ for all $k \in [l]$. For the base case note that $f_1(x) = \sigma(W_1 x)$ and $\|x\|^2 = 1$. Applying Lemma 33 with $\delta = \frac{1}{l}$, if $d_1 \gtrsim l^2 \log(l/\epsilon)$ then $\mathbb{P}(\omega_1) \geq 1 - \frac{\epsilon}{l}$. Now suppose for $k \in [l - 1]$ that $\mathbb{P}(\omega_k) \geq (1 - \frac{\epsilon}{l})^k$. Note
$$\mathbb{P}(\omega_{k+1}) \geq \mathbb{P}(\omega_{k+1} \mid \omega_k)\mathbb{P}(\omega_k) \geq \mathbb{P}(\omega_{k+1} \mid \omega_k)(1 - \tfrac{\epsilon}{l})^k$$
Recall $f_{k+1}(x) = \sigma(W_1 f_k(x))$. Conditioned on $\omega_k$, then again applying Lemma 33 with $\delta = \frac{1}{l}$ and as $d_{k+1} \gtrsim l^2 \log(l/\epsilon)$ we have
$$\mathbb{P}(\omega_{k+1} \mid \omega_k) \geq 1 - \tfrac{\epsilon}{l}$$
which completes the proof of the induction hypothesis. As $(1 - \epsilon/l)^l \geq 1 - \epsilon$ and $e^{-1} \leq (1 - 1/l)^l \leq (1 + 1/l)^l \leq e$ then
$$e^{-1}\left(\prod_{h=1}^l \frac{d_h}{2}\right) \leq \|f_l(x)\|^2 \leq e\left(\prod_{h=1}^l \frac{d_h}{2}\right)$$
holds with probability at least $1 - \epsilon$. $\qquad\square$

### D.3 Proof of Lemma 34

**Lemma 34.** *Let $x \in \mathbb{S}^{d_0-1}$, $L \geq 2$ and assume $d_k \gtrsim L^2 \log\left(\frac{L}{\epsilon}\right)$ for all $k \in [L-1]$. For any $l \in [L-1]$ with probability at least $1 - \epsilon$ over the network parameters the following holds,*

$$\|S_l(x)\|_F^2 \asymp 2^{-L+l+1} \prod_{k=l}^{L-1} d_k.$$

*Proof.* In what follows for convenience we define an empty product of scalars or matrices as the scalar one. Let $K \in \{L-1\}$, $l \in [K]$, and for some arbitrary $x \in \mathbb{S}^{d_0-1}$ define

$$S_{l,K} = \Sigma_l(x) \prod_{k=l+1}^{K} W_k^T \Sigma_k(x).$$

Let $\omega_{l,K}$ denote the event

$$\frac{1}{2}\left(1 - \frac{1}{L}\right)^K \leq \|S_{l,K}\|_F^2 \prod_{k=l}^{K} \frac{2}{d_l} \leq 2\left(1 + \frac{1}{L}\right)^K \tag{24}$$

It suffices to lower bound the probability of the event $\omega_{l,L-1}$. Let $\mathcal{F}_K$ denote the $\sigma$-algebra generated by $W_1, \cdots, W_K$ and note that $S_{l,K} \in \mathcal{F}_K$. Let $\gamma_l$ denote the event that $f_l(x) \neq 0$, then

$$\begin{aligned}
\mathbb{P}(\omega_{l,L-1}) &\geq \mathbb{P}(\omega_{l,L-1} \mid \omega_{l,L-2})\mathbb{P}(\omega_{l,L-2}) \\
&\geq \mathbb{P}(\omega_{l,L-1} \mid \omega_{l,L-2})\mathbb{P}(\omega_{l,L-2} \mid \omega_{l,L-3})\mathbb{P}(\omega_{l,L-3}) \\
&\geq \left(\prod_{h=l}^{L-2} \mathbb{P}(\omega_{l,h+1} \mid \omega_{l,h})\right)\mathbb{P}(\omega_{l,l} \mid \gamma_l)\mathbb{P}(\gamma_l).
\end{aligned}$$

Fixing $\epsilon \in (0,1)$, our goal is to show each term in this product is at least $(1 - \frac{\epsilon}{L})$: indeed, if this is true then

$$\mathbb{P}(\omega_{l,L-1}) \geq \left(1 - \frac{\epsilon}{L}\right)^{L-l} \geq 1 - \epsilon$$

and our task is complete. To this end, first observe that as $d_k \gtrsim L^2 \log(L/\epsilon)$ for all $k \in [L-1]$, then $\mathbb{P}(\gamma_l) \geq 1 - \frac{\epsilon}{L}$ by Lemma 10. Proceeding to the term $\mathbb{P}(\omega_{l,l} \mid \gamma_l)$, recall $[\Sigma_l(x)]_{jj} = \mathbb{1}([W_l f_{l-1}(x)]_j > 0)$. By symmetry the diagonal entries of $\Sigma_l(x)$ are mutually iid Bernoulli random variables with parameter $\frac{1}{2}$. Therefore, using Hoeffding's inequality for all $t \geq 0$

$$\mathbb{P}\left(\left|\|\Sigma_l(x)\|_F^2 - \frac{d_l}{2}\right| \geq t \,\bigg|\, \gamma_l\right) \leq 2\exp\left(-\frac{t^2}{d_l}\right).$$

Let $t = d_l$, if $d_l \geq \log \frac{2L}{\epsilon}$ then with $K \geq 1$, $L \geq 2$ it follows that

$$\begin{aligned}
\mathbb{P}(\omega_{l,l} \mid \gamma_l) &= \mathbb{P}\left(\frac{1}{2}\left(1 - \frac{1}{L}\right)^K \leq \|\Sigma_l(x)\|_F^2 \frac{2}{d_l} \leq 2\left(1 + \frac{1}{L}\right)^K \,\bigg|\, \gamma_l\right) \\
&\geq \mathbb{P}\left(\frac{1}{2} \leq \|\Sigma_l(x)\|_F^2 \frac{2}{d_l} \leq \frac{3}{2} \,\bigg|\, \gamma_l\right) \\
&\geq 1 - \mathbb{P}\left(\left|\|\Sigma_l(x)\|_F^2 - \frac{d_l}{2}\right| \geq \frac{d_l}{4} \,\bigg|\, \gamma_l\right) \\
&\geq 1 - \frac{\epsilon}{L}.
\end{aligned}$$

We now proceed to analyze $\mathbb{P}(\omega_{l,h+1} \mid \omega_{l,h})$ for $h \in [l, K-1]$. Note if $\omega_{l,h}$ is true then $\|S_{l,h}\|_F^2 > 0$. By definition this implies $\|\Sigma_l(x)\|_F^2 > 0$, however, if $f_h(x) = 0$ then $\|\Sigma_l(x)\|_F^2 = 0$. Therefore $\omega_{l,h}$ being true implies $f_h(x) \neq 0$. For convenience in what follows we denote the $j$th column of $W_{h+1}$ as $w_j$. By definition

$$S_{l,h+1} = S_{l,h} W_{h+1}^T \Sigma_{h+1}(x),$$

therefore,

$$\mathbb{E}[\|\boldsymbol{S}_{l,h+1}\|_F^2 \mid \mathcal{F}_h] = \mathbb{E}[\|\boldsymbol{S}_{l,h}\boldsymbol{W}_{h+1}^T\boldsymbol{\Sigma}_{h+1}(\boldsymbol{x})\|_F^2 \mid \mathcal{F}_h]$$

$$= \mathbb{E}\left[ \sum_{j=1}^{d_{h+1}} \|\boldsymbol{S}_{l,h}\boldsymbol{w}_j\|^2 \, \dot\sigma(\langle \boldsymbol{w}_j, f_h(\boldsymbol{x})\rangle) \,\middle|\, \mathcal{F}_h \right].$$

As highlighted already, if we condition on $\omega_{l,h}$ then $f_h(\boldsymbol{x}) \neq 0$ and therefore the random variables $(\dot\sigma(\langle \boldsymbol{w}_j, f_h(\boldsymbol{x})\rangle))_{j \in d_{h+1}}$ are mutually iid Bernoulli random variables with parameter $\frac{1}{2}$. Again by symmetry $\dot\sigma(\langle \boldsymbol{w}_j, f_h(\boldsymbol{x})\rangle)$ is independent of $\|\boldsymbol{S}_{l,h}\boldsymbol{w}_j\|^2$. Therefore conditioned on $\omega_{l,h}$

$$\sum_{j=1}^{d_{h+1}} \mathbb{E}[\|\boldsymbol{S}_{l,h}\boldsymbol{w}_j\|^2 \mid \mathcal{F}_{d_{h+1}}]\mathbb{E}[\dot\sigma(\langle \boldsymbol{w}_j, f_h(\boldsymbol{x})\rangle) \mid \mathcal{F}_h] = \frac{1}{2}\sum_{j=1}^{d_{h+1}} \mathbb{E}[\|\boldsymbol{S}_{l,h}\boldsymbol{w}_j\|^2 \mid \mathcal{F}_h]$$

$$= \frac{1}{2}\sum_{j=1}^{d_{h+1}} \|\boldsymbol{S}_{l,h}\|_F^2$$

$$= \frac{d_{h+1}}{2}\|\boldsymbol{S}_{l,h}\|_F^2.$$

Moreover, under the same conditioning

$$\left\| \|\boldsymbol{S}_{l,h}\boldsymbol{w}_j\|^2 \, \dot\sigma(\langle \boldsymbol{w}_j, f_h(\boldsymbol{x})\rangle) \right\|_{\psi_1} \leq \left\| \|\boldsymbol{S}_{l,h}\boldsymbol{w}_j\|^2 \right\|_{\psi_1}$$

$$= \left\| \|\boldsymbol{S}_{l,h}\boldsymbol{w}_j\| \right\|_{\psi_2}^2$$

$$\lesssim \|\boldsymbol{S}_{l,h}\|_F^2$$

where the last line follows from Theorem 6.3.2 of Vershynin (2018). As a result, conditioned on $\omega_{l,h}$ then using Bernstein's inequality (Vershynin, 2018, Theorem 2.8.1) there exists an absolute constant $c$ such that for all $t \geq 0$

$$\mathbb{P}\left( \left| \|\boldsymbol{S}_{l,h+1}\|_F^2 - \frac{d_{h+1}}{2}\|\boldsymbol{S}_{l,h}\|_F^2 \right| \geq t \,\middle|\, \mathcal{F}_h \right) \leq 2\exp\left( -c\min\left( \frac{t^2}{d_{h+1}\|\boldsymbol{S}_{l,h}\|_F^4}, \frac{t}{\|\boldsymbol{S}_{l,h}\|_F^2} \right) \right).$$

If $d_{h+1} \geq \frac{4L^2}{c}\log\frac{2L}{\epsilon}$ and $t = \frac{d_{h+1}\|\boldsymbol{S}_{l,h}\|_F^2}{2L}$ then conditioning on $\omega_{l,h}$ we obtain

$$\mathbb{P}\left( \left| \|\boldsymbol{S}_{l,h+1}\|_F^2 - \frac{d_K}{2}\|\boldsymbol{S}_{l,h}\|_F^2 \right| \geq \frac{d_{h+1}}{2L}\|\boldsymbol{S}_{l,h}\|_F^2 \,\middle|\, \mathcal{F}_h \right) \leq \frac{\epsilon}{L}.$$

As a result, for any $h \in [l, K-1]$ we have $\mathbb{P}(\omega_{l,h+1} \mid \omega_{l,h}) \geq 1 - \frac{\epsilon}{L}$ from which the result claimed follows. $\qquad\square$

### D.4 Proof of Lemma 35

**Lemma 35.** *Let $\boldsymbol{x} \in \mathbb{S}^{d_0-1}$, $L \geq 3$ and assume $d_k \geq d_{k+1}$ and $d_k \gtrsim \sqrt{\log\frac{1}{\epsilon}}$ for all $k \in [L-1]$. For any $l \in [L-1]$ with probability at least $1 - \epsilon$ over the network parameters the following holds,*

$$\|\boldsymbol{S}_l(\boldsymbol{x})\|^2 \lesssim \prod_{k=l}^{L-2} d_k.$$

*Proof.* By Theorem 4.4.5 of Vershynin (2018), for any $k \in [L-1]$ and all $t \geq 0$

$$\mathbb{P}\left( \|\boldsymbol{W}_k\| \leq C(\sqrt{d_{k-1}} + \sqrt{d_k} + t) \right) \geq 1 - 2e^{-t^2}.$$

As $d_{k-1} \geq d_k \geq \sqrt{\log\frac{2L}{\epsilon}}$, then setting $t = \sqrt{\log\frac{2}{\epsilon}}$ yields

$$\mathbb{P}\left( \|\boldsymbol{W}_k\| \leq 3C_1\sqrt{d_{k-1}} \right) \geq \mathbb{P}\left( \|\boldsymbol{W}_k\| \leq C(\sqrt{d_{k-1}} + \sqrt{d_k} + t) \right)$$

$$\geq 1 - \frac{\epsilon}{L}.$$

Using a union bound it follows that

$$\mathbb{P}\left(\|\boldsymbol{W}_k\| \leq 3C_1 \max\{\sqrt{d_{l-1}}, \sqrt{d_l}\} \;\; \forall k \in [L-1]\right) \geq 1 - \epsilon.$$

Note that $\|\boldsymbol{\Sigma}_k(\boldsymbol{x})\| \leq 1$ for all $k \in [L-1]$, therefore conditional on the above event we have

$$\|\boldsymbol{S}_l(\boldsymbol{x})\| = \left\|\boldsymbol{\Sigma}_l(\boldsymbol{x})\left(\prod_{k=l+1}^{L-1} \boldsymbol{W}_k^T \boldsymbol{\Sigma}_k(\boldsymbol{x})\right)\right\|$$

$$\leq \|\boldsymbol{\Sigma}_l(\boldsymbol{x})\|\left(\prod_{k=l+1}^{L-1} \|\boldsymbol{W}_k\|\|\boldsymbol{\Sigma}_k(\boldsymbol{x})\|\right)$$

$$\leq \prod_{k=l+1}^{L-1} \|\boldsymbol{W}_k\|$$

$$\lesssim \prod_{k=l}^{L-2} \sqrt{d_k}.$$

To conclude we square both sides. $\qquad\square$

### D.5 Proof of Lemma 11

**Lemma 11.** *Let $\boldsymbol{x} \in \mathbb{S}^{d_0-1}$, suppose $L \geq 3$, $d_k \geq d_{k+1}$ for all $k \in [L-1]$ and $d_{L-1} \gtrsim 2^L \log\left(\frac{L}{\epsilon}\right)$. Then, for any $l \in [L-1]$, with probability at least $1 - \epsilon$ over the network parameters*

$$\|\boldsymbol{S}_l(\boldsymbol{x})\boldsymbol{W}_L^T\|^2 \asymp 2^{-L+l+1} \prod_{k=l}^{L-1} d_k.$$

*Proof.* Let $\boldsymbol{x} \in \mathbb{S}^{d_0-1}$ be arbitrary and recall $\boldsymbol{S}_l(\boldsymbol{x}) = \boldsymbol{\Sigma}_l(\boldsymbol{x})\left(\prod_{k=l+1}^{L-1} \boldsymbol{W}_k^T \boldsymbol{\Sigma}_k(\boldsymbol{x})\right)$. Also recall that $\boldsymbol{W}_L^T \in \mathbb{R}^{d_{L-1}}$ is distributed as $\boldsymbol{W}_L^T \sim \mathcal{N}(\boldsymbol{0}_{d_{L-1}}, I_{d_{L_1}})$. Therefore by Vershynin (2018, Theorem 6.3.2) for any $\boldsymbol{A} \in \mathbb{R}^{d_2 \times d_{L-1}}$ and $t \geq 0$

$$\mathbb{P}(|\|\boldsymbol{A}\boldsymbol{W}_L^T\|_2 - \|\boldsymbol{A}\|_F| \geq t) \leq 2\exp\left(-\frac{Ct^2}{\|\boldsymbol{A}\|_2^2}\right)$$

for some constant $C > 0$. As a result, with $t = \frac{1}{2}\|\boldsymbol{A}\|_F^2$ then for some constant $C > 0$

$$\mathbb{P}\left(\frac{1}{4}\|\boldsymbol{A}\|_F^2 \leq \|\boldsymbol{A}\boldsymbol{W}_L^T\|_2^2 \leq \frac{3}{4}\|\boldsymbol{A}\|_F^2\right) \geq 1 - \exp\left(-C\frac{\|\boldsymbol{A}\|_F^2}{\|\boldsymbol{A}\|_2^2}\right).$$

Therefore, in order to lower bound $\|\boldsymbol{S}_l(\boldsymbol{x})\boldsymbol{W}_L^T\|_2^2$ with high probability it suffices to condition on a suitable upper bound for $\|\boldsymbol{S}_{L-1}(\boldsymbol{x})\|_2^2$ and a suitable lower bound for $\|\boldsymbol{S}_{L-1}(\boldsymbol{x})\|_F^2$. Let $\omega$ denote the event that both

$$\|\boldsymbol{S}_l\|_F^2 \asymp 2^{L-l-1} \prod_{k=l}^{L-1} d_k$$

and

$$\|\boldsymbol{S}_l(\boldsymbol{x})\|^2 \lesssim \prod_{k=l}^{L-2} d_k$$

are true. Combining Lemmas 34 and 35 using a union bound, then as long as $L \geq 3$, $d_k \geq d_{k+1}$ and $d_k \gtrsim L^2 \log \frac{nL}{\epsilon}$ for all $k \in [L-1]$ then $\mathbb{P}(\omega) \geq 1 - \frac{\epsilon}{2}$. As a result and also as $d_{L-1} \gtrsim 2^L \log(2/\epsilon)$

then

$$\mathbb{P}\left(\|\boldsymbol{S}_l(\boldsymbol{x})\boldsymbol{W}_L^T\|_2^2 \asymp 2^{L-l-1}\prod_{k=l}^{L-1} d_k\right) \geq \mathbb{P}\left(\|\boldsymbol{S}_l(\boldsymbol{x})\boldsymbol{W}_L^T\|_2^2 \asymp 2^{L-l-1}\prod_{k=l}^{L-1} d_k \mid \omega\right)\mathbb{P}(\omega)$$

$$\geq \mathbb{P}\left(\frac{1}{4}\|\boldsymbol{S}_l(\boldsymbol{x})\|_F^2 \leq \|\boldsymbol{S}_l(\boldsymbol{x})\boldsymbol{W}_L^T\|_2^2 \leq \frac{3}{4}\|\boldsymbol{S}_l(\boldsymbol{x})\|_F^2 \mid \omega\right)\mathbb{P}(\omega)$$

$$\geq 1 - \exp\left(-C2^{-L}\frac{\prod_{k=l}^{L-1} d_k}{\prod_{k=l}^{L-2} d_k}\right)\mathbb{P}(\omega)$$

$$\geq 1 - \exp\left(-C2^{-L}d_{L-1}\right)\mathbb{P}(\omega)$$

$$\geq \left(1 - \frac{\epsilon}{2}\right)\left(1 - \frac{\epsilon}{2}\right)$$

$$\geq 1 - \epsilon$$

as claimed. □

### D.6 Proof of Theorem 8

**Theorem 8.** *Suppose $\epsilon \in (0, 1/3)$, $\delta \in (0, \sqrt{2}]$, $d_0 \geq 3$, the data $\boldsymbol{x}_1, \boldsymbol{x}_2, \cdots, \boldsymbol{x}_n \in \mathbb{S}^{d_0-1}$ is $\delta$-separated and define*

$$\lambda = \left(1 + \frac{d_0 \log(1/\delta)}{\log d_0}\right)^{-3}\delta^4.$$

*With regard to the network architecture, let $L \geq 3$, $d_l \geq d_{l+1}$ for all $l \in [L-1]$, $d_{L-1} \gtrsim 2^L \log\left(\frac{nL}{\epsilon}\right)$ and $d_1 \gtrsim \frac{n}{\lambda}\log\left(\frac{n}{\lambda}\right)\log\left(\frac{n}{\epsilon}\right)$. Then with probability at least $1 - \epsilon$ over the network parameters*

$$\lambda \lesssim \lambda_{\min}(\boldsymbol{K}) \lesssim L.$$

*Proof.* Recall (8),

$$2^{L-1}\left(\prod_{l=1}^{L-1}\frac{1}{d_l}\right)\lambda_{\min}\left(\boldsymbol{F}_1\boldsymbol{F}_1^T\right)\min_{i\in[n]}\|[\boldsymbol{B}_2]_{i,:}\|^2 \leq \lambda_{\min}(\boldsymbol{K}) \leq 2^{L-1}\left(\prod_{l=1}^{L-1}\frac{1}{d_l}\right)\sum_{l=0}^{L-1}\|f_l(\boldsymbol{x}_i)\|^2\|[\boldsymbol{B}_{l+1}]_{i,:}\|^2,$$

where the upper bound holds for any $i \in [n]$. We start by analyzing the lower bound. Observe that $\boldsymbol{F}_1\boldsymbol{F}_1^T = \sigma(\boldsymbol{W}_1\boldsymbol{X})^T\sigma(\boldsymbol{W}_1\boldsymbol{X})$ has the same distribution as $d_1\boldsymbol{K}_2$ in the shallow setting; see (3). Let $\lambda_2$ be defined as in Lemma 4:

$$\lambda_2 = d_0\lambda_{\min}\left(\mathbb{E}_{\boldsymbol{u}\sim U(\mathbb{S}^{d_0-1})}\left[\sigma(\boldsymbol{u}^T\boldsymbol{X})^T\sigma(\boldsymbol{u}^T\boldsymbol{X})\right]\right) = \lambda_{min}\left(\boldsymbol{K}_{\sqrt{d_0}\sigma}^\infty\right).$$

As the dataset $\boldsymbol{x}_1, \boldsymbol{x}_2, \cdots, \boldsymbol{x}_n \in \mathbb{S}^{d_0-1}$ is $\delta$-separated then by Lemma 7

$$\lambda_2 \gtrsim \left(1 + \frac{d_0 \log(1/\delta)}{\log d_0}\right)^{-3}\delta^4.$$

Furthermore, if $d_1 \gtrsim \frac{n}{\lambda_2}\log\left(\frac{n}{\lambda_2}\right)\log\left(\frac{n}{\epsilon}\right)$ then by Lemma 4

$$\lambda_{\min}(\boldsymbol{F}_1\boldsymbol{F}_1^T) \gtrsim d_1\lambda_2$$

with probability at least least $1 - \frac{\epsilon}{4}$ and as a result

$$\lambda_{\min}(\boldsymbol{F}_1\boldsymbol{F}_1^T) \gtrsim d_1\left(1 + \frac{\log(n/\epsilon)}{\log(d_0)}\right)^{-3}\delta^4$$

with probability at least $1 - \frac{\epsilon}{4}$. Furthermore, as $L \geq 3$, $d_l \geq d_{l+1}$ for all $l \in [L-1]$ and $d_{L-1} \gtrsim 2^L \log\left(\frac{4nL}{\epsilon}\right)$ then

$$\min_{i\in[n]}\|[\boldsymbol{B}_2]_{i,:}\|^2 \gtrsim 2^{-L}\prod_{k=2}^{L-1} d_k$$

with probability at least $1 - \frac{\epsilon}{4}$. Via a union bound we conclude that the condition

$$2^{L-1} \left( \prod_{l=1}^{L-1} \frac{1}{d_l} \right) \lambda_{\min}(\boldsymbol{F}_1 \boldsymbol{F}_1^T) \min_{i \in [n]} \|[\boldsymbol{B}_2]_{i,:}\|^2 \gtrsim \left( 1 + \frac{\log(n/\epsilon)}{\log(d_0)} \right)^{-3} \delta^4$$

holds with probability at least $1 - \frac{\epsilon}{2}$. Fixing some $i \in [n]$, for the upper bound observe trivially by construction that

$$\|f_0(\boldsymbol{x}_i)\|^2 \|[\boldsymbol{B}_1]_{i,:}\|^2 = 1.$$

By assumption $d_k \gtrsim L^2 \log(4L^2/\epsilon)$ for all $k \in [L-1]$. With $l \in [0, L-1]$ then by Lemma 10

$$\|f_l(\boldsymbol{x}_i)\|^2 \lesssim 2^{-l} \prod_{k=1}^{l} d_k$$

holds with probability at least $1 - \frac{\epsilon}{4L}$. Likewise by Lemma 34 for $l \in [2, L]$,

$$\|[\boldsymbol{B}_l]_{i,:}\|^2 = \|\boldsymbol{S}_l(\boldsymbol{x}_i) \boldsymbol{W}_L^T\| \lesssim 2^{-L+l+1} \prod_{k=l}^{L-1} d_k$$

with probability at least $1 - \frac{\epsilon}{4L}$. Combining these via a union bound then for any $l \in [0, L-1]$,

$$\|f_l(\boldsymbol{x}_i)\|^2 \|[\boldsymbol{B}_l]_{i,:}\|^2 \lesssim 2^{-L+1} \prod_{k=1}^{L-1} d_k$$

holds with probability at least $1 - \frac{\epsilon}{2L}$. Again using a union bound now over the layers, it follows that

$$2^{L-1} \left( \prod_{l=1}^{L-1} \frac{1}{d_l} \right) \sum_{l=0}^{L-1} \|f_l(\boldsymbol{x}_i)\|^2 \|[\boldsymbol{B}_{l+1}]_{i,:}\|^2 \lesssim L 2^{L-1} \left( \prod_{l=1}^{L-1} \frac{1}{d_l} \right) 2^{-L+1} \left( \prod_{l=1}^{L-1} d_l \right) = L \quad (25)$$

with probability at least $1 - \frac{\epsilon}{2}$. As a result, using a final union bound we conclude both the upper and lower bounds hold with probability at least $1 - \epsilon$. $\qquad \square$

