# OpenReview forum: "Bounds for the smallest eigenvalue of the NTK for arbitrary spherical data of arbitrary dimension"
_NeurIPS.cc/2024/Conference — NeurIPS 2024 poster_

### Official Review · Reviewer_Hffy · 2024-07-02

**Soundness:** 3
**Presentation:** 3
**Contribution:** 3
**Rating:** 7
**Confidence:** 3

**Summary:**

This paper gives a lower bound of the smallest eigenvalue of the NTK matrix for shallow and deep connected ReLU networks through the application of hemisphere transform.

**Strengths:**

The main significance of this paper is dropping the requirement of the input data dimension from [Nguyen2021] on the same topic, enabling more flexible application in other machine learning problems. The main result is presented with a clear step-by-step proof sketch.


Reference:
- *Quynh Nguyen, MarcoMondelli, and GuidoMontúfar. Tight bounds on the smallest eigenvalue of the neural tangent kernel for deep ReLU networks. In Proceedings of the 38th International Conference on Machine Learning, volume 139 of Proceedings of Machine Learning Research, pp. 8119–8129. PMLR, 18–24 Jul 2021.*

**Weaknesses:**

I do not see any major weakness of this paper. It could be nicer if this paper also offers experimental results to support their calm.

**Questions:**

How tight is the lower bound shown in Theorem 1 where the quantity $\lambda$ is defined in terms of the $\delta$-seperated-ness of the data, besides the case where data distributed uniformly on the sphere mentioned in line 152-153?

**Limitations:**

This paper is theoretical and limitations are stated clearly in the paragraphs (lines 317-320).

---

> ### Author Rebuttal · Authors · 2024-08-07
>
> We are grateful for the positive feedback. In regard to your question,
>
> > How tight is the lower bound shown in Theorem 1 where the quantity  is defined in terms of the $\delta$-seperatedness of the data, besides the case where data distributed uniformly on the sphere mentioned in line 152-153,
>
> this is an interesting question and one we hope to pursue in future works. In particular, for rotationally invariant distributions reasonable estimates of $\delta$ holding with high probability are relatively easy to derive. For more general distributions however this is more challenging.

---

> > ### Comment · Reviewer_Hffy · 2024-08-09
> >
> > Thank you for your answer. After reading the other reviews, I would tend to accept this paper as it makes the first step to bound for the smallest eigenvalue of the NTK with arbitrary input dimension, under the condition that the authors include the discussion in other reviews.

---

### Official Review · Reviewer_qjpA · 2024-07-09

**Soundness:** 3
**Presentation:** 3
**Contribution:** 3
**Rating:** 6
**Confidence:** 5

**Summary:**

This theory paper fits within a general framework in which one tries to get information on training of deep learning models using the formalism of the so-called Neural Tangent Kernel.

Specifically, the topic is smallest eigenvalue control for the NTK kernel, and the authors study the minimum eigenvalue under the assumption that one has datasets of unit norm and assume that the data are "well spread" i.e. they have controlled separation constants (and sometimes controlled covering, meaning the data are "uniformly spread"). The obtained bounds depend on this separation constant and on the input and output space dimensions.

The technique uses the so-called hemisphere transform and basic harmonic analysis on the sphere. These methods have not been used before for this particular problem.

**Strengths:**

The studied problem is arguably relevant for dynamical study of NN evolution.

The techniques used are innovative within this field.

**Weaknesses:**

The main weakness is that requirement on the data distribution to be delta-separated is not as "harmless" or "general" as the authors claim (furthermore, I did not find a justification of this claim in the paper; the authors just state that $\delta$-separation is "milder" than previous work requirements, without explaining why and without verifying that).
In practice, it is not trivial to ensure that a sample from a data distribution is well separated in the sense of theorem 8, or a Delone set with controlled constants, making it uniformly separated in the sense of Theorem 1. The assumption of iid data is in practice easier to justify, and checking for delta-separation may be itself a hard problem.

**Questions:**

Main question:

A step for a good comparison to previous work is in having formulated Corollary 2, that is an "iid data analogue" of the main result of thm 1. However it is not clear how the bounds from previous works compare to this result. I suggest to put some effort to explicit this comparison in the most explicit way possible.

Other minor observations and questions:
1) the notion of "$\delta$-separatedness" is a terminology used in the community for the case that points are at minimum distance larger or equal than $\delta$. The notion is not the same as used in this paper, and defined in line 44. Also, at 3 instances in the paper the notion of "collinearity" is used, which is a bit misleading: any two points are collinear. So I suggest to replace "collinearity" with something more explicit such as "being on the same line through the origin" and that the notion of "$\delta$-separated" is either called by a different name, or that it be emphasized the difference with the usual notion.

 2) In the paragraph before line 39, there is an instance in which $\mathbb R^{d\times n}$ should be replaced by $\mathbb R^{d_0\times n}$.

 3) Section 2 has a large overlap with the introduction. Could it be shortened or merged?

4) lines 141-142, about data in $\mathbb S^1$: this sentence is not clear to me, and it is not clear how passing from $\mathbb S^1\subset \mathbb R^2$ to $\mathbb S^1\times\{0\}\subset \mathbb R^3$ affects the constant $\delta'$ from Theorem 1.

5) line 243-244 and lines 313-315: the fact that data are required to be $\delta$-separated has to be mentioned, since it restricts generality.

**Limitations:**

The main concerns were mentioned in the "weaknesses" part.

---

> ### Author Rebuttal · Authors · 2024-08-07
>
> We thank the reviewer for their overall positive feedback and are confident we can address each of the concerns raised. In light of our responses below we hope the reviewer will consider increasing their score.
> > The main weakness is that requirement on the data distribution to be delta-separated is not as "harmless" or "general" as the authors claim (furthermore, I did not find a justification of this claim in the paper; the authors just state that $\delta$-separation is "milder" than previous work requirements, without explaining why and without verifying that).
>
> In order for a dataset to be $\delta$-separated for some value of $\delta\neq 0$, it is sufficient for the data points to not be scalar multiples of one another. Compared to other works, which depend on the data being drawn iid from specific families of high-dimensional probability distributions, we believe that this condition is a relaxation. Note $\delta \neq 0$ is necessary for $\lambda\_{min}>0$. Furthermore, all datasets drawn mutually iid from a continuous distribution are almost surely $\delta$-separated for some $\delta>0$, but not all $\delta$-separated datasets must be drawn mutually iid from some high distribution. Of course, the strength of our results depends on the actual amount of separation between data points, however we remark that this issue is also implicitly present for the prior works, e.g., they require $d = \Omega(\log(n))$. We remark that our goal in including Corollary 2 was to give a sense of the implications of this assumption for a specific probability distribution. We will make this point more clear in the final version.
>
> > In practice, it is not trivial to ensure that a sample from a data distribution is well separated in the sense of Theorem 8, or a Delone set with controlled constants, making it uniformly separated in the sense of Theorem 1.
>
> There could indeed be challenges in computing good bounds for $\delta$ for which certain probability distributions are $\delta$-separated with high probability. Improving the dependence of our bounds on the $\delta$ particularly with regards to Theorem 8 is an important avenue for future work.
>
> > the notion of "$\delta$-separatedness" is a terminology used in the community for the case that points are at minimum distance larger or equal than $\delta$. The notion is not the same as used in this paper, and defined in line 44.
>
> Thank you for pointing this out. We will emphasize that our notion of $\delta$-separatedness is different from definitions used in some other works. We decided to use this terminology following Assumption 1 of [1].
>
> > Also, at 3 instances in the paper the notion of "collinearity" is used, which is a bit misleading: any two points are collinear. So I suggest to replace "collinearity" with something more explicit such as "being on the same line through the origin
>
> Thank you for highlighting this potential source of confusion, we will seek to rectify this point in future revisions of the paper.
>
> > In the paragraph before line 39, there is an instance in which $\mathbb{R}^{d \times n}$ should be replaced by $\mathbb{R}^{d_0 \times n}$.
>
> Thanks for noting this typo. We will fix it in the final version.
>
> > Section 2 has a large overlap with the introduction. Could it be shortened or merged?
>
> Thanks for the feedback. Section 2 provides a background and deep dive on the results in related work, the Introduction was intended to introduce and motivate the topic. In particular, we felt it important to clarify the contributions of this work versus that of Bombari et al and Nguyen et al. However, we will certainly look at ways to shorten and combine both sections.
>
>
> > 4. lines 141-142, about data in $\mathbb{S}^1$: this sentence is not clear to me, and it is not clear how passing from $\mathbb{S}^1 \subset \mathbb{R}^2$ to $\mathbb{S}^1 \times 0 \in \mathbb{R}^3$
>  affects the constant
>  from Theorem 1.
>
> The purpose of this comment was to address the constraint $d \geq 3$ in Theorem 1. In principle, one could take data in 1 or 2 dimensions and pad it with zeros so that it lies in $\mathbb{S}^2$ and Theorem 1 applies. All other aspects of the theorem would be unaffected, including the constant $\delta'$. This is not a particularly interesting case, but we mentioned it for the sake of completeness.
>
> > line 243-244 and lines 313-315: the fact that data are required to be
> $\delta$-separated has to be mentioned, since it restricts generality.
>
> We will clarify this point in the final version.
>
> [1] Oymak, Samet, and Mahdi Soltanolkotabi. "Toward moderate overparameterization: Global convergence guarantees for training shallow neural networks." IEEE Journal on Selected Areas in Information Theory 1.1 (2020): 84-105.

---

> > ### Comment · Reviewer_qjpA · 2024-08-09
> >
> > Thank you for the rebuttal, as indicated in the original review I considered the paper acceptance-worthy anyway.
> >
> > About the $\delta$-uniform-separation versus earlier approaches, after some thought I think that the main question is regarding the counterpart to Corollary 2 obtainable by previous work. Corollary 2 gives one estimate for iid uniform data, but with a nontrivial dependence on  a comparison to previous results without passing through $\delta$-separation would make a good addition to the main text. Is it possible to directly compare the bound from this Corollary to what one would get by using prior work?
> >
> > Since the property $P(\delta,\epsilon,n)=$"$\delta$-separation holds with probability $\ge 1-\epsilon$ on a sample on $n$ i.i.d. points in the unit sphere" is true only in a nontrivial shaped region in $(\delta,\epsilon,n)$-space, it's hard (for me, and I'm sure for the average reader) to really check whether your "metric/banach geometry based" result is or is not better than earlier "probabilistic based" results that you cite in the introduction.
> >
> > So my main question on the topic of what I (pompously) called "the main weakness" in the original review is: Is it possible to give a counterpart to the bound of Corollary 2 using the methods from previous works? (if not, I still think the score is above acceptance level anyway)

---

### Official Review · Reviewer_k95g · 2024-07-10

**Soundness:** 3
**Presentation:** 3
**Contribution:** 3
**Rating:** 6
**Confidence:** 3

**Summary:**

This work provides bounds on the smallest eigenvalue of the Neural Tangent Kernel corresponding to fully connected ReLU networks trained on data supported on spheres. The novelty is that usual assumptions coupling the input data dimension to the sample size are able to be weakened. Similarly, assumptions on the data distribution are replaced by a condition on its realization, that the datapoints are $\delta$-separated.

**Strengths:**

This paper is well written and provides a novel result that gives valuable insight into the behaviour of the NTK.

**Weaknesses:**

The major weaknesses of the paper are addressed by the authors and provide avenue for future research.

**Questions:**

None

**Limitations:**

The authors have addressed the limitations of the work. This is a theoretical paper so the broader societal impact is negligible.

---

### Official Review · Reviewer_fqQK · 2024-07-11

**Soundness:** 3
**Presentation:** 3
**Contribution:** 3
**Rating:** 6
**Confidence:** 5

**Summary:**

This paper investigates the neural network optimization and memorization  in terms of the bounds on the smallest eigenvalue of NTK, without requiring distributional assumptions on the data. The theoretical results are technically sound and contribute to the understanding of neural network convergence behavior.

**Strengths:**

1. The bounds hold without requiring the distributional assumptions on the data and being applicable to high-dimensional settings.

2. The authors introduce a novel application of the hemisphere transform and the addition formula for spherical harmonics, which serves as an innovative approach to analyzing the NTK.

3. The results are applicable to both shallow and deep networks.

**Weaknesses:**

1. The current results are constrained to scenarios where the activation function is exclusively ReLU, which potentially limits the applicability. What is the primary impediment to generalizing the current results ?

2. The structure of the paper needs to be improved. The main conclusion are presented in Theorem 1 and 8, but it takes a lot of space to present the proof sketch in the corresponding section, more discussions on how the upper/lower bounds influence the performance of the DNNs should be included ?

3. Theorem 8 requires the layer width satisfy a pyramidal condition, I wonder whether modern DNNs architecture fullfills this requirement?

4. I acknowledge the theoretical contributions made by the authors, but  I would recommend the authors to add some empirical studies to support their theoretical claims.

minors:
$X \in \mathbb{R}^{d \times n}$ should be $X \in \mathbb{R}^{d_0 \times n}$

**Questions:**

Please refer to weaknesses

---

> ### Author Rebuttal · Authors · 2024-08-07
>
> We thank the reviewer for their positive feedback and thoughtful comments. We are confident that we can address your concerns and hope in light of our responses the reviewer might consider increasing their score.
>
> > The current results are constrained to scenarios where the activation function is exclusively ReLU, which potentially limits the applicability
>
> We do indeed restrict ourselves to the ReLU activation. There are a couple of places where ReLU plays an important role in our proofs, which may shed light on generalizations to other activations. First, we use the scale-invariance of ReLU to reduce expectations over Gaussian measures to expectations over the sphere, as in Lemmas 3 and 4. As we remark after the statements of these lemmas, this reduction allows us to work with integrals over the sphere and spherical harmonics, for which there is an abundance of theory we can port over. For non-homogeneous activation functions, we suspect that one could apply similar results from harmonic analysis on $L^2(\mathbb{R}^d, \nu)$ in place of $L^2(\mathbb{S}^{d - 1})$ (where $\nu$ is a Gaussian probability measure). Second, our bounds rely on computing Fourier coefficients for ReLU in Lemmas 20 and 21. One could compute coefficients in a similar manner for other specific activation functions such as leaky ReLU or polynomials, but in general computing these integrals exactly is not feasible. A possible solution is to bound the integrals in terms of $r, d$, and regularity properties of the activation.
> > The structure of the paper needs to be improved. The main conclusion are presented in Theorem 1 and 8, but it takes a lot of space to present the proof sketch in the corresponding section, more discussions on how the upper/lower bounds influence the performance of the DNNs should be included
>
> We are happy to receive any feedback or ideas as to how the presentation and content of the paper can be improved and will incorporate it into any revised version. Currently we summarize both Theorem 1 and 8 in the introduction in order to give the reader an overview of our contributions straight away. Our key technical innovations are for shallow networks in Section 3, which we then extend to deep networks using more standard techniques in Section 4. As a result we spend significant space describing these new techniques in Section 3. In general, as the key contributions of this paper are theoretical in nature we wanted to provide detailed proof sketches.
>
> > Theorem 8 requires the layer width satisfy a pyramidal condition, I wonder whether modern DNNs architecture fulfills this requirement
>
> The reviewer correctly points out that the strict pyramidal assumption does not necessarily hold in practice. However, given our key technical innovations are for shallow networks, we chose to adopt this assumption for simplicity in order to highlight a way in which our results can be extended to cover deep networks. We note that the strict pyramidal condition is convenient as it allows us to easily bound products involving factors of the form $\sqrt{d_l} + \sqrt{d\_{l-1}}$, however this condition could likely be relaxed to a weak pyramidal condition or potentially removed altogether.

---

> ### Comment · Reviewer_fqQK · 2024-08-10
> **Response**
>
> Thank you for the rebuttal. My concerns have been well addressed, and I would like to keep my positive score of this paper.

---

### Official Review · Reviewer_NBnj · 2024-07-12

**Soundness:** 3
**Presentation:** 3
**Contribution:** 3
**Rating:** 6
**Confidence:** 3

**Summary:**

The paper derives new bounds on the smallest eigenvalue in NTK kernel matrices crucially used in the analyses of neural network training and generalization. Hereby it uses new analytical techniques. One main point improving over most previous bounds is that they are widely distribution independent. The only (standard) assumptions are that data lies on the unit sphere, and they are not too collinear as introduced by Oymak & Soltanolkotabi, 2020. It seems that the new results improve over previous work Banerjee et al., 2023. But I am missing direct comparisons which could be addressed in the rebuttal.

**Strengths:**

* widely distribution independent bounds
* only standard assumptions
* linear dependence on network width
* extension to multilayer NN

**Weaknesses:**

* limited to ReLU activation
* missing out some direct comparisons to previous bounds
* study is motivated by NN optimization, but only the initialization phase is actually considered

**Questions:**

* how do the new bounds affect the full (S)GD optimization e.g. for squared loss? (for instance while the width is linear for your bounds, many previous analyses require larger polynomial width in the course of optimization, even where low width suffices in the initialization)
* how do the new bounds relate to existing bounds for classification under cross entropy loss? For instance the delta dependence that is between linear and quadratic in Thm 1 reminds closely of the results in https://arxiv.org/abs/2206.12802 that are also between linear and quadratic in gamma, the separation margin in NTK for classification (which seems closely related to your delta).
* any idea on resolving the linear/quadratic gap?

**Limitations:**

-

---

> ### Author Rebuttal · Authors · 2024-08-07
>
> We thank the reviewer for their overall positive feedback on our work. We are confident that we are able to address your concerns and hope in light of our responses below that you might consider raising your score. First in regards to the highlighted weaknesses we offer the following comments.
> > limited to ReLU activation
>
> We do indeed restrict ourselves to the ReLU activation. There are a couple of places where ReLU plays an important role in our proofs, which may shed light on generalizations to other activations. First, we use the scale-invariance of ReLU to reduce expectations over Gaussian measures to expectations over the sphere, as in Lemmas 3 and 4. As we remark after the statements of these lemmas, this reduction allows us to work with integrals over the sphere and spherical harmonics, for which there is an abundance of theory we can port over. For non-homogeneous activation functions, we suspect that one could apply similar results from harmonic analysis on $L^2(\mathbb{R}^d, \nu)$ in place of $L^2(\mathbb{S}^{d - 1})$ (where $\nu$ is a Gaussian probability measure). Second, our bounds rely on computing Fourier coefficients for ReLU in Lemmas 20 and 21. One could compute coefficients in a similar manner for other specific activation functions such as leaky ReLU or polynomials, but in general computing these integrals exactly is not feasible. A possible solution is to bound the integrals in terms of $r, d$, and regularity properties of the activation.
> > missing out some direct comparisons to previous bounds
>
> In the related works Section 2 of the paper we  highlight Nguyen et al. and Bomari et al. as the most relevant works and discuss the bounds they provide. We emphasize that these bounds are hard to compare like-for-like as they hold / are non-vacuous under different assumptions. Our work provides bounds on the smallest eigenvalue of the NTK in a more general setting than the prior works with respect to two important aspects: first and unlike prior works we make no distributional assumptions on the data, instead we express our bounds in terms of the smallest difference between two points. Second we do not require the input dimension to scale with the number of samples; prior works cannot cover this classical statistical setting. In particular, as long as the first two layer widths satisfy $d\_0d\_1 = \Omega(n^{1 + 4/(d\_0-1)})$ then $\lambda\_{\min} = \Omega(n^{- 4/(d\_0-1)})$ with high probability. This bound holds as long as $d\_0\geq 3$ and furthermore if $d_0 = \Omega(\log(n))$ then it is tight up to logarithmic factors and matches the bounds provided in the prior works (which require $d\_0 = \Omega(\log(n))$ to hold in the first place).
> > study is motivated by NN optimization, but only the initialization phase is actually considered.
>
> To recap the standard procedure for proving global optimization guarantees via the NTK: 1) bound the distance from the parameters to their position at initialization throughout training using a real induction argument, 2) use the previous step to lower bound the smallest eigenvalue of the NTK at any time $t$ in terms of the value of the smallest eigenvalue at initialization, 3) lower bound the smallest eigenvalue of the NTK at initialization. Our work covers step 3 and is therefore a key ingredient of this pipeline for proving global convergence guarantees.
>
>
>
> Next, in regard to the questions:
>
> > how do the new bounds affect the full (S)GD optimization e.g. for squared loss? (for instance while the width is linear for your bounds, many previous analyses require larger polynomial width in the course of optimization, even where low width suffices in the initialization)
>
> The reviewer correctly points out that the key bottleneck in terms of width requirements originates from the real inductive step used to ensure the network parameters remain in some ball around their initialization. Therefore, even if linear width suffices to bound the smallest eigenvalue of the NTK at initialization one still requires polynomial width to use this to lower bound the smallest eigenvalue of the NTK throughout training. However, even with polynomial width, without a quantitative lower bound on the smallest eigenvalue of the NTK at initialization it is not possible to provide a quantitative bound on the rate of convergence. Our work provides such a bound and does so under general assumptions on the input data, thereby extending the applicability of the framework as a whole.
>
> > how do the new bounds relate to existing bounds for classification under cross entropy loss? For instance the delta dependence that is between linear and quadratic in Thm 1 reminds closely of the results in https://arxiv.org/abs/2206.12802 that are also between linear and quadratic in gamma, the separation margin in NTK for classification (which seems closely related to your delta).
> any idea on resolving the linear/quadratic gap?
>
> The work referenced by the reviewer shows for a two-layer ReLU network that when running gradient descent on binary classification using a coupled initialization technique, width $m = O(\gamma^{-2} \log n)$ suffices to achieve arbitrarily small training error and $m =\Omega(\gamma^{-1})$ is necessary, where $\gamma$ is a quantity that determines the separation margin of the NTK. Their discussion indicates that the coupled initialization is a key ingredient allowing them to reduce width requirements to $m=\Omega(\gamma^{-2}\log n)$ when bounding the logistic loss averaged over $T$ gradient descent iterations. Thus there appear to be differences in the settings (coupled initialization), objects originating the width requirements (loss along gradient descent), as well as in the considered quantities (separation margin of the NTK). Nonetheless, we agree that exploring possible connections between the linear and quadratic dependencies might be interesting and thank the reviewer for the pointer.

---

> > ### Comment · Reviewer_NBnj · 2024-08-12
> >
> > Thank you for your response. I am happy to raise my score by one point, conditioned on more thorough discussion to the related work, focusing on similarities, that could potentially benefit either lines of work, rather than focusing on differences in details of their settings

---

> > > ### Author Response · Authors · 2024-08-13
> > >
> > > We thank the reviewer for the discussion and for helping us improve our work. We will gladly endeavor to incorporate the highlighted points into our next revision.

---

### Decision · Program_Chairs · 2024-09-25

**Decision:**

Accept (poster)

**Comment:**

A fine work deriving bounds for the smallest eigenvalue of the NTK that hold under relaxed distributional assumptions. The reviewers have unanimously voted for acceptance.